# Markedly different impacts of primary emissions and secondary aerosol formation on aerosol mixing states revealed by simultaneous measurements of CCNC, V/HTDMA, and SP2

Jiangchuan Tao[1,8], Biao Luo[1,8], Weiqi Xu[3], Gang Zhao[6], Hanbin Xu[5], Biao Xue[1,8], Miaomiao Zhai[1,8], Wanyun Xu[4], Huarong Zhao[7], Sanxue Ren[7], Guangsheng Zhou[7], Li Liu[2,*], Ye Kuang[1,8,*], Yele Sun[3]

[1] Institute for Environmental and Climate Research, Jinan University, Guangzhou, Guangdong, China

[2] Key Laboratory of Regional Numerical Weather Prediction, Institute of Tropical and Marine Meteorology, China Meteorological Administration, Guangzhou, China.

[3] State Key Laboratory of Atmospheric Boundary Layer Physics and Atmospheric Chemistry, Institute of Atmospheric Physics, Chinese Academy of Sciences, Beijing, China.

[4] State Key Laboratory of Severe Weather, Key Laboratory for Atmospheric Chemistry, Institute of Atmospheric Composition, Chinese Academy of Meteorological Sciences, Beijing, China

[5] Experimental Teaching Center, Sun Yat-Sen University, Guangzhou, China

[6] State Key Joint Laboratory of Environmental Simulation and Pollution Control, International Joint Laboratory for Regional Pollution Control, Ministry of Education, College of Environmental Sciences and Engineering, Peking University, Beijing 100871, China

[7] Hebei Gucheng Agricultural Meteorology National Observation and Research Station, Chinese Academy of Meteorological Sciences, Beijing, 100081, China

[8] Guangdong-Hongkong-Macau Joint Laboratory of Collaborative Innovation for Environmental Quality, Jinan University, Guangzhou, Guangdong, China

Correspondence: Ye Kuang (kuangye@jnu.edu.cn), Li Liu (liul@gd121.cn)

**Abstract**

This study compares aerosol mixing state parameters obtained via simultaneous measurements using DMA-CCNC, H/V-TDMA, and DMA-SP2, shedding light on the impacts of primary aerosol emissions and secondary aerosol (SA) formation. The analysis reveals significant variations in mixing-state parameters among different techniques, with V-TDMA and DMA-SP2 indicating that non-volatile particles mainly stem from BC-containing aerosols, while a substantial proportion of nearly hydrophobic aerosols originates from fossil fuel combustion and biomass burning emissions. Synthesizing the results, some nearly hydrophobic BC-free particles were found to be CCN-inactive under the measured supersaturated conditions, likely from fossil fuel combustion emissions, while others were CCN-active, linked to biomass burning emissions. Moreover, BC-containing aerosols emitted from fossil fuel combustion exhibit more external mixing with other aerosol components compared to those from biomass burning. Secondary nitrate and organic aerosol formation significantly affect aerosol mixing states, enhancing aerosol hygroscopicity and volatility while reducing heterogeneity among techniques. The study also highlights distinct physical properties of two resolved secondary organic aerosol factors, hinting at formation through different mechanisms. These findings underscore the importance of comparing aerosol mixing states from different techniques as a tool in understanding aerosol physical properties from different sources and their responses to SA formation, as well as aiding in the exploration of SA formation mechanisms.

## 1 Introduction

The aerosol mixing state is a crucial physicochemical property of aerosol particles (Riemer et al., 2019), exerting a significant impact on their optical properties and cloud condensation nuclei (CCN) activity, thus affecting their impact on the climate and environment (Fierce et al., 2017; Riemer et al., 2019; Stevens et al., 2022). For example, variations in the mixing state of black carbon (BC) particles can significantly alter their absorption and radiative effects (Bond et al., 2013; Lack et al., 2012; Zhao et al., 2019; Moffet et al., 2016; Matsui et al., 2018; Peng et al., 2016). Using simple internal mixing state assumptions for aerosol chemical components to estimate CCN number concentrations can lead to substantial overestimations (up to 30%; Deng et al., 2013; Farmer et al., 2015; Ren et al., 2018; Ching et al., 2017, 2019; Tao et al., 2021). The aerosol mixing state varies widely due to complex emissions and atmospheric transformations, leading to significant uncertainties in estimating the effects of aerosols based on simplified mixing state assumptions (Ervens, 2015; Wang et al., 2022; Fu et al., 2022).

The aerosol mixing state describes the mixture of aerosol chemical components within each particle and the distribution of these particles in the aerosol population. This property can be directly measured using single-particle chemical composition techniques (Fierce et al., 2017; Riemer et al., 2019), such as the single-particle soot photometer (SP2), which measures refractory black carbon (rBC) mass concentrations and the mixing state of rBC with other aerosol components, or single-particle chemical composition measurement techniques (e.g., single-particle aerosol mass spectrometer, SP-AMS) that have been developed in recent years (Lee et al., 2019; Riemer et al., 2019 and reference therein). Alternatively, the aerosol mixing state can be inferred from indirect measurements of aerosol properties, such as size-resolved aerosol CCN activity (measured by coupling a differential mobility analyzer (DMA) and a CCN counter (CCNC)), size-resolved aerosol hygroscopicity distributions, or volatility distributions (measured by a Humidified/Volatility Tandem differential mobility analyzer (H/V-TDMA)).

However, each technique yields information on aerosol mixing states based on different aerosol microphysical properties, thus obtaining aerosol mixing states that are different but linked to one another. For instance, while both CCN activity and hygroscopic growth measurements are associated with aerosol hygroscopicity, an intercomparison between CCNC and HTDMA measurements has

prompted investigations into aerosol hygroscopicity variations under different saturation conditions (Su et al., 2010; Juranyi et al., 2013; Lance et al., 2013; Kawana et al., 2016; Tao et al., 2020; Jiang et al., 2021). Although the SP2 and VTDMA techniques depend on the evaporation of non-refractory compositions, only rBC remains in the SP2 measurements. In contrast, non-refractory composition evaporation depends on the thermodynamic temperature in the VTDMA measurements. Thus, measurements of an SP2 are highly correlated to those of a VTDMA at high temperatures (200 °C–300 °C ), with their differences reflecting variations in aerosol density, shape, or volatility (Philippin et al., 2004; Wehner et al., 2009; Adachi et al., 2018, 2019; Wang et al., 2022). HTDMA and VTDMA can be combined to study the influence of the aerosol mixing state on hygroscopicity and volatility (Zhang et al., 2016; Cai et al., 2017; Wang et al., 2017). Strong correlations were found between the hydrophobic and non-volatile particles, suggesting they might have similar chemical compositions (Zhang et al., 2016). In addition, some studies have shown that, except for BC, low-volatility particles correlate well with CCN-inactive particles based on VTDMA and CCNC measurements (Kuwata et al., 2007; Kuwata and Kondo, 2008; Rose et al., 2011; Cheng et al., 2012). Therefore, intercomparisons between mixing state parameters measured by distinct techniques provide a better characterization of the aerosol mixing state and insight into aerosol physiochemical properties. Previous studies have mainly compared two types of aerosol mixing state measurements and lacked a comprehensive comparative analysis among SP2, DMA-CCN, and HV-TDMA measurements, hindering the wide application of derived aerosol mixing states obtained by individual techniques.

The mixing state of primary aerosols can vary greatly depending on their type and emission conditions (Cheng et al., 2012; Wang et al., 2017; Wang et al., 2022; Ting et al., 2018; Liu et al., 2021) and can be significantly altered during aging processes or secondary formation (Wehner et al., 2009; Cheng et al., 2012; Wang et al., 2022; Tomlin et al., 2021; Lata et al., 2021). Primary aerosol emissions, such as biomass burning, fossil fuel combustion, and cooking, tend to contribute to weak hygroscopicity (Herich et al., 2008, 2009; Wang et al., 2020; Kim et al., 2020) and low-volatility aerosols (Hong et al., 2017; Saha et al., 2018). The formation of secondary aerosols (SAs), including the aging of BC-containing particles and primary organic aerosols, mainly contributes to aerosols with strong CCN activity (Mei et al., 2013; Ma et al., 2016; Tao et al., 2021) and high hygroscopicity (Chen et al., 2018; Kim et al., 2020; Wang et al., 2020). It is important to study the impact of specific primary

aerosol emissions and SA formation on aerosol mixing states and the influence of aerosol mixing state

parameters derived from different techniques to enhance our understanding of the mixing state of

aerosols from different emission sources and improve their characterization in models.

The North China Plain (NCP) is among the most polluted regions in China, with various primary

emission sources and strong SA formations that play critical roles in air pollution (Xu et al., 2011; Tao

et al., 2012; Liu et al., 2015). The complex mixing state of aerosols in the NCP contributes to

uncertainties in evaluating their climate and environmental effects (Zhuang et al., 2013; Nordmann et

al., 2014; Zhang et al., 2016; Tao et al., 2020; Shi et al., 2022), particularly regarding BC particles

(Wu et al., 2017; Liu et al., 2019; Zhao et al., 2019; Wang et al., 2011; Zheng et al., 2019).

Meteorological conditions can greatly affect SA formation in the NCP and can be significantly

exacerbated during severe pollution events. SA formation under low relative humidity (RH) conditions,

mainly through the condensation of gaseous-phase oxidation products, would change to that mainly

occurring in the aqueous phase under high RH conditions (Kuang et al., 2020). Because SAs formed

through different mechanisms, have different chemical compositions and add mass to different aerosol

populations, SA formation under different meteorological conditions can affect the aerosol mixing

states differently (Tao et al., 2021). This study obtained the aerosol mixing state through concurrent

measurements of the CCN activity, hygroscopicity, volatility, and BC particles at a regional site in the

NCP using CCNC, HTDMA, VTDMA, and SP2 instruments. This provides a unique opportunity to

perform a comprehensive inter-comparison of the aerosol mixing states among different techniques to

gain insight into the impact of primary aerosol emissions and SA formation on the observed aerosol

mixing states.

**2 Materials and methods**

**2.1 Campaign information and instruments setup**

From the 16th of October to the 16th of November 2021, aerosol mixing states were continuously

and concurrently monitored using different techniques at the Gucheng site in Dingxing County, Hebei

Province, China, as part of a campaign to investigate AQueous Secondary aerOsol Formation in Fog

and Aerosols and their Radiative effects in the NCP (AQ-SOFAR). The observation site, located at

39°09'N, 115°44'E, is an Ecological and Agricultural Meteorology Station of the Chinese Academy

of Meteorological Sciences, situated between the megacities of Beijing (approximately 100 km away) and Baoding (approximately 40 km away) and surrounded by farmlands and small towns. This site provides a representative view of the background atmospheric pollution conditions in the NCP (Kuang et al., 2020; Li et al., 2021).

Different measurement techniques were used to simultaneously obtain the aerosol mixing state through CCN activity, hygroscopicity, volatility, and BC particle observations. In addition to aerosol mixing state measurements, the AQ-SOFAR campaign includes measurements of aerosol number size distribution, chemical composition, aerosol scattering, and absorption properties. Aerosol number size distributions in the diameter range of 13 nm–4 µm were measured by the scanning mobility particle sizer (13–550 nm) and the aerodynamic aerosol classifier (100 nm–4 µm), and they are merged by assuming an aerosol density of 1.6 g/cm$^3$. The total BC mass concentrations were determined using an aethalometer (Magee, AE33; Drinovec et al., 2015); more information on the correction of absorption measurements and mass concentration calculations is available in Luo et al. (2022). All aerosol measurement instruments were housed in a temperature-controlled container at 24 °C. The inlet was switched among three impactors: TSP (Total Suspended Particles), $PM_{2.5}$ (Particulate Matter with an aerodynamic diameter of less than 2.5 µm), and $PM_1$ (Particulate Matter with an aerodynamic diameter of less than 1 µm). Inlet changes among impactors affect dry-state aerosol sampling owing to ambient aerosols are enlarged through aerosol hygroscopic growth or activation. However, the aerosol mixing state and aerosol chemical composition measurements were made on submicron aerosols, and the inlet change almost did not affect those measurements under conditions of RH less than 90%. The sampled aerosol was dried by two parallelly assembled Nafion dryers with a length of 1.2 m. Two Nafion driers used because of the high RH and sample flow rate (~16 L/min) during the campaign to ensure drying efficiency. The flow rate is carefully adjusted in the inlet in order to ensure accurate aerosol particle size cutoff. In addition, during autumn and winter in the NCP, ambient air temperature (<20 °C and sometimes <0 °C) can be significantly lower than the room temperature (~24 °C). Therefore, this dryer system can maintain the RH of sampled aerosols to below 20%. Meteorological data such as temperature, pressure, wind speed, wind direction, and RH were obtained from an automatic weather station operated by the station.

The chemical composition of the submicron aerosols was analyzed using a High-Resolution Time-of-Flight Aerosol Mass Spectrometer (HR-ToF-AMS). The ionization efficiency (IE) was

calibrated using 300 nm diameter pure $NH_4NO_3$ particles, following the standard protocols outlined in Jayne et al. (2000) in the middle of the campaign, with the relative ionization efficiency (RIE) of ammonium determined to be 5.26. The RIE of sulfate was 1.28 using pure $(NH_4)_2SO_4$ particles, and the default RIEs of 1.4 for organic aerosols, 1.1 for nitrates, and 1.3 for chlorides were used as the organic aerosols. The composition-dependent collection efficiency reported by Middlebrook et al. (2012) was used. Elemental ratios were derived using the "Improved-Ambient (I-A)" method as described in Canagaratna et al. (2015), including hydrogen to carbon (H/C), oxygen to carbon (O/C), and organic mass to organic carbon (OM/OC) ratios. Two primary organic aerosol (POA) and two oxygenated organic aerosol (OOA) factors were identified by High-Resolution Positive Matrix Factorization (HR-PMF; Ulbrich et al., 2009; Paatero and Tapper, 1994). This study used the summation of the two OOA factors to represent secondary organic aerosols (SOA). The mass spectra of the organic aerosol (OA) factors and their correlations with external species are shown in Figs. S1 and S2. The Biomass Burning Organic Aerosol (BBOA) spectrum was characterized by abundant fragments of $m/z$ 60 (mainly $C_2H_4O_2^+$) and 73 (mainly $C_3H_5O_2^+$), two indicators of biomass burning (Mohr et al., 2009). BBOA correlated well with $C_2H_4O_2^+$ ($R^2$=0.91) and $C_3H_5O_2^+$ ($R^2$=0.90). Consistent with previous studies in Beijing (Xu et al., 2019), the PMF analysis revealed a mixed factor named Fossil Fuel Organic Aerosol (FFOA), which comprises traffic emissions and coal combustion and is characterized by a typical hydrocarbon ion series. FFOA had a relatively high $f_{44}$ (0.083) value, which was likely due to aging during regional transportation, similar to the results observed in the winter of 2016 in Beijing (Xu et al., 2019) and coal combustion organic aerosols in Gucheng (Chen et al., 2022). Secondary organic aerosol formation from volatile organic compound precursors could occur in different formation pathways, such as aqueous-phase, heterogeneous, or gas-phase reactions. It might also be oxidized under different conditions, such as oxidation under different nitrogen oxide conditions with different oxidation capacities and oxidants. The two resolved OOA factors displayed different spectral patterns, correlations with tracers, and diurnal variations, suggesting that they resulted from different chemical processes. However, their formation mechanisms remain to be explored in future studies. In general, the OOA factor 1 (OOA1) has higher $CO_2^+/C_2H_3O^+$ (3.9) and O/C (0.91) ratios than OOA factor 2 (OOA2) with 2.1 and 0.78, respectively. The mass fraction (MF) of each chemical composition is calculated as the bulk mass fraction of each chemical composition in in non-refractory $PM_1$ (NR-$PM_1$).

This study did not consider losses in the inlet line and sampling systems for the following
reasons: (1) investigated mixing state parameters are represented by number fractions (NFs) of
different diameters, which are much less affected by losses in sampling systems compared with
absolute number concentrations; and (2) good consistency was achieved between measurements of
particle number size distributions (PNSD) and mass concentrations measured by AMS. The average
ratio between volume concentration derived from AMS and rBC measurements (densities of
compounds are the same as Kuang et al., 2021) and the volume concentration derived from PNSD
measurements was 0.79 (R=0.97, as shown in Fig. S3), consistent with previous reports as AMS cannot
detect aerosol components, such as dust (Kuang et al., 2021).

**2.2 Aerosol mixing states measurement techniques**

**2.2.1 DMA-CCNC measurements**

The CCN activity of the particles under supersaturated conditions was measured using a DMA-
CCNC system, which consisted of a differential mobility analyzer (DMA; model 3081, TSI, Inc., MN,
USA), condensation particle counter (CPC; model 3756, TSI, Inc., MN, USA), and continuous-flow
CCNC (model CCN100, Droplet Measurement Technologies, USA). The system was operated in size-
scanning mode and provided the Size-resolved Particle Activation Ratio (SPAR) by combining CPC
and CCNC measurements at different particle sizes. To compare the instruments, three
supersaturations (SSs) of 0.08%, 0.14%, and 0.22% were applied in a single cycle of approximately
15 min. CCN measurements under these three SSs revealed that the CCN activity of aerosols resides
in the accumulation mode with an aerosol diameter range of approximately 100–200 nm, which is
close to the diameters of the HV-TDMA measurements. Higher SSs would reveal CCN activities of
smaller aerosol particles (<100 nm), where the DMA-SP2 measurement is unavailable. The sample
and sheath flow rates of DMA were set at 1 and 5 lpm, respectively, resulting in a measured particle
diameter range of 9–500 nm, with a running time of 5 min per cycle. Supersaturation in the CCNC
was calibrated with monodisperse ammonium sulfate particles (Rose et al., 2008) before and after the
campaign. The flow rates were also calibrated before and after the campaign and checked daily to
minimize uncertainties in droplet counting and supersaturation formed in the column (Roberts and
Nenes, 2005; Lance et al., 2006). SPAR deviations due to multiple-charge particles were corrected

using a modified algorithm based on Hagen and Alofs (1983) and Deng et al. (2011). Further details

regarding this system can be found in Ma et al. (2016) and Tao et al. (2021).

## 2.2.2 H/V-TDMA measurements

The mixing state of the aerosols in terms of hygroscopicity and volatility was measured using a

Hygroscopicity/Volatility Tandem Differential Mobility Analyzer (H/V-TDMA; Tan et al., 2013). The

H/V-TDMA consisted of two DMA (Model 3081 L, TSI Inc.), with the first DMA (DMA1) selecting

dried particles without conditioning (RH ~15%) and the second DMA (DMA2) selecting conditioned

particles. H/V-TDMA can operate in either H- or V-mode, controlled by a three-way solenoid valve.

A Nafion humidifier was used in the H-mode to condition the selected dry particles to 90% RH

equilibrium. The number-size distribution of humidified particles ($D_p$) was measured using DMA2 and

CPC (Model 3772, TSI Inc.). The RH-dependent hygroscopic growth factor (GF) at a specific diameter

($D_d$) was calculated as follows:

$$GF=\frac{D_p(RH)}{D_d}$$ (1)

where $D_p(RH)$ is the size of particles undergoing humidification. Four dry electrical mobility diameters

(50, 100, 150, and 200 nm) were measured in this mode. The instrument was regularly calibrated using

standard polystyrene latex spheres (PSL) and ammonium sulfate particles.

In V-mode, a heated tube evaporated the volatile coatings from the previously selected dry

particles. Six temperature settings were used for the heated tube, ranging from 25–200°C. The number-

size distributions of the heated particles were measured using DMA2 and CPC. In addition to the four

particle sizes measured in the H-mode, three additional particle sizes (250, 300, and 350 nm) were

measured in the V-mode (residence time inside the heated tube to be about 1.6 s; Hong et al., 2017).

The temperature-dependent shrinkage factor (SF), which is the ratio of heated particle size to dry

particle size without heating ($D_d$), is defined as:

$$SF=\frac{D_p(T)}{D_d}$$ (2)

where $D_p(T)$ denotes the particle diameter during heating. A complete cycle of H-mode

measurements under one RH condition and V-mode measurements at six temperatures took

approximately 3 h. The Probability Density Function (PDF) of the GF (or SF) was calculated from the

measured density function using the inversion algorithm described by Stolzenburg and McMurry (2008).

### 2.2.3 DMA-SP2 measurements

The size-resolved BC mixing states were measured using an SP2 (Droplet Measurement Technology, Inc., USA) after DMA (Model 3081, TSI, USA). The DMA selected aerosols of various dry particle sizes, which were then introduced into SP2. The DMA-SP2 setup was able to measure the mixing states of aerosols with diameters (detection limit of approximately 80 nm based on the calibration) of 100, 120, 160, 200, 235, 270, 300, 335, 370, 400, 435, 470, 500, 535, 570, 600, 635, 670, and 700 nm within 20 min when it was not placed after a denuder-bypass switch system (during the following time periods: the 13th to the 24th of October, 09:00 am of the 5th of November to 09:00 am of the 8th of November). However, it only measured mixing states at diameters of 120, 160, 200, 250, 300, 400, and 500 nm when it was placed after a thermodenuder-bypass switch system (during the following time periods: 11:00 am of the 24th of October to 08:00 am of the 5th of November, and 09:00 am of the 8th of November to 06:00 pm of the 17th of November). Because the HTDMA and VTDMA measurements were conducted solely by a single H/VTDMA system operating in different modes, the time needed for a single particle size measurement of HTDMA and VTDMA was much longer than that of the DMA-SP2 system. Thus, for the same measurement cycle (2h), more particle sizes were selected in the DMA-SP2 system to acquire the BC mass concentration and mixing state at larger diameters than HTDMA and VTDMA.

The SP2 chamber had a continuous Nd:YAG laser beam with a wavelength of 1064 nm. The BC-containing particles passing through the laser beam become incandescent by absorbing radiation. The mass concentration of the BC was calculated by measuring the intensity of the emitted incandescent light. The sheath flow/sample flow ratio was maintained at 10 for the DMA to reduce the width of the diameter distribution of the selected monodisperse aerosols. Additionally, the flow rate of the SP2 was changed from 0.1 to 0.12 L/min starting on the 22nd of October (allowed flow rate range of SP2: 0.03–0.18 L/min from the specification). SP2 was calibrated using quadag soot particles, as reported by Gysel et al. (2011). Further details regarding the calibrations are provided in Section 1 of the Supplementary Information.

### 2.3 Derivations of mixing state parameters

**2.3.1 Fitting SPAR curves measured by the DMA-CCNC system**

The SPAR curves were parameterized using a sigmoidal function with three parameters. As shown in Fig. S4, a sigmoidal curve generally characterized the measured SPAR. This parameterization assumes that the aerosol is an external mixture of CCN-active hydrophilic and CCN-inactive hydrophobic particles (Rose et al., 2010). The formula used to parameterize the SPAR ($R_a(D_d)$) for a specific SS is as follows (Rose et al., 2008):

$$R_a(D_d) = \frac{MAF}{2}\left(1 + \text{erf}\left(\frac{D_d - D_a}{\sqrt{2\pi}\sigma}\right)\right) \tag{7}$$

where erf denotes the error function. The Maximum Activation Fraction (MAF) is an asymptote of the measured SPAR curve for large particles, as shown in Fig. S4, representing the number fraction of CCNs relative to the total number of particles. $D_a$ is the midpoint activation diameter, is linked to the hygroscopicity of the CCNs, and indicates the diameter where the SPAR equals half of the MAF value. The $\sigma$ is the standard deviation of the cumulative Gaussian distribution function and characterizes the heterogeneity of CCN hygroscopicity. In Fig. S4, the $\sigma$ indicates the slope of the steep increase in the SPAR curves when the diameter is close to Da. Generally, hydrophilic particles larger than $D_a$ can become CCN. Therefore, these three parameters can be used to characterize the hygroscopicity of these hydrophilic particles. This study did not consider the impact of nearly hydrophobic particles on SPAR, as deviations from this parameterization scheme due to this impact were negligible at low SSs, as stated in Tao et al. (2020).

**2.3.2 Classification of particle type based on hygroscopicity or volatility**

In this study, ambient aerosol particles were classified into two groups based on their hygroscopicity (hydrophobic and hydrophilic) and two groups based on their volatility (non-volatile and volatile) based on the measurements from H/V-TDMA (Wehner et al., 2009; Liu et al., 2011; Zhang et al., 2016). Each group can be defined using the critical values of GF or SF as follows: hydrophobic population: $GF < GF_C$; hydrophilic population: $GF \geq GF_C$; non-volatile population: $SF \geq SF_C$; and volatile population: $SF < SF_C$.

The critical values of GF ($GF_C$) and SF ($SF_C$) in H/V-TDMA depend on the particle size and working conditions, such as relative humidity and heating temperature. During this campaign, the $SF_C$

was set to 0.85 for all seven measured particle sizes at a temperature of 200 °C. The $GF_C$ for the four

measured particle sizes of 50, 100, 150, and 200 nm were 1.1, 1.15, 1.175, and 1.2, respectively, and

the corresponding hygroscopicity parameter, $\kappa$, was approximately 0.07. These values of $GF_C$ and $SF_C$

divide the probability density functions (PDFs) of SF and GF into two modes as shown in Figure 2c,

consistent with prior NCP studies (Liu et al., 2011; Zhang et al., 2016), and may be different from

those $GF_C$ and $SF_C$ in other studies because of the difference in aerosol micro-physical properties. The

NF for the hydrophilic group ($NF_H$) and volatile group ($NF_V$) can be calculated as follows:

$$NF_H = \int_{GF_C}^{\infty} GFPDF(GF) \, dGF \qquad (7)$$

$$NF_V = \int_{0}^{SF_C} SFPDF(SF) \, dSF \qquad (8)$$

where GFPDF and SFPDF are the PDFs of GF and SF, respectively, derived from H/V-TDMA

measurements.

## 311 2.3.3 Classification of particle type based on DMA-SP2 measurements

BC-containing particles can be categorized into two groups based on coating thickness: bare

BC/thinly coated BC particles and thickly coated BC particles. For the measurement of coated BC

particles at SP2, the incandescence signal is generally detected later than the scattering signals and the

time difference between the occurrence of the peaks of the incandescence and scattering signals is

defined as the lag time (Moteki & Kondo, 2007; Sedlacek et al., 2012; Subramanian et al., 2010). The

coating thickness of BC-containing particles in the SP2 measurement can be indicated by the lag time

(Moteki and Kondo, 2007; Schwarz et al., 2006; Sedlacek et al., 2012; Subramanian et al., 2010;

Metcalf et al., 2012), which has exhibited a clear two-mode distribution in previous studies (Zhang et

al., 2018; Zhao et al., 2021). A critical lag time threshold can be used to differentiate between the

different types of BC-containing particles and calculate the NF of bare and coated BC particles in the

total identified particles. In this study, a two-mode distribution of the lag time ($\Delta t$) was observed, and

a critical value of 0.8 μs was used to classify the BC-containing particles into thinly coated (or bare)

BC ($\Delta t < 0.8$ μs) and thickly coated BC ($\Delta t \geq 0.8$ μs). The definitions of all abbreviations are listed in

Table 1.

**3 Results and discussions**

**3.1 Campaign overview**

The time series of the meteorological parameters, aerosol mixing state measurements using different techniques, and mass concentrations of the aerosol chemical components are shown in Fig. 1. In detail, the measurements of aerosol mixing states include SPAR at an SS of 0.08% by DMA-CCNC, GF-PDF (PDF of GF) at 200 nm by HTDMA, SF-PDF (PDF of SF) at 200 nm and 200 °C by VTDMA, and lag time PDF of 200 nm BC-containing particles by DMA-SP2. The SIA, SOA, POA, and BC mass concentrations are shown in Fig. 1 (b). Three periods with significantly different aerosol pollution conditions were identified during the campaign. As shown in Fig. 1(b), before the 23rd of October (moderately polluted period), the accumulation of aerosols led to SIA mass concentrations <20 $\mu g/m^3$. In contrast, the highest mass concentrations of SOA, POA, and BC all reached 10 $\mu g/m^3$. The mass concentrations of different aerosol components increased significantly from the 23rd of October to the 6th of November (heavily polluted period with an average non-refractory $PM_1$ mass concentration of 49.5±22.5 $\mu g/m^3$) and decreased to much lower levels after the 6th of November (clean period with a non-refractory $PM_1$ mass concentration of 5.1±3.3 $\mu g/m^3$). Two particle groups were identified concerning the CCN activity, hygroscopicity, volatility, and coating thickness, as demonstrated by the SPAR, GF-PDF, SF-PDF, and lag-time PDF of BC-containing particles. Significant variations in the aerosol mixing states were also observed during the three periods with different pollution conditions, as demonstrated by the variations in SF-PDF measured by VTDMA. For example, the SF of the non-volatile particle group decreased during the heavily polluted period. Aerosol mixing states may have changed because of various transformations in existing aerosol particles and distinct secondary formation processes under different pollution conditions (Kuang et al., 2020; Tao et al., 2021; Shi et al., 2022; Yang et al., 2022). Diurnal variations in the mass concentrations of different aerosol chemical components and mixing states can be observed in the variations in the SPAR measurements, as previously observed in this region (Liu et al., 2011; Ma et al., 2012; Kuang et al., 2015; Tao et al., 2020).

Fig. 2 shows the campaign-averaged SPAR at the three SSs, PDF of the lag time of BC-containing particles, GFPDF, and SFPDF at 200 °C for different particle sizes. The sigmoidal SPAR curves were characterized by a rapid increase, followed by a gradual increase to unit 1, similar to the measured

SPAR curves previously observed in this region (Deng et al., 2011; Zhang et al., 2014; Ma et al., 2016; Tao et al., 2018). At lower SSs, the particle size required for CCN activation was larger; thus, rapid increases in the SPAR curves occurred at larger particle sizes as expected. In addition, the maximum AR of the SPAR curves decreases as fewer particles are CCN-active under low SSs. For the three measured SSs, the particle sizes where SPAR equals approximately 0.5 are approximately 90, 120, and 180 nm for the three SSs of 0.08%, 0.14%, and 0.22%, respectively, consistent with the average $D_a$ (see Eq. 7) values of the campaign. The NF of CCN-active particles in large-diameter ranges (which varies with SS and, for example, is greater than 200 nm for 0.08%) can be indicated by the gradual increase in the SPAR curves and quantified by the fitting parameter, MAF (see Eq. 7). The PDFs of the lag time, GF, and SF were all characterized by a bimodal distribution, which indicates two particle groups of BC-containing particles with different coating thicknesses, hygroscopicity, and volatility. The variations in the aerosol mixing states were further analyzed based on the measured mixing state parameters.

**3.2 Intercomparisons among aerosol mixing state parameters derived using four techniques**

The size-dependent characteristics of the aerosol mixing state parameters derived from the measurements of the four techniques and the MFs of different aerosol chemical components during the three pollution periods are shown in Fig. 3. In general, the size-dependent characteristics of MAF, $NF_H$, $NF_V$, and $NF_{noBC}$ were similar, suggesting that they were likely dominated by the same particle group, namely BC-free particles. This particle group had the highest number fraction ($>0.7$) during the heavily polluted period and the lowest number fraction (down to 0.5) during the clean period, with the fraction decreasing with increasing particle size. This suggests that primary emissions tend to have higher number fractions of BC-containing particles in larger diameter ranges; for example, the number fraction of BC-containing particles increases from ~0.1 to ~0.4 as the particle size increases from 200 to 500 nm during the clean period. Because the bulk aerosol MF is mostly contributed by particles $>300$ nm, there may have been more hydrophilic, volatile, CCN-active, and BC-free particles with larger sizes ($>300$ nm) during the heavily polluted period owing to strong SA formation in larger diameter ranges (Kuang et al., 2020), resulting in a higher NF of these particles compared to the clean period. As for $R_{exBC}$, which is defined as the number concentration ratio of externally mixed BC particles in total BC-containing particles, the small size dependence of $R_{exBC}$ during the moderately polluted period

might have been associated with stronger primary emissions, while the decrease in $R_{exBC}$ with increasing particle diameter in the polluted period confirmed that SA formation is more efficient for particles with larger diameters.

As for the difference among the aerosol mixing state parameters, $NF_V$ and $NF_{noBC}$ agreed with each other with a <0.1 difference, and both were higher than $NF_H$ by at least 0.1 $NF_H$ in the moderately polluted period. Compared with $NF_{noBC}$, $NF_V$ was higher during the heavily polluted period, when the nitrate mass fraction was the highest (~30%). The SOA mass fraction was the lowest (~7%) among all three periods, suggesting that some BC-containing particles in this period were also identified as volatile, consistent with the fact that the formation of semi-volatile nitrate in BC-containing particles increases their volatility. However, during the clean period, $NF_V$ was even lower than $NF_{noBC}$, suggesting that some BC-free particles were characterized as low volatile and non-negligible number fractions of BC-free particles dominated these less volatile aerosol components, which were likely less volatile organic aerosols (not likely contributed by BC-containing particles with a BC smaller than the SP2 detection limit, because this type of volatile BC-containing particles has an SF lower than 0.4 (=80 nm/200 nm), which is substantially lower than the threshold SF of 0.85 for $NF_V$ calculation). In addition, the MAF values generally agreed with the $NF_H$ during the clean period. However, they were larger than the $NF_H$ during the moderately and heavily polluted periods (by ~0.2) when the POA/SOA mass fractions were higher (~40% vs. ~35%). POA generally has a lower hygroscopicity than SOA. The critical κ of hydrophilic mode aerosols was 0.07, suggesting that a higher number fraction of aerosols had κ below 0.07 (i.e., hydrophobic mode aerosols in this study) during the moderately polluted period. However, under supersaturated conditions, they demonstrate enhanced hygroscopicity by becoming CCN-active. $NF_H$ was consistently lower than $NF_V$ and $NF_{noBC}$ (the average difference between $NF_H$ and $NF_{noBC}$ was approximately 0.2). As mentioned above, $NF_H$ was also lower than MAF during moderately polluted periods, and there may be a significant number fraction of volatile BC-free particles with hygroscopicity lower than the critical κ value of 0.07; however, they were still CCN-active and therefore not fully hydrophobic.

The diurnal variations in MAF, $NF_H$, $NF_V$, and $NF_{noBC}$, along with the MFs of the aerosol chemical components during the three periods, are shown in Fig. 4. Except for a particle size of 50 nm, the diurnal variations in these four mixing state parameters were generally similar for all measured

sizes. The different diurnal variations at a particle size of 50 nm may be due to the different effects of emissions and aging processes on the different aerosol modes, as particles <100 nm mainly reside in the Aitken mode, which is where particles >100 nm mainly reside in the accumulation mode (Wang et al., 2022). For particles >100 nm (Fig. 4 and S5), there was a maximum in the afternoon for MAF, $NF_H$, $NF_V$, and $NF_{noBC}$, indicating a peak during this time due to an increase in SA compositions, such as nitrate and SOA, and a decrease in POA and BC. Diurnal variations in the aerosol mixing state parameters and chemical compositions were more pronounced during the moderately polluted period. During heavily polluted periods, the diurnal variation was least pronounced for $NF_V$ and most pronounced for $NF_H$. In the clean-air period, there was another maximum at midnight for MAF and $NF_{noBC}$, which may be attributed to the diurnal variations in SA compositions, such as sulfate and SOA, and the decrease in BC and FFOA. The average-size dependence of the aerosol mixing state parameters over different time ranges during a heavily polluted period is shown in Fig. S6. It can be seen that the differences among the four parameters were the least from 12:00 to 18:00, with the most SOA and the least POA. This is consistent with the results shown in Fig. 3, where the difference between the MAF and $NF_H$ decreased when the POA mass fractions were the smallest. $R_{exBC}$ tended to be lower during the daytime, and its diurnal variation was more significant for larger particle sizes. In general, the diurnal variations for $R_{exBC}$ were opposite to those of $NF_{noBC}$ and agreed better with those of the primary aerosol MFs. This is because BC particles originate from primary emissions and are mainly mixed externally. After aging in the atmosphere, BC particles can be coated by SAs, resulting in more coated BC particles and fewer externally mixed BC particles. As SAs tend to form on larger particles, the diurnal variations in SA formation may significantly affect the $R_{exBC}$ of larger particle sizes.

As summarized in Table. S1, the comparison among MAF, NFH, NFV, and NFnoBC was conducted based on their correlations with different particle sizes. Note that the MAF at SSs of 0.08%, 0.14%, and 0.22% were used for comparison at 200, 150, and 100 nm particle sizes. This is because the diameter range of rapid increases in the SPAR curves is determined by aerosol hygroscopicity in this particle size range. The midpoints of the rapidly increasing diameter ranges of the SPAR curves at SSs of 0.08%, 0.14%, and 0.22% were approximately 180 nm, 120 nm, and 90 nm, respectively (as shown in Fig. 2). In general, there were moderate correlations (r=~0.5) between MAF, $NF_H$, and $NF_V$, suggesting that a similar particle group contributed to the dominance of CCN-active, hygroscopic, and

volatile aerosols (Zhang et al., 2016). The agreement between MAF and $NF_V$ was slightly higher than that between MAF and $NF_H$ or between $NF_H$ and $NF_V$. In detail, compared to the other two, the agreement between MAF and $NF_V$ has a similar correlation coefficient (r~0.65) and a smaller systematic difference (slope and intercept were much closer to 1 and 0, respectively). This is consistent with the previous finding that a substantial number fraction of volatile but less hygroscopic aerosols are CCN-active. For smaller particle sizes, the correlation became weaker (r=~0.4), whereas the degree of reduction was the lowest for the correlation between MAF and $NF_V$ (r~0.529).

## 3.3 Impacts of primary aerosol emissions on aerosol mixing states and parameter intercomparisons

Fig. 5 presents the correlation between each aerosol mixing state parameter at 200 nm and the MF of each primary organic aerosol composition during the three periods. The four mixing state parameters (MAF, $NF_H$, $NF_V$, and $NF_{noBC}$) were negatively correlated with $MF_{FFOA}$ and $MF_{BBOA}$. However, the anticorrelation with $MF_{FFOA}$ (-0.45~-0.74) was much stronger than $MF_{BBOA}$ (-0.10~-0.45). Biomass-burning emissions and fossil fuel emissions are the two major sources of BC in the NCP (Yang et al., 2022), and $NF_{noBC}$ was negatively correlated with $MF_{FFOA}$ (r=-0.49) and weakly correlated (r=-0.18) with $MF_{BBOA}$, suggesting that fossil fuel emissions were likely the dominant source of BC during this field campaign. The negative correlation between MAF and $MF_{FFOA}$ was weaker than that of $NF_{noBC}$ with $MF_{FFOA}$ (-0.62 vs. -0.49). In particular, at the same $MF_{FFOA}$, the MAF was lower than $NF_{noBC}$, demonstrating that some BC-free particles were CCN-inactive and were likely mainly composed of organic aerosols from fossil fuel combustion emissions. The negative correlation between $NF_V$ and $MF_{FFOA}$ was slightly weaker than between $NF_{noBC}$ and $MF_{FFOA}$ (-0.56 vs. -0.49). At the same $MF_{FFOA}$, $NF_{noBC}$ was close to $NF_V$, and considering that BC-containing particles were dominated by thinly coated BC most of the time (Fig. 5), this demonstrates that the non-volatile population identified by V-TDMA was mainly contributed by BC-containing particles. $NF_H$ had the lowest negative correlation with $MF_{FFOA}$ (r=-0.74), demonstrating significant contributions from fossil fuel emissions to nearly hydrophobic aerosol populations. At the same $MF_{FFOA}$, for example, when conditions of $MF_{FFOA}$ >0.1 were met, $NF_H$ (<0.7) demonstrated a noticeable decrease compared to $NF_{noBC}$ (>0.7), and $NF_H$ showed a negative correlation with both $MF_{BBOA}$ and $MF_{FFOA}$, suggesting that a substantial portion of nearly hydrophobic particles originated from FFOA- or BBOA-dominant rather

than BC-containing particles. Additionally, markedly different correlations were observed between MAF and $MF_{FFOA}$ (r=-0.62), and between MAF and $MF_{BBOA}$ (r=-0.2), implying that nearly hydrophobic but CCN-active aerosols likely originated from biomass burning. The correlations between the ratio of thinly coated BC in the total BC-containing particles ($R_{exBC}$) and the MFs of BBOA and FFOA are shown in Fig. 6. Weak correlations (r<0.3) between $R_{exBC}$ and $MF_{BBOA}$ and $MF_{FFOA}$ were observed. However, $R_{exBC}$ tended to increase with $MF_{FFOA}$, suggesting that BC-containing particles emitted from fossil fuel combustion tended to be more externally mixed with other aerosol components than those emitted from biomass burning, which is consistent with the results of previous studies (Schwarz et al., 2008; Laborde et al., 2013; Liu et al., 2017; Zhang et al., 2020). These results demonstrate remarkably different mixing states and the physical and chemical properties of fossil fuel combustion and biomass-burning aerosols.

The impact of primary emissions on the differences among the four aerosol mixing state parameters at a particle size of 200 nm was analyzed and is shown in Fig. 7. The difference between $NF_{noBC}$ and $NF_H$ ($NF_{noBC}-NF_H$) was significantly positively correlated with $MF_{FFOA}$ and $MF_{BBOA}$ (r>0.5), suggesting that a substantial proportion of POA resided in BC-free particles and was volatile, but contributed substantially to nearly hydrophobic aerosols; as did the differences between $NF_V$ and $NF_H$ ($NF_V-NF_H$). The MFs of BBOA and FFOA were poorly correlated with the differences between the MAF and $NF_V$ (MAF-$NF_V$), MAF and $NF_{noBC}$ (MAF-$NF_{noBC}$), and $NF_V$ and $NF_{noBC}$ ($NF_V-NF_{noBC}$) (Fig. S7). The difference between MAF and $NF_H$ was positively correlated with $MF_{BBOA}$, further suggesting that BBOA contributed to nearly hydrophobic aerosols under subsaturated conditions; however, their hygroscopicity was enhanced, and they became CCN-active under supersaturated conditions. The enhanced hygroscopicity of BBOA under supersaturated conditions may be attributed to: (1) surface tension lowered by surface-active organic solutes (Hodas et al., 2016; Ruehl et al., 2016); (2) liquid–liquid phase separation (Ovadnevaite et al., 2017; Liu et al., 2018); (3) dissolution of sparingly soluble compounds at higher saturated conditions (Wex et al., 2009; Dusek et al., 2011); (4) highly viscous organic aerosol which takes up water by surface water adsorption under sub-saturated conditions and by absorption of water under super-saturated conditions (Pajunoja et al., 2015). The correlations between the mixing-state parameters and primary aerosol composition during the campaign and different pollution periods are summarized in Fig. S7.

In general, both field and laboratory studies have shown that primary organic aerosols from the combustion of biomass and fossil fuels are less hygroscopic. In laboratory experiment, it is found that organic aerosols produced by fossil fuels have very low hygroscopicity, significantly less than 0.1 (Vu et al., 2015, 2017; Fofie et al., 2018; Zhang et al., 2018; Mukherjee et al., 2021). Observations have also found that the organic aerosols associated with fossil fuel combustion have low hygroscopicity, which may be due to the poorly water soluble substances in FFOA (Qiu et al., 2019; Li et al., 2021). The aerosol composition produced by biomass burning is complex, with a large number of organic aerosols (BBOA) and inorganic components being produced at the initial stage, making important contributions to CCN (Spracklen et al., 2011; Bougiatioti et al., 2016; Pöhlker et al., 2018). These primary organic aerosols (i.e. BBOA) is generally semi-volatile (May et al. 2013) and less hygroscopic (Engelhart et al., 2012; Hennigan et al., 2012), which has a negative contribution to overall hygroscopicity (Bougiatioti et al., 2016; Kuang et al., 2020b, 2021, Cai et al., 2022), resulting in weaker overall aerosol hygroscopicity in the initial stage of the biomass burning (Engelhart et al., 2012, Hennigan et al., 2012, Pöhlker et al., 2018). However, laboratory experiments found that BBOA may contain organic substances with different hygroscopicity under different saturation ratios (Malek et al., 2022), lead to increased hygroscopicity and enhanced CCN activity of BBOA under supersaturation conditions (Hersey et al., 2013). Our results generally agree with previous studies and provide evidences of the enhanced CCN activity of BBOA under supersaturation conditions in field campaigns. Furthermore, the different impacts of aerosols emitted from biomass burning and fossil fuel combustion on CCN is directly observed in this campaign used newly developed advanced aerosol-cloud sampling system, which show that biomass burning aerosols are efficient CCN even under low supersaturations (<0.05%), however, aerosols from fossil fuel combustions can only activate at higher supersaturations (~>0.14%). These results suggest simultaneous measurements of aerosol GF distributions, SPAR curves and BC mixing states and their comparisons could shed novel insights into different synergistic hygroscopic, volatile and activation properties of aerosols from different sources in the atmosphere.

**3.4 Impacts of SA formation on aerosol mixing states and parameter intercomparisons**

The correlations between the aerosol mixing state parameters at 200 nm and the MF of each SA component are presented in Fig. 8 for three periods, and the entire campaign is presented. The analysis

is conducted at only 200 nm, where all four aerosol mixing state parameters were measured to compare the four aerosol mixing state parameters and their relationships with aerosol chemical components simultaneously. Generally, MAF, $NF_H$, $NF_V$, and $NF_{noBC}$ exhibited strong positive correlations with $MF_{NH4}$ (r>0.5). This is likely because ammonium was mainly formed through neutralizing sulfuric and nitric acids with ammonia; therefore, variations in ammonium better represent overall secondary inorganic aerosol formation. As shown in Fig. 3, the secondary inorganic aerosol components dominated SA (the mass ratio between SIA and SA is approximately 70%), indicating that SA formation was primarily composed of secondary inorganic aerosol formation, which explains the weaker correlation with SOA (r=~0.3), as shown in Fig. 8.

During the clean-air period, when the MFs of SOA and sulfate were both above 15%, all four parameters had a strong positive correlation with $MF_{SO4}$ and $MF_{SOA}$ (r>0.5), suggesting that when a clean background air mass with higher mass fractions of sulfate and SOA prevailed, the local primary emissions that contributed substantially to BC-containing and less hygroscopic POA aerosols became less significant. The positive correlations between the MAF and SA components have been extensively discussed by Tao et al. (2021), who found that SA formation enhances the hygroscopicity of nearly hydrophobic aerosols, thereby increasing CCN activity. This also explains the strong correlation between the $NF_H$ or MAF and ammonium formation. The strong positive correlations between $NF_V$ and SA formation (r=~0.6) are consistent with the fact that nitrate dominates SA formation during this campaign and is semi-volatile. For the first time, strong positive correlations between $NF_{noBC}$ and SA formation were observed (r=0.6). $NF_{noBC}$ depends primarily on the relative variation between BC-containing and BC-free particles. The increase in $NF_{noBC}$ at 200 nm as a function of the SA MF suggests that SAs migrated higher mass fraction of BC-free particles with particle size smaller than 200 nm to particle size of 200 nm, suggesting that SAs tended to form more quickly on BC-free particles than on BC-containing particles with BC higher than SP2 detection limit. Recent studies reported that catalyst or photochemical reactions on BC particles can contribute the formation of secondary aerosols (Zhang et al., 2020; Zhang et al., 2021). Our results may indicate SA formation on BC particles might not be a significant pathway that contributes substantially to haze formation, and the underlying mechanisms need to be further resolved.

The effects of SA formation on the differences between the four aerosol mixing state parameters were studied and are illustrated in Fig. 9. The two OOA factors (OOA1 and OOA2) were formed through different chemical pathways. The difference between $NF_{noBC}$ and $NF_H$ ($NF_{noBC}$-$NF_H$) showed a strong negative correlation with $MF_{NH4}$ and $MF_{NO3}$ (mainly -0.6), as did the differences between $NF_V$ and $NF_H$ ($NF_V$-$NF_H$). Ammonium nitrate is a pure-scattering semi-volatile component with strong hygroscopicity, the increase of its mass fraction can enhance both aerosol volatility and hygroscopicity, therefore resulting in a smaller difference between $NF_{noBC}$, $NF_H$, and $NF_V$..

Furthermore, the difference between $NF_V$ and $NF_H$ showed a positive correlation with $MF_{OOA2}$ and a negative correlation with $MF_{OOA1}$, indicating different volatility and hygroscopicity of the two SOA factors. The differences between $NF_V$ and $NF_H$ concerning the MF of OOA1 and OOA2 are shown in Fig. 9(e) and (f), respectively. As previously noted, $NF_V$ was generally higher than $NF_H$, and the difference between the two decreased with increasing $MF_{OOA1,}$ which was generally smaller than 0.3. This suggests that the formation of OOA1 enhances the hygroscopicity of volatile particles, which aligns with the highest oxidation state of OOA1 (higher O/C but lower H/C compared to OOA2) and has a significant and overall positive impact on aerosol hygroscopicity (Cerully et al., 2015; Thalman et al., 2017; Zhang et al., 2023). A positive correlation was observed between $NF_V$ and $MF_{OOA2}$ (r=~0.25).

In contrast, the correlation between $NF_H$ and $MF_{OOA2}$ was weak (R was close to 0), implying that OOA2 might be semi-volatile but only weakly hygroscopic, which could contribute to $NF_V$ being higher than $NF_H$ as OOA1 increases. The difference between $NF_{noBC}$ and $NF_V$ ($NF_{noBC}$-$NF_V$) was negatively correlated with $MF_{NO3}$, which is consistent with the semi-volatile nature of nitrate. The negative correlation between $NF_{noBC}$-$NF_V$ and $MF_{OOA2}$ indicates that the difference is smaller when there is more OOA2, implying that OOA2 is also a semi-volatile compound and is likely formed mainly on BC-free particles (particles with BC mass lower than detection limit are not excluded). The correlations between the differences between $NF_V$-MAF and $NF_{noBC}$-MAF and the MF of each SA composition were very weak. The impacts of SA formation on BC mixing states are shown in Fig. S8. In general, the NF of thinly coated BC has a negative correlation with SIA and a weak association with SOA, suggesting that SIA formation mainly enhances the thickness of the BC coating. The correlations between the mixing state parameters and SA composition during the campaign and different pollution

periods are summarized in Fig. S9. Our results on OOA agree with previous studies, that OOA are reported to be volatile (Kim et al., 2020; Cai et al., 2022) but can have a positive or negative impact on hygroscopicity depending on its oxidation level (Kim et al., 2020; Kuang et al., 2021; Cai et al., 2022).

In addition to changes in the MFs of SA compositions, the accumulation of SA pollution may provide insights into the impact of SA formation on aerosol mixing states. As shown in Fig. 10(a), during the heavily polluted periods, there were two distinct pollution accumulation processes from the 23rd to the 27th of October and from the 28th to the 31st of October, respectively. During the pollution accumulation process, the mass concentration of SAs increased by approximately three-fold, indicating the rapid formation of secondary compositions and a significant increase in NR-PM$_1$ mass concentration. Fig. 10(b) and (c) illustrate that this increase in SAs significantly enlarged the value of aerosol mixing state parameters, including MAF, NF$_V$, NF$_H$, and NF$_{noBC}$, which increased from approximately 0.5 to 0.8 with evident diurnal variations. This highlights the impact of SA formation on the aerosol mixing states and the importance of studying the pollution accumulation processes of SAs. The enhancements in the different aerosol mixing state parameters during the pollution accumulation process were not uniform. MAF and NF$_H$ initially exhibited lower values than NF$_V$ and NF$_{noBC}$; however, their later enhancement was stronger than that of NF$_{noBC}$. Fig. 10(d) and (e) show the difference between NF$_{noBC}$ and NF$_V$ at 200 and 300 nm as a function of SA mass concentrations during these two pollution periods, which clearly shows how, during SA formation, NF$_V$ became higher than NF$_{noBC}$ while NF$_V$ remained close to the NF of thickly coated BC-containing particles (NF$_{CBC}$) plus NF$_{noBC}$ (NF$_{CBC}$+NF$_{CBC}$). These results suggest that SA formation increases the volatility of BC-free and BC-containing particles, leading to an increased NF$_V$ compared with NF$_{noBC}$. Almost all BC-free particles and some BC-containing particles become volatile during the accumulation of pollution.

## 4. Conclusions

The aerosol mixing state is one of the most important physicochemical properties of aerosol particles and significantly affects their optical properties and the CCN activity of aerosol particles. The aerosol mixing states vary significantly with complex aerosol emissions and atmospheric

transformations. In this study, aerosol mixing states derived from CCN activity, hygroscopicity, volatility, and BC particle observations, along with their relationship to primary aerosol emissions and SA formation, were systematically analyzed based on simultaneous measurements of CCNC, H/VTDMA, and SP2. Statistical analysis demonstrated that the NFs of CCN-active, hygroscopic, and volatile particles were generally positively correlated and mainly contributed by BC-free particles. Therefore, four mixing state parameters were all negatively correlated to either the MFs of BBOA or FFOA because fossil fuel combustion and biomass burning were the two major sources of BC-containing particles during this field campaign. However, the differences between these mixing state parameters varied significantly under different conditions, have shed new insights into aerosol physical and chemical properties and even secondary aerosol formation mechanisms.

Fossil fuel combustion and biomass burning emissions represent two major primary sources of global aerosol burden and are dominant primary aerosol sources in this campaign. It is known that the chemical compositions of both these primary sources are dominated by organics and BC. However, the intercomparison results among instruments revealed significant differences in the physical and chemical properties of aerosols emitted from these two sources. The combination of HTDMA, DMA-SP2, as well as aerosol source apportionment confirmed that substantial portions of BC-free aerosols from both biomass burning and fossil fuel combustion are nearly hydrophobic under sub-saturated conditions. Additionally, BC from fossil fuel combustion tends to be more externally mixed with other aerosol compositions than those from biomass burning. However, additional insights from DMA-CCN measurements revealed that substantial portions of BC-free aerosols, nearly hydrophobic from biomass burning, could serve as CCN, while a substantial portion of those from fossil fuel combustion could not. Previous studies have confirmed the hygroscopicity difference of aerosols from biomass burning under sub- and supersaturated conditions in laboratory settings; however, such differences have rarely been confirmed in field measurements. Moreover, comparisons between sub- and supersaturated conditions for aerosols from fossil fuel combustion have been rarely undertaken, even in laboratory studies. This finding is quite important because the ability of primary organic aerosols from biomass burning and fossil fuel combustion is often treated as the same in models (Liu et al., 2021; Pöhlker et al., 2023).

Secondary aerosol formation substantially alters aerosol mixing state. The different variations in mixing state parameters can also help reveal mechanisms of secondary aerosol formation. For example, the two resolved SOA factors exhibited different impacts on the differences between $NF_V$ and $NF_H$ ($NF_V$-$NF_H$), and their correlations with $NF_V$ and $NF_H$ revealed that $OOA_1$ was more hygroscopic but less volatile, suggesting distinct formation mechanisms for these two OOA factors. Further analysis might help link SOA formation mechanisms to aerosol physical properties, which is important for connecting aerosol chemistry to aerosol climate effects determined by aerosol physicochemical properties. Additionally, variations in size-resolved $NF_{noBC}$ revealed that secondary organic and inorganic aerosol formations led to the migration of BC-free particles towards larger diameters more quickly than that of BC-containing particles. This phenomenon is more likely to occur when aqueous pathways dominate secondary aerosol formation because BC-containing particles generally exhibit weak hygroscopicity and do not favor aqueous processes.

The findings of this study highlight the markedly different effects of primary emissions and SA formation on aerosol mixing states and suggest that comparisons of aerosol mixing states obtained using various techniques are useful for gaining insights into the hygroscopicity, volatility, and CCN activity of different aerosols. Recommendations are listed for future studies based on the findings of this study: (1) When exploring the impact of aerosol emissions and secondary aerosol formations on aerosol hygroscopic under sub- and supersaturated conditions, we recommend employing simultaneous DMA-SP2 measurements to better represent BC characteristics; (2) Simultaneous DMA-CCNC, V-HTDMA, and DMA-SP2 measurements could enhance studies on secondary aerosol formation mechanisms. Conversely, if formation mechanisms and pathways are clear, these measurements could elucidate how secondary aerosol formation impacts aerosol physical properties from different aspects. (3) To be cautious in the application of aerosol mixing state parameters from HV-TDMA to conduct aerosol optical property investigations because the suitability of HV-TDMA-derived mixing state parameters for representing BC mixing states is largely dependent on the composition and mass of the secondary aerosols, and DMA-SP2 measurements are recommended for this purpose.

**Data availability**. The data used in this study are available from the corresponding author upon request Ye Kuang (kuangye@jnu.edu.cn) and Li Liu (liul@gd121.cn)

**Competing interests**. The authors declare that they have no conflict of interest.

**Author Contributions**.

YK and WY planned this campaign and YK designed the aerosol experiments and conceived this research together with JC, and JC wrote the manuscript. JC performed measurements of CCNC, BL performed measurements of SP2 and analyzed SP2 datasets with the help of GZ, WQ and YL performed AMS measurements, LL performed HV-TDMA measurements and conducted post-data processing as well as some of data analysis. BX, HX, MMZ, HZ and SR participated this campaign and helped instruments maintenance. GZ provided full support for the campaign. All authors contributed to discussions and revisions of this paper.

**Financial supports**. This work is supported by National Natural Science Foundation of China (42175083, 42175127, 42275066), the Guangzhou Science and Information Technology Bureau Project (2023A04J0941), the Guangdong Provincial Key Research and Development Program (grant no. 2020B1111360003), the Science and Technology Innovation Team Plan of Guangdong Meteorological Bureau (grant no. GRMCTD202003).

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

**Table 1. Definition and description of abbreviations.**

| Abbreviation | Full name and/or Definition |
|---|---|
| BBOA | Biomass Burning Organic Aerosol<br>Characterized by obvious m/z 60 (mainly $C_2H_4O_2^+$) and 73 (mainly $C_3H_5O_2^+$), which are two indicators of biomass burning |
| $D_a$ | Midpoint activation diameter<br>Linked to the hygroscopicity of CCNs |
| $D_d$ | Particle diameter under dry conditions without humidification or heating |
| $D_p$ | Particle diameter after humidification or heating |
| GF | Growth factor<br>The ratio between particles with and without humidification and is linked to aerosol hygroscopicity |
| $\kappa$ | Hygroscopicity parameter |
| MF | Mass Fraction |
| MAF | Maximum Activation Fraction<br>An asymptote of the measured SPAR curve at large particle sizes and represents the number fraction of CCNs to total particles |
| $NF_H$ | Number Fraction of Hydrophilic aerosol whose hygroscopicity parameter is $>\sim0.07$ at particle size of 50, 100, 150 and 200 nm |
| $NF_V$ | Number Fraction of Volatile aerosol whose Shrinkage Factor at 200 °C is $<0.85$ at particle size of 50, 100, 150 and 200 nm |
| $NF_{noBC}$ | Number Fraction of black carbon (BC)-free particles at particle size of 200, 250, 300 and 370 nm |
| $NF_{CBC}$ | Number Fraction of thickly coated BC particles at particle size of 200, 250, 300 and 370 nm |
| $NF_A$-$NF_B$<br>($NF_{noBC}$-$NF_H$, $NF_V$-$NF_H$, $NF_{noBC}$-$NF_V$, $NF_V$-MAF, | The difference between the number fraction of A and B at particle size of 200 nm |

NF$_{noBC}$-MAF)

| | |
|---|---|
| OOA1 and OOA2 | Two OOA factors resolved from the PMF analysis |
| PDF | Probability Distribution Function |
| PM$_{2.5}$ | Particulate Matter with an aerodynamic diameter <2.5 μm |
| PM$_1$ | Particulate Matter with an aerodynamic diameter <1 μm |
| POA | Primary Organic Aerosol<br>Summation of BBOA and FFOA |
| R$_{exBC}$ | The number concentration ratio of externally mixed BC particles in total BC-containing particles<br>Externally mixed BC particles are defined as identified bare/thinly coated BC-containing particles at particle size of 200, 250, 300 and 370 nm |
| SA | Secondary Aerosols, including nitrate, sulfate, ammonium and the two OOA factors |
| SF | Shrinkage Factor<br>The ratio between particles with and without heating and is linked to aerosol volatility |
| SIA | Secondary Inorganic Aerosols, including nitrate, sulfate, and ammonium |
| SOA | Secondary Organic Aerosols, including the two OOA factors |
| SPAR | Size-resolved Particle Activation Ratio<br>Size-dependent CCN activity under a specific SS |

1174

1175

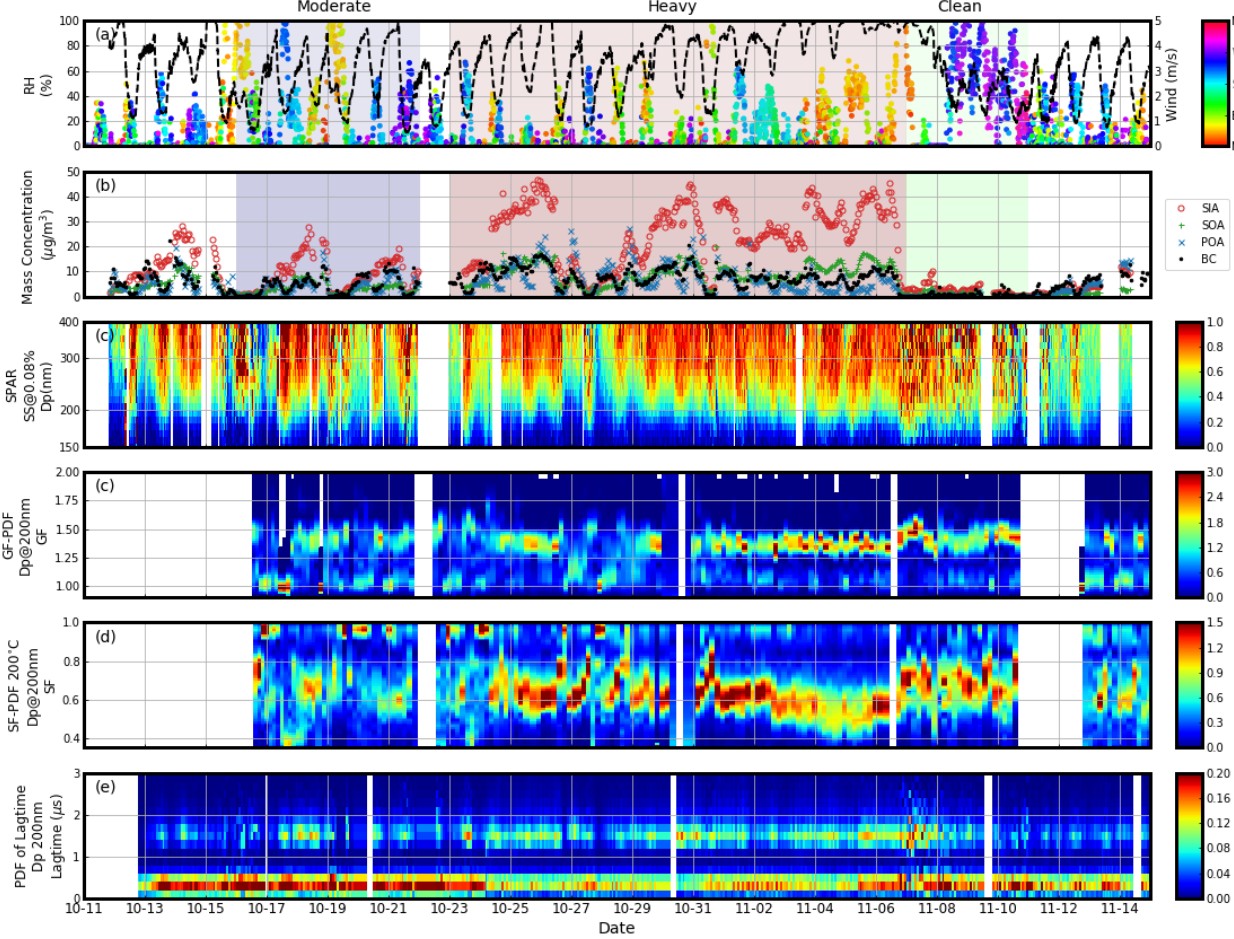

1176

**Figure 1**. Overview of the measurements during the campaign: **(a)** meteorological parameters: wind speed (dots) and relative humidity (RH) (black line), with colors of dots representing wind direction; **(b)** mass concentrations of aerosol chemical components: secondary inorganic aerosols (SIA, red circle), secondary organic aerosols (SOA, green plus), primary organic aerosols (POA, blue x) and black carbon (BC, black dots); **(c)** Size-resolved Particle Activation Ratio (SPAR) under supersaturation (SS) of 0.08% observed by the DMA-CCNC, with warmer colors corresponding to higher values; **(d)** Probability Density Function (PDF) of growth factor (GF-PDF) at 200 nm observed by the HTDMA; **(e)** PDF of shrinkage factor (SF-PDF) at 200 nm and 200 °C observed by the VTDMA; **(f)** PDF of lag time at 200 nm observed by the DMA-SP2. The blue, red, and green shaded periods represent the three periods with moderate pollution, heavy pollution, and clean conditions, respectively.

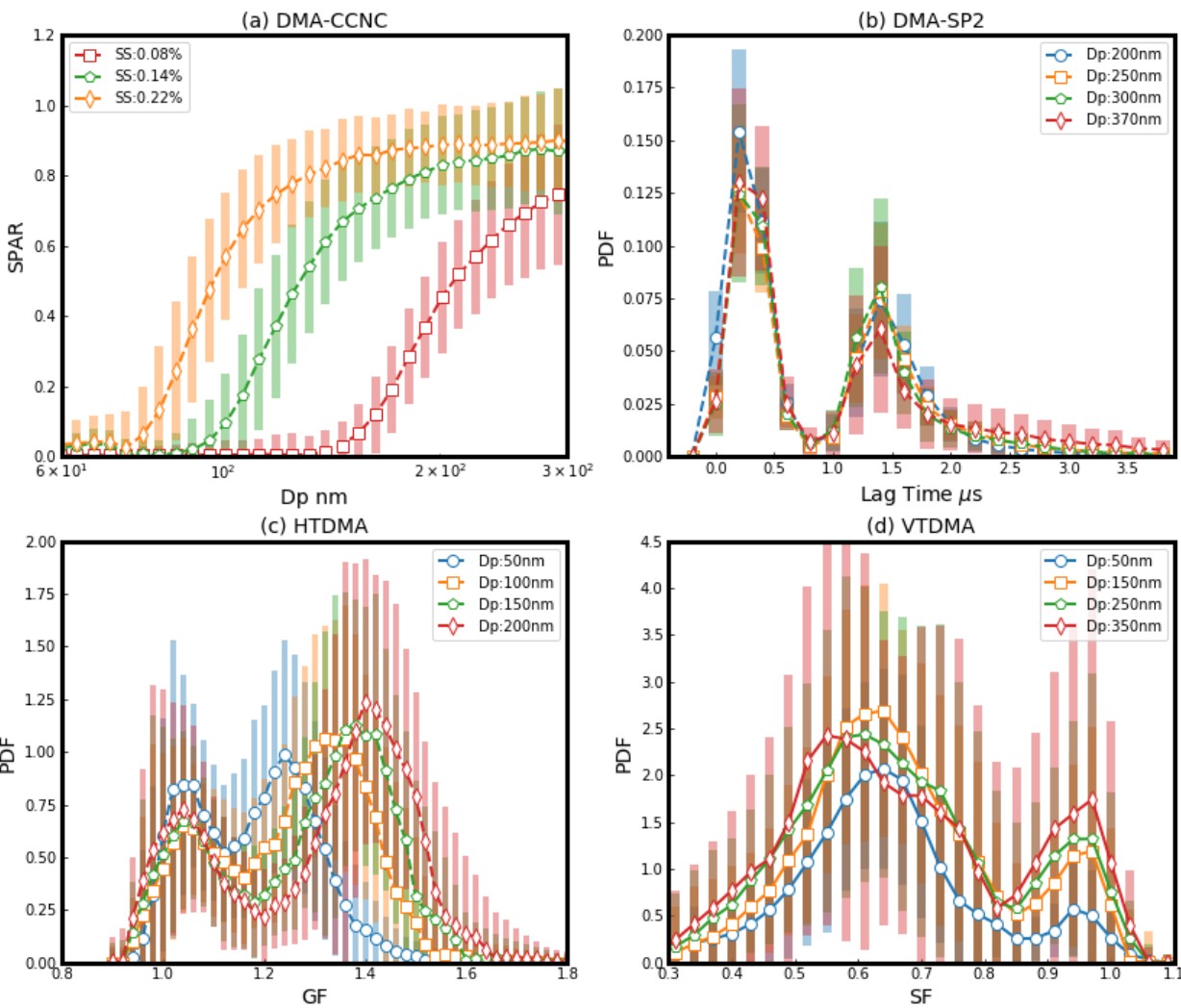

1186

**Figure 2**. The campaign average of **(a)** Size-resolved Particle Activation Ratio (SPAR) curves measured by DMA-CCNC at the three supersaturations (SSs, represented by different colors and markers), **(b)** Probability Density Function (PDF) of lag time measured by DMA-SP2 at four particle sizes (represented by different colors and markers), **(c)** PDF of growth factor (GF) measured by HTDMA at four particle sizes (represented by different colors and markers), **(d)** PDF of shrinkage factor (SF) measured by VTDMA under the temperature of 200 °C at five particle sizes (represented by different colors and markers). The shaded areas indicate the standard deviations.

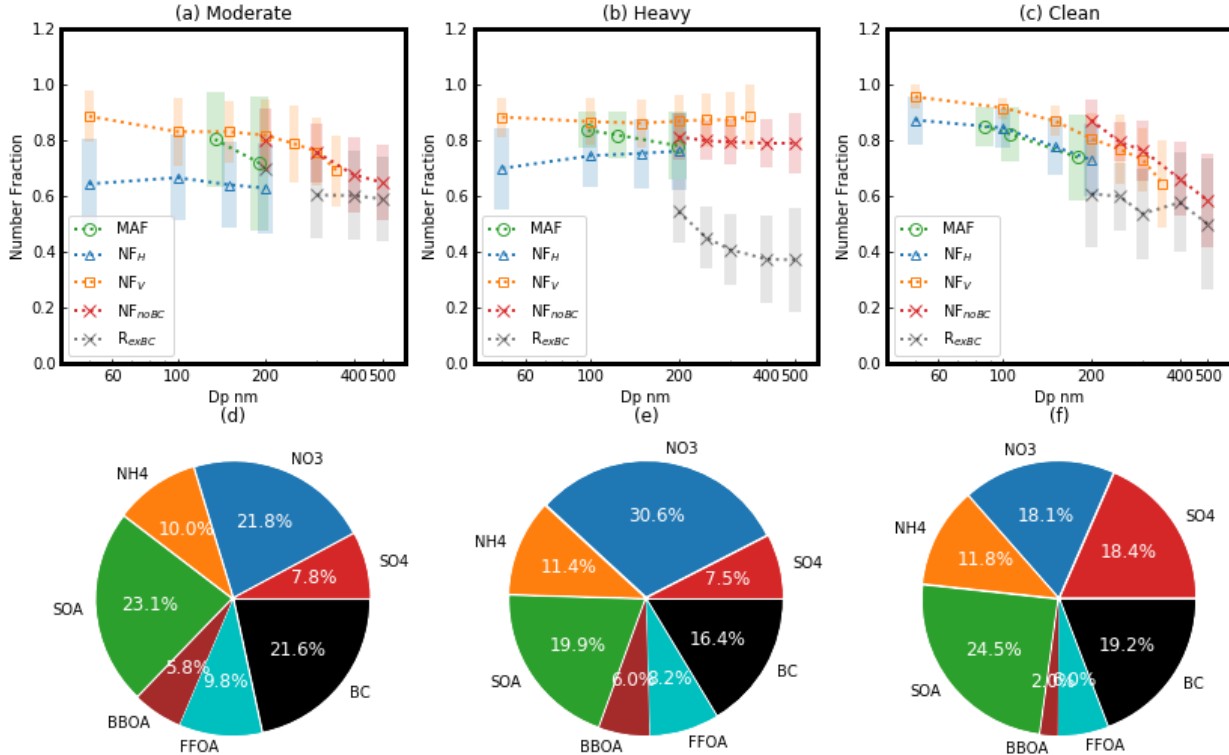

1193

**Figure 3**. **(a–c)** Size dependence of MAF (green circle), $NF_H$ (blue triangle), $NF_V$ (yellow square), $NF_{noBC}$ (red x), and $R_{exBC}$ (black x) during the three periods. **MAF**: Maximum Activation Fraction, an asymptote of the measured Size-resolved Particle Activation Ratio (SPAR) curve at large particle size. **$NF_H$**: Number Fraction of Hydrophilic aerosol whose hygroscopicity parameter is higher than ~0.07. **$NF_V$**: Number Fraction of Volatile aerosol whose Shrink Factor at 200 °C is lower than 0.85. **$NF_{noBC}$**: Number Fraction of black carbon (BC)-free particles. **$R_{exBC}$**: Number fraction of externally mixed BC particles in total BC-containing particles. **(d–f)** Corresponding mass fractions (MFs) of aerosol chemical components (identified by colors) during the three periods, including secondary organic aerosols (SOA), biomass burning organic aerosol (BBOA), fossil fuel organic aerosols (FFOA), and inorganic ions including sulfate ($SO_4$), nitrate ($NO_3$), and ammonium ($NH_4$).

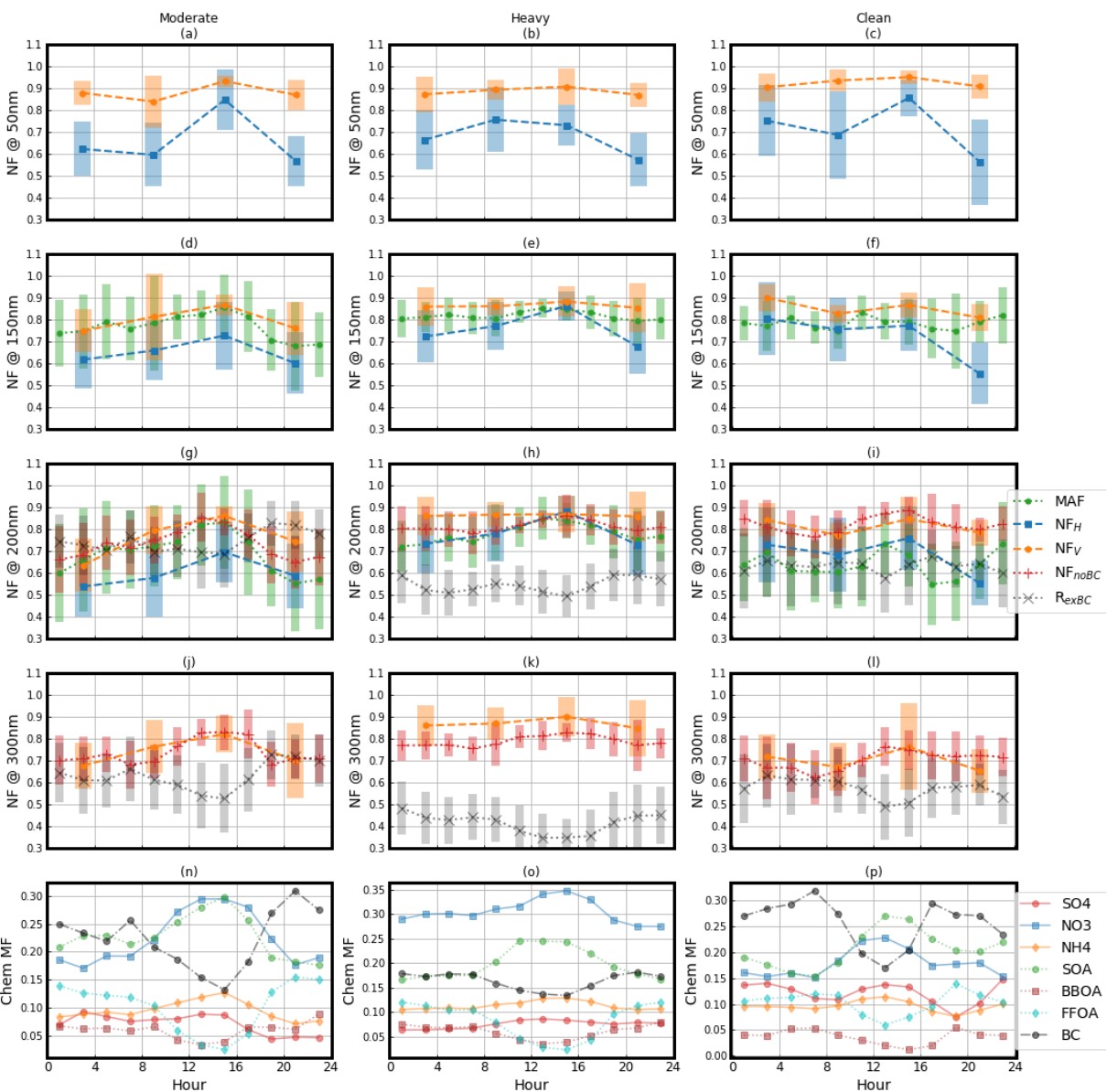

**Figure 4. (a–l)** Diurnal variations of aerosol mixing state parameters (identified by color and marker) at different particle sizes (50, 150, 200, and 300 nm) during the three periods. The shaded areas indicate the standard deviations. **(m–o)** Diurnal variations of mass fractions (MFs) of aerosol chemical components, including secondary organic aerosols (SOA), biomass burning organic aerosol (BBOA), fossil fuel organic aerosols (FFOA), and inorganic ions including sulfate ($SO_4$), nitrate ($NO_3$), and ammonium ($NH_4$) (identified by color and marker) during the three periods.

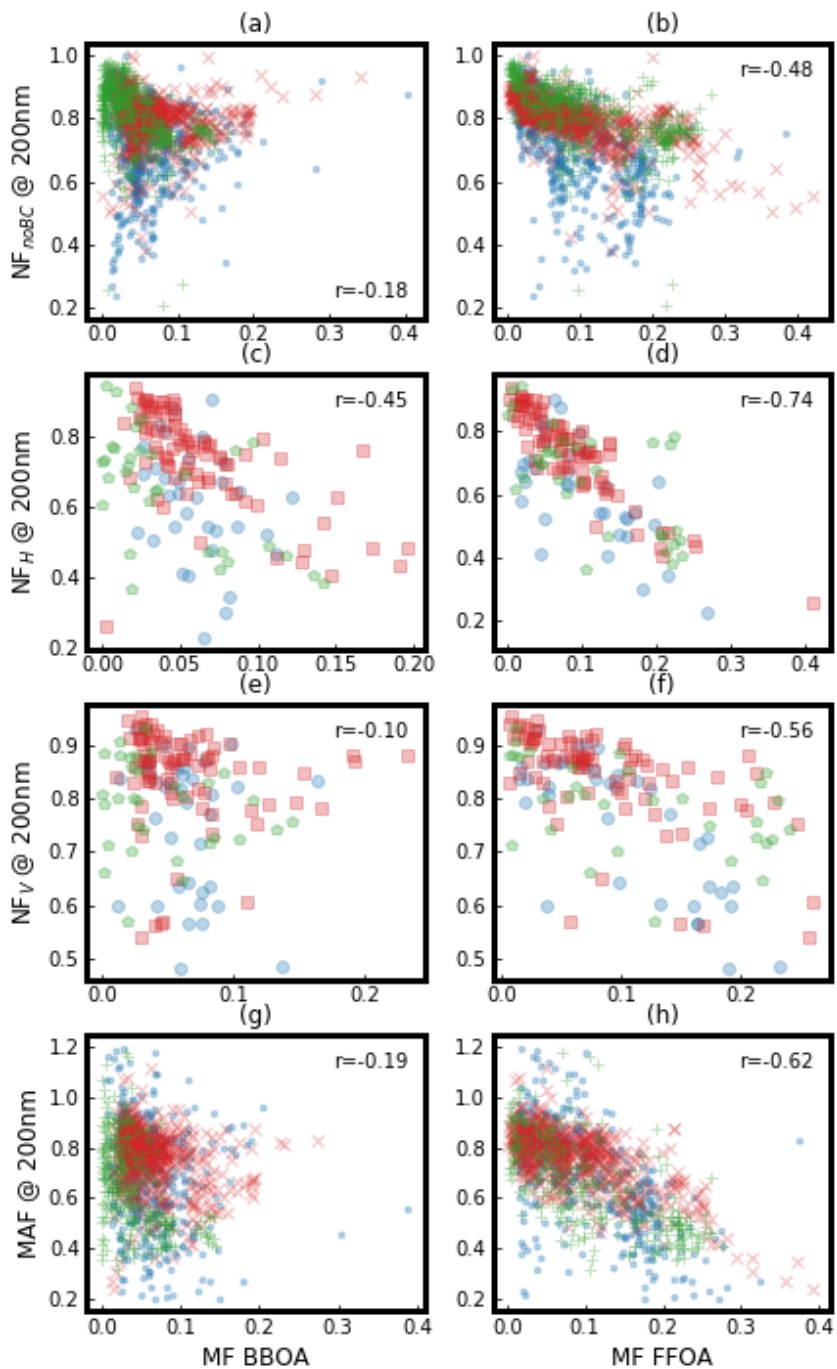

**Figure 5**. The correlations between aerosol mixing state parameters and mass fractions (MFs) of biomass burning organic aerosol (BBOA) and fossil fuel organic aerosols (FFOA) during different periods (moderately polluted period: blue dot or circle; heavily polluted period: red x or square; clean period: green plus or pentagon), with r representing the correlation coefficient.

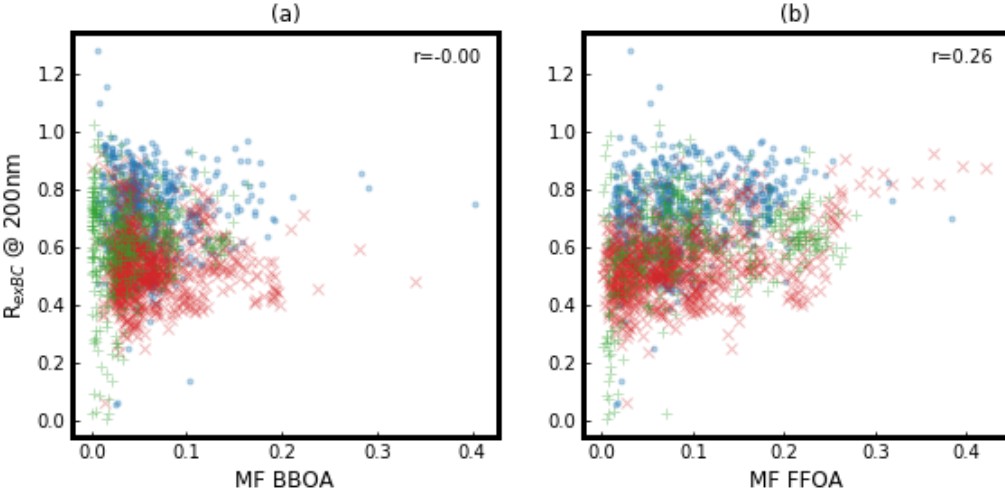

**Figure 6**. The correlations between the ratio of externally mixed black carbon (BC) in total BC particles ($R_{exBC}$) and mass fractions (MFs) of biomass-burning organic aerosol (**BBOA**) and fossil fuel organic aerosols (**FFOA**) during different periods (moderately polluted period: blue dot; heavily polluted period: red x; clean period: green plus), with r representing correlation coefficient.

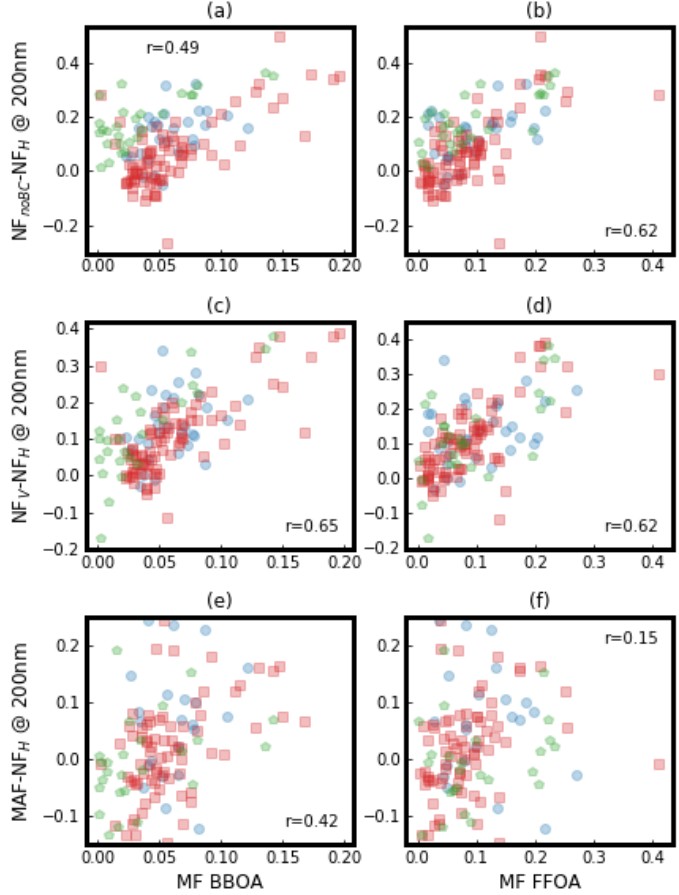

1220

**Figure 7**. The correlations between the difference among the four aerosol mixing state parameters at particle size 200 nm and mass fractions (MFs) of biomass burning organic aerosol (**BBOA**) and fossil fuel organic aerosols (**FFOA**) during different periods (moderately polluted period: blue circle; heavily polluted period: red square; clean period: green pentagon), with r representing correlation coefficient.

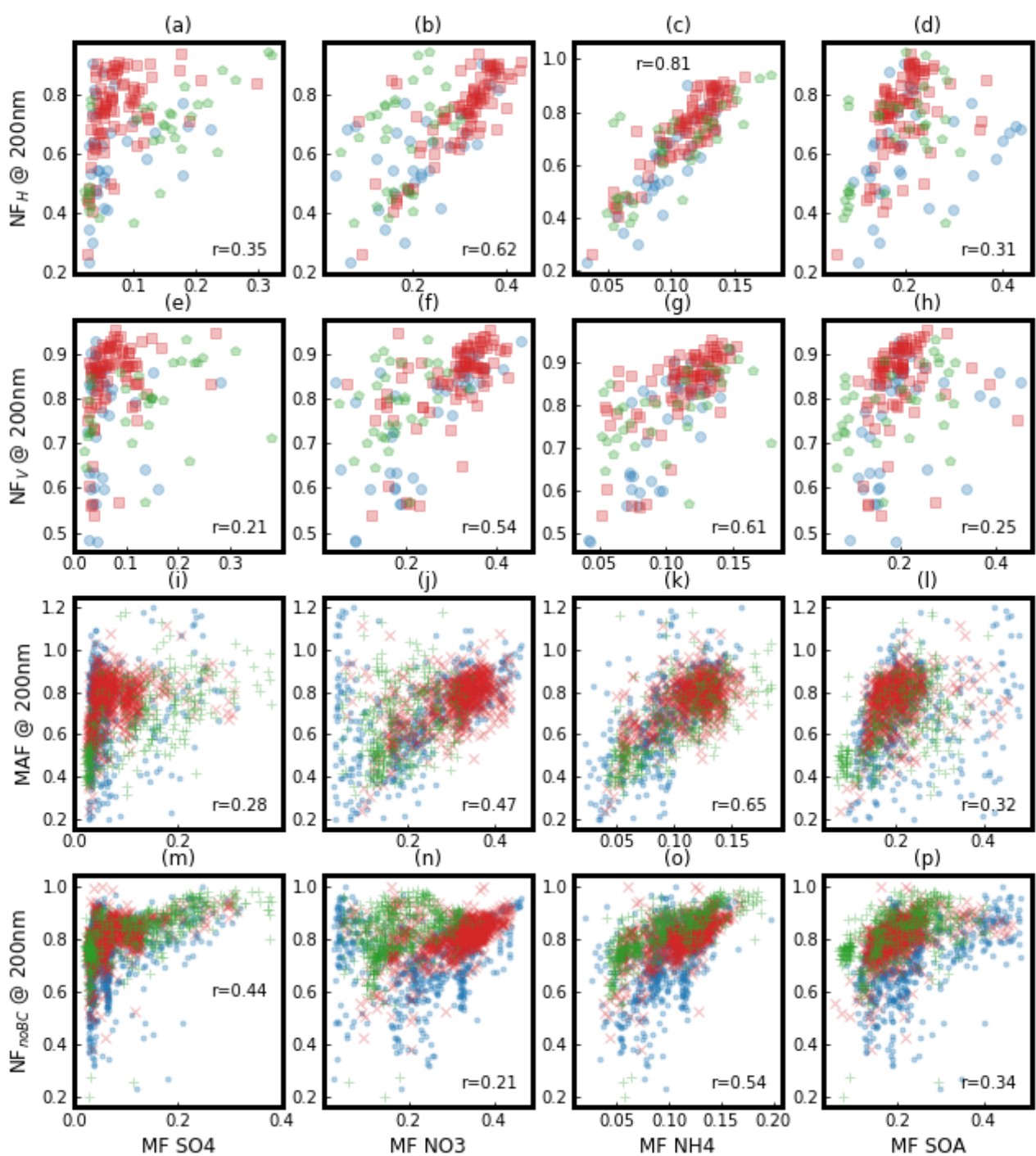

**Figure 8**. The correlation between the four aerosol mixing state parameters and mass fraction (MF) of secondary aerosol (SA) components during different periods (moderately polluted period: blue dot or circle; heavily polluted period: red x or square; clean period: green plus or pentagon), with r representing correlation coefficient. SA components include secondary organic aerosols (SOA), sulfate ($SO_4$), nitrate ($NO_3$), and ammonium ($NH_4$)

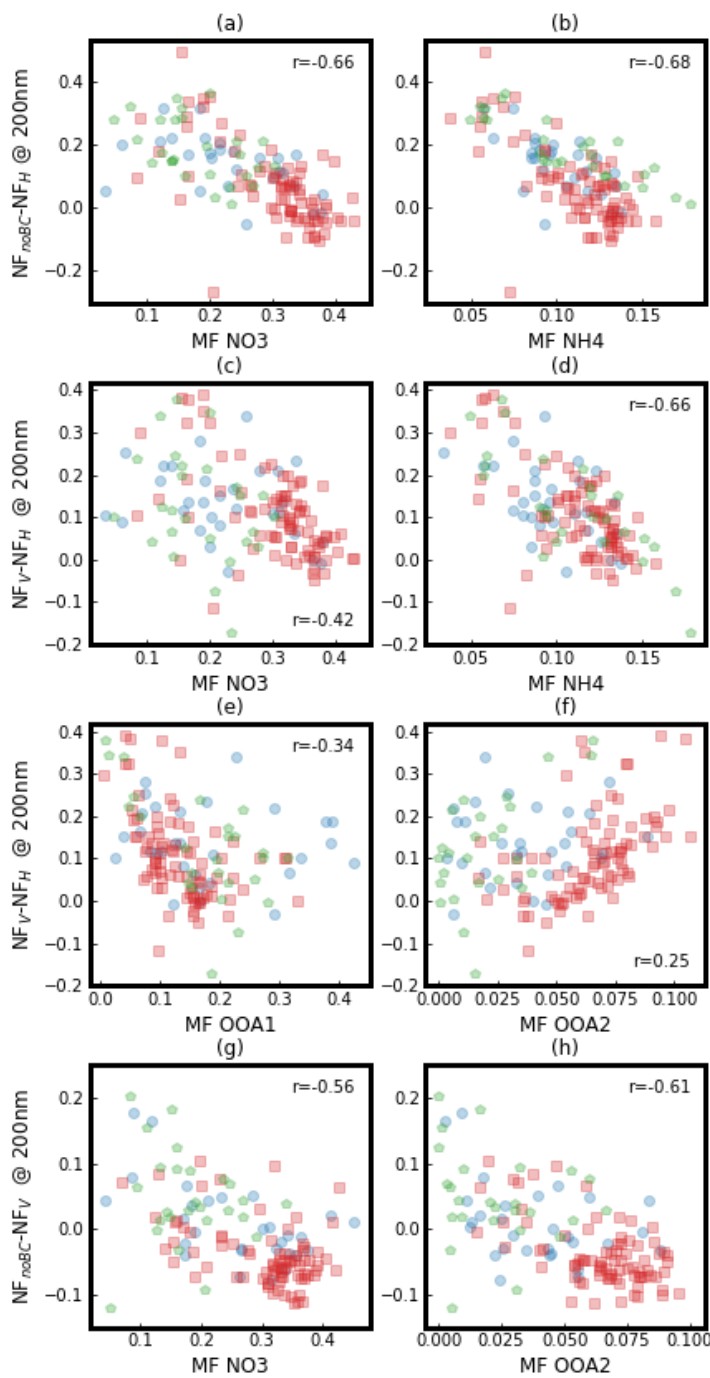

1230

**Figure 9**. The correlation between the difference among the four aerosol mixing state parameters and mass fraction (MF) of secondary aerosol (SA) chemical components during different periods. OOA1 and OOA2 are two secondary organic aerosol (SOA) factors resolved from aerosol mass spectrometer (AMS) measurements using the Positive Matrix Factorization (PMF) technique. Moderately polluted period: blue circle; heavily polluted period: red square; clean period: green pentagon.

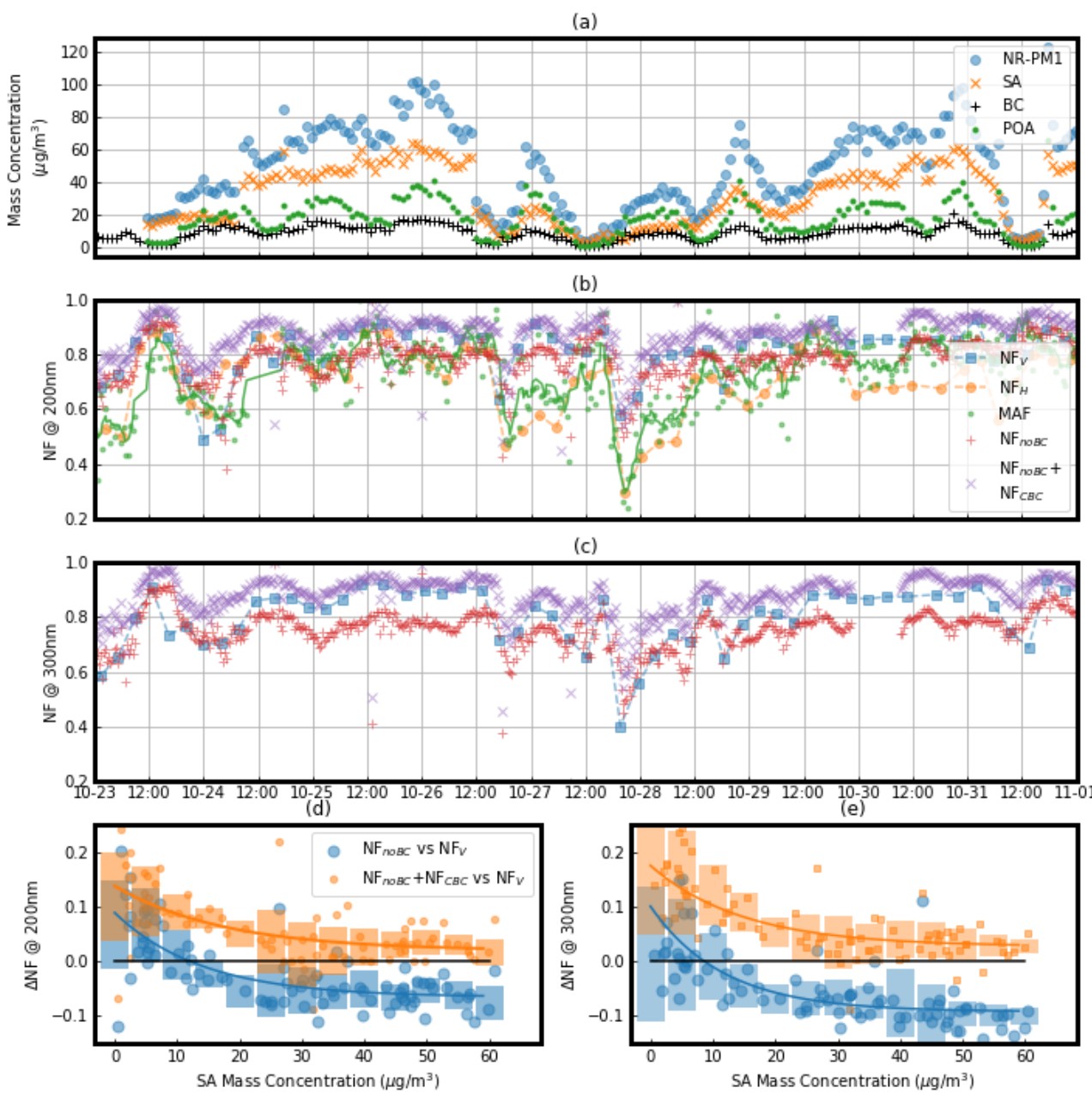

**Figure 10**. Variations of different aerosol mixing state parameters during the pollution accumulation process. **(a)** The time series of mass concentrations of non-refractory PM$_1$ (NR-PM$_1$), secondary aerosols (SAs) (including inorganic ions and secondary organic aerosols (SOA)), primary organic aerosols (POA) and black carbon (BC) (identified by colors and markers). **(b and c)** The variations of different aerosol mixing state parameters (identified by colors and markers) at particle size 200 nm **(b)** and 300 nm **(c)**. **(d and e)** The variations of the difference between NF$_V$ and NF$_{noBC}$ (NF$_V$-NF$_{noBC}$, blue large circle) and the difference between NF$_V$ and NF$_{noBC}$+NF$_{CBC}$ (NF$_V$-(NF$_{noBC}$+NF$_{CBC}$), yellow small circle) with the mass concentration of SA at particle size 200 nm **(d)** and 300 nm **(e)** **NF$_{CBC}$:** Number Fraction of thickly coated black carbon (BC) particles.