# Peer review of "Markedly different impacts of primary emissions and secondary"

_Atmospheric Chemistry and Physics, 2022_

## Referee Comment (RC1)

Review on "Markedly different impacts of primary emissions and secondary aerosols formations on aerosol mixing states revealed by simultaneous measurements of CCNC, V/HTDMA and SP2" by Jiangchuan Tao et al.

General comments:

The presented study investigates the mixing state of aerosol particles using different techniques: H- and V-TDMA, CCNC, and SP2 measurements are available in connection with chemical measurements for a 1-month campaign in the North China plain (NCP). This combination provides a useful data set to investigate the aerosol mixing state. However, this combination of measurements gives a lot of information and in this study many parameters were calculated. To understand the relationships and differences between these parameters, they need to be explained and presented in more detail. I believe, the data itself are worth to be published, but the quality of analysis and publication should be improved. The authors use too many abbreviations that disrupt the flow of reading. Some abbreviations are not explained at all, but I think even those that are well known in a particular community should be written out at least once. Furthermore, the statistical analysis is not convincing. Linear correlations are applied to all data points, but in my view, they do not well describe the data in all cases. A critical analysis is needed here to determine which of these statistical results are meaningful. This is my main criticism of this work.

The quality of the language is not very good and the manuscript is not easy to read. I recommend a complete check by a native speaker.

Thus, the paper needs major revision regarding the statistical analysis. After that, the text should be partly rewritten or at least significantly revised before it can be accepted for publication in ACP.

**Comments in detail:**

There are basic criticisms of the manuscript, so I will go into less detail. Most of my comments are more general, only few of them focus on typos and so on, which does not mean, that these are all minor comments. But I would focus on the detail after the rest is done.

Examples for abbreviations, that are never written out:

SOA, POA, SSOA, BBOA, CCOA, FFOA, MAF..

Some of them are well known, others not. I do not know all of them which makes the reading really difficult. Each abbreviation has to be explained once, but I would suggest to use less abbreviations in general. Even if abbreviations are explained in the technical section and used later without explanations does not really help. I prefer written text, it helps a lot to understand the text much better. In my view it is required to explain those abbreviations, which are not widely known, such as MAF, CCOA, regularly again, also in figure captions.

Section 2.2:

Some more technical details about the aerosol measurements would be helpful. What type of inlet was used? Was the measurement flow dried? How was the relative humidity in the inlet flow?

Were losses in inlet line and sampling systems considered?

$D_d$ is probably the dry diameter?! This is not explained. What means 'dry'? Just not humidified?

The same diameters $D_d$ and $D_P$ are used in the definition for the shrinking factor, what is the meaning here?

Section 2.3:

Parameterization of the SPAR function is not easy to understand without knowing how it looks like. Can the authors give an example?

People, who are not familiar with the SP2 do not understand the explanation given here. What does the lag time mean? Why is it called lag time? By the way, there are three different ways of writing in the manuscript: lagtime, lag-time and lag time, for consistency one should be chosen. I would take the latter one.

Section 3:

Figure 1 is very complex. Figures c – e also have a color scale on the right hand side, but this is not explained at all.

Line 316 ff: '…corresponding fitting parameters, Da…' Da is just one parameter and means probably the mean diameter? How are these diameters obtained? Da should probably be $D_a$?! Other fitting parameters are needed?

MAF seems to be another fitting parameter, but what does MAF mean?

There appear again lots of abbreviations, such as RexBC. This is explained once, but since it is not common, I had to look it up again and again. I would prefer reading without so many abbreviations.

Line 371, caption figure 4 and others: the word 'composition' is used in the wrong context. The authors mean probably component(s). This appears several times in the text.

Figure 4: what are the shaded areas? Standard deviations? Uncertainty? This has to be explained in the figure caption! My question is, if the differences e.g., between the different diameters are significant? For me, the shapes of the curves of NF for different diameters look very similar, in particular if the shaded area represents an uncertainty range.

In the description of this figure 4 the word 'peak' is frequently used, but I see only slight maxima between different times of the day. This has to be checked and needs to be adapted.

Line 388: What means 'consistency' here? I simply do not understand it.

Figures 5 – 9: linear correlations were fitted here, but the results do not always look convincing. E.g., Figure 7: 2 lowest plots show dots widely distributed and one does not expect a linear correlation. What is the meaning of such a correlation? I strongly suggest to check the quality of these correlations and reduce to number of these plots.

Figure 8: the lower plots seem to follow more an exponential growth, does the linear fit makes sense here?

Figure 9: what means OOA1 and OOA2?

All figure captions need more text to explain the figure. One should understand the general content of a figure without reading the full text around.

Line 472: exemplarily 'difference between $NF_V - NF_H$' means difference between $NF_V$ and $NF_H$? This appears several times around this section.

Minor comments/ typos:

Comment: I did not look explicitly for all typos, because I think, several parts need to be rewritten und after that it should be read again carefully.

Line 293: PA means probably POA

Line 319: in large diameters ranges

Line 334:a dot after 'size' is missing

Line 439: 'are presented' should be 'is presented'

---

## Referee Comment (RC2)

Comment on acp-2022-840

General comments:

The present paper focuses on improving our understanding of the aerosol mixing state in a background site of the North China Plain in China. This is achieved by combining various techniques, including HTDMA, CCN counter, VTDMA, and SP2. The study provides a first-time intercomparison of the four aerosol mixing state parameters from the instruments above and offers insights into the interlink among these parameters and potential sources. I find this research to be important and interesting for aerosol mixing state studies. The manuscript is well-structured and scientifically engaging for the aerosol society. However, in terms of writing, it would be beneficial for non-expert readers if certain sentences were shortened and explanations were provided before reaching conclusions. Please see the detailed comments below. I suggest publishing the manuscript after a minor revision.

Specific comments:

1)  Line 70. "..lead to substantial overestimation". Could you provide more details about the magnitude of the substantial overestimation?
2)  Line 94. "highly correlated to those of a VTDMA at high temperature". Which temperature do you refer to and why?
3)  Line 127-129. Please summarize the key messages of the meteorology influences on the aerosol mixing state.
4)  Line 168 BBOA, line 173 FFOA.., please explain the full name when introducing a new term and check out the remaining of manuscript.
5)  Line 177, what do you mean by "different chemical process" and could you give more details?
6)  Line 187, why do you choose these three supersaturations for CCN measurements?
7)  Line 211, the maximum temperature you chose is 200 degree Celsius, why do you choose this threshold?
8)  Line 225-229, regarding the chosen size for SP2, which system was conducted for this study, with or without thermodenuder-bypass? Since you are expected to compare with HTDMA and VTDMA, why not choose the same sizes to measure for the three instruments?
9)  Line 235, does the flow rate influence the measurements and by how much?
10) Section 2.3.1, the MAF is a fitting parameter from eq.7, what is the physical meaning of this parameter? Is it the maximum activation fraction?
11) Line 267, add sizes for the GF "The GFC for the four measured particle sizes were 1.1, 1.15, 1.175 and 1.2".
12) Section 2.3.3. Here you use the lag time between the peak of the scattering signal and the incandescence signal to classify the bare and coated BC. Is it related to the BC-coating mass ratio? The mass ratio is more commonly used and intuitive to understand.
13) Line 297-299, please give exact values of PM mass for the heavily polluted and clean period.
14) Line 315-316. "At lower SSs, the rapid increases in SPAR curves occur at larger particle sizes and the maximum AR of SPAR curves becomes smaller". Please briefly explain why.

15) Line 318, add SS for the "increases in SPAR curves, are approximately 90 nm, 120 nm and 180 nm"

16) Fig 2. Are bars representing the standard deviation of the campaign average?

17) Line 331-333, "In general, the size dependence of MAF, NFH, NFV and NFnoBC were similar to one another, suggesting they were dominated by the same particle group, namely BC-free particles". I think this statement is not well supported, I would suggest weakening it or proving it with more evidence. For example, thickly coated BC particles can be very CCN-activate, hydrophilic and volatile, if mostly contain SIA.

18) Line 335, please provide exact values of the fraction of BC-containing particles and the applied diameter range, because the terms "higher" or "larger" are not accurate. Check out similar issues for the remaining manuscript too.

19) Line 342, what do you mean by "the more efficient secondary aerosol formation", increase by secondary aerosol mass or particle size?

20) Line 356-357, what is the kappa value for hydrophobic mode aerosol?

21) Line 361, how do you get this statement with "lower than 0.07 but still CCN active"? please explain in detail.

22) Fig4, I would suggest simplifying the plot and keeping the sizes with most concurrent measurements, e.g. 150, 200 and 300 nm. Put other sizes to the supplement. Line 362-366, the diurnal variations should be described more explicitly as the pattern of RexBC is clearly different from the other three mixing state parameters and explain why.

23) Line 384, table S1 is quite interesting for readers thus I suggest putting it or making a correlation plot into the main context.

24) Line 385, why do you choose these three sizes? The critical size for the setting SS?

25) Line 386. A classification of the correlation should be clarified, such as the r range for the weak, moderate, and strong correlation.

26) Line 392, what do you mean by saying "..while the degree was the least for the correlation.."?

27) Fig. 5, what is the r in the plot? It would be more intuitive to use the same marker to represent different periods in the plot.

28) A summary table (or correlation matrix plot) of r in Fig5-7 will be helpful for readers to better understand the interlink between mixing state parameters and chemical composition.

29) Line 400, please give values to the sentence "correlation with MFFFOA was much weaker compared to MFBBOA".

30) Fig.7. Which size of data do you use?

31) Line 428, please introduce what the difference ($NF_{noBC}$-$NF_H$ and $NF_V$-$NF_H$) represents first before jumping to the results.

32) Line 438, why do you choose 200nm?

33) Line 459-462, out of curiosity, does the transport of ageing aerosols play a role in the increasing fraction of non-BC particles?

---

## Author Comment (AC1)

Reviewer #1:

*General comments:*

*The presented study investigates the mixing state of aerosol particles using different techniques: H- and V-TDMA, CCNC, and SP2 measurements are available in connection with chemical measurements for a 1-month campaign in the North China plain (NCP). This combination provides a useful data set to investigate the aerosol mixing state. However, this combination of measurements gives a lot of information and in this study many parameters were calculated. To understand the relationships and differences between these parameters, they need to be explained and presented in more detail. I believe, the data itself are worth to be published, but the quality of analysis and publication should be improved. The authors use too many abbreviations that disrupt the flow of reading. Some abbreviations are not explained at all, but I think even those that are well known in a particular community should be written out at least once. Furthermore, the statistical analysis is not convincing. Linear correlations are applied to all data points, but in my view, they do not well describe the data in all cases. A critical analysis is needed here to determine which of these statistical results are meaningful. This is my main criticism of this work.*

*The quality of the language is not very good and the manuscript is not easy to read. I recommend a complete check by a native speaker.*

*Thus, the paper needs major revision regarding the statistical analysis. After that, the text should be partly rewritten or at least significantly revised before it can be accepted for publication in ACP.*

**Response: Thanks for your comments. This study provides a first-time intercomparison of aerosol mixing state parameters from the instruments including DMA-SP2, DMA-CCN, HTDMA and VTDMA and offers insights into the interlink among these parameters and potential influencing factors. Aerosol mixing states were usually investigated using one or two of instruments listed above, however, none of them could deliver a full picture of aerosol mixing state variations. The purpose of this paper is to investigate what's the difference among these mixing state parameters and mechanism behind those difference under atmospheric conditions of the campaign, which helps aerosol scientists to understand better aerosol mixing states obtained using different techniques and also design better their future aerosol experiments, because differences among mixing state parameters might deliver important message about physical and chemical properties of primary and secondary aerosols as discussed in this study. Some observed differences can be qualitatively explained based on existing knowledge; however, some differences help us explore possible properties of primary and secondary aerosols and might deliver phenomena that urge explanation in the future. If we go very detail into the variations of each aerosol mixing**

state parameters, the manuscript would be very long and more difficult for readers because it was very difficult to find readers who understand very well all instruments listed above (DMA-SP2, DMA-CCN, HTDMA, VTDMA) and aerosol primary emissions as well as atmospheric chemistry related with secondary aerosol formations. The authors struggled in writing this manuscript because this is also difficult for us, although the first author has very good records of research using DMA-CCN and HTDMA, and the corresponding author have good records of aerosol physical properties and atmospheric chemistry. We decided to write this paper because we find this might be important and interesting for aerosol community, and also helpful for us and we want to share these insights. Actually, we plan to dig more into these variations based on insights obtained in this research in the our future studies.

We agree with the reviewer that some places should be explained more in detail, and therefore more explanations were added before reaching conclusions in some parts as recommended by the reviewer#2.

In terms of statistical analysis, we use linear correlations to examine whether the primary emissions or the secondary aerosol formations have significant impacts on the aerosol mixing state parameters, rather than getting linear relationships. Linear fittings in the manuscript delivered false impression thus all linear fittings are removed in related figures and leave only correlation coefficients. We discussed this with authors and believe that there are no explicit relations among these parameters, thus correlation test is the only way we could have now based on our limited measurements to explore potential influencing factors as what was done in most previous papers that discuss possible mechanisms behind variations in mixing state parameters.

In terms of writing, the reviwer#2 have raised a lot of comments to help improve the readability, and we revised the manuscript based on comments of both reviewers which is beneficial for non-expert readers.

*Comments in detail:*

*There are basic criticisms of the manuscript, so I will go into less detail. Most of my comments are more general, only few of them focus on typos and so on, which does not mean, that these are all minor comments. But I would focus on the detail after the rest is done.*

*Examples for abbreviations, that are never written out:*

*SOA, POA, SSOA, BBOA, CCOA, FFOA, MAF...*

*Some of them are well known, others not. I do not know all of them which makes the reading really difficult. Each abbreviation has to be explained once, but I would suggest to use less abbreviations in general. Even if abbreviations are explained in the technical section and*

*used later without explanations does not really help. I prefer written text, it helps a lot to understand the text much better. In my view it is required to explain those abbreviations, which are not widely known, such as MAF, CCOA, regularly again, also in figure captions.*

**Response: Thanks for your suggestion. We have added a table listing the definition and description of the abbreviations as follow:**

**Table 1. Definition and description of the abbreviations.**

| Abbreviations | Full name and/or Definition |
|---|---|
| BBOA | Biomass Burning Organic Aerosol

In this study, characterized by obvious m/z 60 (mainly $C_2H_4O_2^+$) and 73 (mainly $C_3H_5O_2^+$), which are two indicators of biomass burning |
| FFOA | Fossil Fuel Organic Aerosol

A mixed factor in this study that comprises traffic emissions and coal combustions, which was characterized by typical hydrocarbon ion series |
| OOA | Oxygenated Organic Aerosol |
| OOA1 and OOA2 | Two OOA factors resolved from the PMF analysis |
| SOA | Secondary Organic Aerosols

Summation of OOA1 and OOA2 |
| POA | Primary Organic Aerosols

Summation of BBOA and FFOA |
| $PM_{2.5}$ | Particulate Matter with aerodynamic diameter less than 2.5 $\mu m$ |
| $PM_1$ | Particulate Matter with aerodynamic diameter less than 1 $\mu m$ |
| $NR\text{-}PM_1$ | Non-refractory $PM_1$ |
| MF | Mass Fraction |

| | |
|---|---|
| $D_p$ | Particle diameter after humidification or heating |
| $D_d$ | Particle diameter under dry conditions without humidification or heating |
| κ | Hygroscopicity parameter |
| SS | Super-saturation |
| SPAR | Size-resolved Particle Activation Ratio

Size-dependent CCN activity under a specific SS |
| MAF | Maximum Activation Fraction

An asymptote of the measured SPAR curve at large particle sizes and represents the number fraction of CCNs to total particles |
| $D_a$ | Midpoint activation diameter

Linked to the hygroscopicity of CCNs |
| GF | Growth factor

The ratio between particle with and without humidification, and is linked to aerosol hygroscopicity |
| SF | Shrinkage Factor

The ratio between particle with and without heating, and is linked to aerosol volatility |
| PDF | Probability Distribution Function |
| $NF_H$ | Number Fraction of Hydrophilic aerosol whose hygroscopicity parameter is higher than ~0.07. |
| $NF_V$ | Number Fraction of Volatile aerosol whose Shrink Factor at 200 °C is lower than 0.85. |
| $NF_{noBC}$ | Number Fraction of BC-free particles |
| $NF_{CBC}$ | Number Fraction of thickly coated BC particles |

| | |
|---|---|
| $R_{exBC}$ | Number concentration ratio of externally BC particles in total BC-containing particles.

Externally BC particles are defined as identified bare/thinly coated BC-containing particles. |
| $NF_A$-$NF_B$

($NF_{noBC}$-$NF_H$,$NF_V$-$NF_H$, $NF_{noBC}$-$NF_V$,$NF_V$-MAF, $NF_{noBC}$-MAF) | The difference between the number fraction of A and B. |

**In addition, we have also added the definition and description of the abbreviations in the caption of the figures for clarification and improving readability.**

*Section 2.2:*

*Some more technical details about the aerosol measurements would be helpful. What type of inlet was used? Was the measurement flow dried? How was the relative humidity in the inlet flow?*

**Response: Thanks for your suggestion. We have added technical details about the aerosol measurements in Section 2.1 as follow:**

**"The inlet was switched among three impactors: TSP (Total Suspended Particles), PM2.5 (Particulate matter with aerodynamic diameter less than 2.5 μm) and PM1 (Particulate matter with aerodynamic diameter less than 1 μm). Inlet changes would affect the dry state aerosol sampling due to aerosol hygroscopic growth or activation. However, the aerosol mixing state and aerosol chemical composition measurements were made on submicron aerosols, inlet change almost does not affect those measurements under conditions of RH less than 90%, and this would be discussed very carefully in our next paper. The sampled aerosol was dried by two parallelly assembled Nafion dryers with length of 1.2 m. During autumn and winter in the NCP, ambient air temperature (lower than 20 °C and can down to 0 °C) can be significantly lower than the room temperature (~24 °C), this dryer can maintain the RH of sampled aerosols to below 20%."**

*Were losses in inlet line and sampling systems considered?*

**Response: losses in inlet line and sampling systems are not considered in this study. reasons are listed below: (1) investigated mixing state parameters are represented by**

number fractions of different diameters which are much less affected by losses in sampling systems compared with absolute umber concentrations; (2) good consistency was achieved between measurements of particle number size distributions (PNSD) by particle sizer and mass concentrations measured by AMS, with the average ratio between volume concentration derived from AMS and rBC measurements and volume concentration derived from PNSD measurements is 0.79 (R=0.97, as shown in the following), which is consistent with previous reports due to that AMS cannot detect aerosol components such as dust (Kuang et al., 2021). This means that almost same aerosol populations were sampled by AMS and instruments of measuring aerosol mixing states.

[Figure]

**Fig. S3. Comparison between aerosol volume concentration derived from measurements of PNSD and aerosol chemical compositions.**

The following sentences are added in the revised manuscript.

"Note that losses in inlet line and sampling systems are not considered in this study. reasons are listed below: (1) investigated mixing state parameters are represented by number fractions of different diameters which are much less affected by losses in sampling systems compared with absolute umber concentrations; (2) good consistency was achieved between measurements of particle number size distributions (PNSD) by particle sizer and mass concentrations measured by AMS, with the average ratio between volume concentration derived from AMS and rBC measurements and volume concentration derived from PNSD measurements is 0.79 (R=0.97, as shown in Fig.S3), which is consistent with previous reports due to that AMS cannot detect aerosol components such as dust (Kuang et al., 2021). This means that almost same aerosol populations were sampled by AMS and instruments of

**measuring aerosol mixing states."**

*Dd is probably the dry diameter?! This is not explained. What means 'dry'? Just not humidified?*

**Response: Yes, Dd is the dry diameter particle, which corresponds to particle diameter under dry conditions (RH<20%) and not humidified. For clarification, we have revised the description of Equation (1) as:**

**"… The H/V-TDMA consists of two DMA (Model 3081L, TSI Inc.), with the first DMA (DMA1) selecting dried particles without conditioning and the second DMA (DMA2) selecting conditioned particles. … The RH-dependent hygroscopic growth factor (GF) at a certain dry diameter ($D_d$) is calculated as follows:**

$$GF=\frac{D_p(RH)}{D_d} \qquad (1)$$

**Where Dp(RH) is the particle diameter undergo humidification. In this mode, four dry electrical mobility diameters (50, 100, 150 and 200 nm) were measured. …"**

*The same diameters Dd and DP are used in the definition for the shrinking factor, what is the meaning here?*

**Response: Thanks for your suggestion. In the definition for the shrinking factor, Dd is the dry diameter particle, which corresponds to particle diameter under dry conditions and not heated while Dp is the particle diameter after heating. we have revised the description of Equation (2) as:**

**"The temperature dependent shrinkage factor (SF), which is the ratio of the heated particle size to the dry particle size without heating (Dd), is defined as:**

$$SF=\frac{D_p(T)}{D_d} \qquad (2)$$

**Where Dp(T) is the particle diameter undergo heating. …"**

*Section 2.3:*

*Parameterization of the SPAR function is not easy to understand without knowing how it looks like. Can the authors give an example?*

**Response: Thanks for your suggestion. As shown in Figure S4, the measured SPAR is generally characterized as a sigmoidal curve (the black line). MAF is the asymptote of**

the measured SPAR curve at large particle sizes and Da indicates the diameter where SPAR equals the half of the MAF value. The parameter σ corresponds to the slope of steep increase of SPAR curves when diameter is close to Da.

[Figure]

**Fig. S4. Schematic of the parameterization scheme of SPAR curves. The black solid curve and the black crossing are the measured SPAR and fitted SPAR with the parameterization scheme. The red, green and blue dashed lines indicate the fitting parameters of Maximum Activation Fraction (MAF), the midpoint activation diameter (Da) and s, respectively.**

We have added this figure into the supplement and revised the description of SPAR parameterization scheme as:

"The measured SPAR curves can be parameterized with a sigmoidal function with three parameters. As shown in Fig. S4, the measured SPAR is generally characterized as a sigmoidal curve. This parameterization assumes that the aerosol is an external mixture of hydrophilic particles that are CCN-active and hydrophobic particles that are CCN-inactive (Rose et al., 2010). The formula used to parameterize SPAR ($R_a(D_d)$) for a specific SS is as follows (Rose et al., 2008):

$$R_a(D_d) = \frac{MAF}{2}\left(1 + erf\left(\frac{D_d - D_a}{\sqrt{2\pi}\sigma}\right)\right) \qquad (7)$$

where erf is the error function. MAF (Maximum Activation Fraction) is an asymptote of the measured SPAR curve at large particle sizes as shown in Fig. S4, and it represents the number fraction of CCNs to total particles. Da is the midpoint activation diameter and is linked to the hygroscopicity of CCNs, and indicates the diameter where SPAR

equals the half of the MAF value. σ is the standard deviation of the cumulative Gaussian distribution function and characterizes the heterogeneity of CCN hygroscopicity. In Fig. S4, the parameter corresponds to the slope of steep increase of SPAR curves when diameter is close to Da. …"

*People, who are not familiar with the SP2 do not understand the explanation given here. What does the lag time mean? Why is it called lag time? By the way, there are three different ways of writing in the manuscript: lagtime, lag-time and lag time, for consistency one should be chosen. I would take the latter one.*

**Response: Thanks for your suggestion. The lag time is defined as the time difference between the occurrences of the peaks of the incandescence and scattering signals measured by SP2 (Moteki & Kondo, 2007; Sedlacek et al., 2012; Subramanian et al., 2010), and for coated BC particles, the incandescence signals is generally detected later than the scattering signals. As shown in former studies (Zhang et al., 2018; Zhao et al., 2021), the distribution of the lag time for ambient particles exhibits a clear two-mode distribution and this lag time can be used to indicate the coating thickness of the BC-containing aerosols.**

**We have revised lagtime and lag-time to lag time, and have revised this paragraph as:**

"… For measurement of coated BC particles in SP2, the incandescence signals is generally detected later than the scattering signals and the time difference between the occurrences of the peaks of the incandescence and scattering signals is defined as the lag time (Moteki & Kondo, 2007; Sedlacek et al., 2012; Subramanian et al., 2010). The coating thickness of the BC-containing aerosols in the SP2 measurement can be indicated by the lag time (Moteki and Kondo, 2007; Schwarz et al., 2006; Sedlacek et al., 2012; Subramanian et al., 2010; Metcalf et al., 2012), which exhibits a clear two-mode distribution in former studies (Zhang et al., 2018; Zhao et al., 2021). A critical lag time threshold can be used to differentiate between the different types of BC-containing aerosols and calculate the number fraction of bare BC particles and coated BC particles in the total identified aerosols. …"

"

*Section 3:*

*Figure 1 is very complex. Figures c – e also have a color scale on the right hand side, but this is not explained at all.*

**Response: Thanks for your suggestion. We have revised the caption of Figure 1 by**

**adding descriptions of each panel and the color scale as:**

"**Figure 1. Overview of the measurements during the campaign: (a) meteorological parameters: wind speed (dots) and RH (black line), with colors of dots representing wind direction; (b) mass concentrations of aerosol chemical compositions: secondary inorganic aerosols (SIA, red circle), secondary organic aerosols (SOA, green plus), primary organic aerosols (POA, blue x) and black carbon (BC, black dots); (c) Size-resolved Particle Activation Ratio (SPAR) under supersaturation (SS) of 0.08% observed by DMA-CCN, with warmer colors corresponding to higher value; (d) PDF of growth factor (GFPDF) at 200 nm observed by HTDMA; (e) PDF of shrinkage factor (SFPDF) at 200 nm and 200 °C observed by VTDMA; (f) PDF of lag time at 200 nm observed by DMA-SP2. The blue, red and green shaded periods represent the three periods with moderate pollution, heavy pollution and clean condition, respectively.**"

*Line 316 ff: '...corresponding fitting parameters, Da...' Da is just one parameter and means probably the mean diameter? How are these diameters obtained? Da should probably be Da?! Other fitting parameters are needed?*

**Response: Da is the midpoint activation diameter, not the mean diameter. Da is not shown in Figure 2 and may be mistaken as particle size Dp. Here we are referring to Da values during the campaign. It can be found that Da value agree with the particle size where SPAR equals about 0.5. We have revised this sentence as:**

"**For the three measured SSs, the particle size where SPAR equals about 0.5 are approximately 90 nm, 120 nm and 180 nm for the three SSs of 0.08%, 0.14% and 0.22%, respectively. These particle size agree with the value of the fitting parameter Da (midpoint activation diameter, see Eq.7) during the campaign, as the fitting parameter MAF (Maximum Activation Fraction, an asymptote of the measured SPAR curve at large particle sizes) is close to 1.**"

*MAF seems to be another fitting parameter, but what does MAF mean?*

**Response: MAF is Maximum Activation Fraction and an asymptote of the measured SPAR curve at large particle sizes. We have added the description of MAF where MAF is first mentioned in this section as:**

"**These particle size agree with the value of the fitting parameter Da (midpoint activation diameter, see Eq.7) during the campaign, as the fitting parameter MAF (Maximum Activation Fraction, an asymptote of the measured SPAR curve at large particle sizes) is close to 1.**"

*There appear again lots of abbreviations, such as RexBC. This is explained once, but since it is not common, I had to look it up again and again. I would prefer reading without so many abbreviations.*

**Response: Thanks for your suggestion. We added a table listing the description of these abbreviations as mentioned earlier, and have added the explanations of these abbreviations like RexBC where they are first introduced in each section and in caption of each figure.**

*Line 371, caption figure 4 and others: the word 'composition' is used in the wrong context. The authors mean probably component(s). This appears several times in the text.*

**Response: Thanks for your suggestion. We have revised it accordingly including:**

**L47: "Fossil fuel combustion-emitted BC-containing aerosols tended to be more externally mixed with other aerosol components …"**

**L298: "The mass concentrations of different chemical components …"**

**L308: "The diurnal variations in mass concentrations of different aerosol chemical components …"**

**L363: "the mass fractions of aerosol chemical components"**

**L398 : "the mass fraction of each primary organic aerosol components"**

**Figure 7, 8 and 9: "the mass fraction of secondary aerosol chemical components"**

**Figure 4: "mass fraction of aerosol chemical components"**

*Figure 4: what are the shaded areas? Standard deviations? Uncertainty? This has to be explained in the figure caption! My question is, if the differences e.g., between the different diameters are significant? For me, the shapes of the curves of NF for different diameters look very similar, in particular if the shaded area represents an uncertainty range.*

**Response: Thanks for your suggestion. The shaded areas indicate the standard deviations, The difference between those of different diameters are not significant, especially for particle diameters larger than 100nm. As the reviewer #2 suggested, we keep the sizes with most concurrent measurements, e.g. 150, 200 and 300 nm and move the rest particle sizes into the supplement. In detail, we have revised Figure 4 and its caption as:**

[Figure]

**"Figure 4. (a-l) Diurnal variations of aerosol mixing state parameters (identified by color and marker) at different particle sizes (50, 150, 200 and 300 nm) during the three periods. The shaded areas indicate the standard deviations. MAF (Maximum Activation Fraction): An asymptote of the measured SPAR curve at large particle. $NF_H$: Number Fraction of Hydrophilic aerosol whose hygroscopicity parameter is higher than ~0.07. $NF_V$: Number Fraction of Volatile aerosol whose Shrink Factor at 200 °C is lower than 0.85. $NF_{noBC}$: Number Fraction of BC-free particles. $R_{exBC}$: Number fraction of externally BC particles in total BC-containing particles. (m-o) Diurnal variations of mass fractions of aerosol chemical compositions including secondary organic aerosols (SOA), biomass burning organic aerosol (BBOA), fossil fuel organic aerosols (FFOA), and**

inorganic ions including sulfate (SO4), nitrate (NO3) and ammonium (NH4) (identified by color and marker) during the three periods."

We have revised the corresponding description of Figure 4 as:

"For particle sizes larger than 100 nm (shown in both Fig. 4 and Fig. S5), there were maxima in the afternoon for MAF, $NF_H$, $NF_V$ and $NF_{noBC}$, indicative of a peak during this time due to the increase in secondary aerosol compositions like nitrate and SOA, and the decrease of POA and BC."

We have also added Figure S4 into the supplement as:

[Figure]

"Fig. S5. (a-f) Diurnal variations of aerosol mixing state parameters (identified by color and marker) at different particle sizes (100 and 250 nm) during the three periods. The shaded areas indicate the standard deviations. (g-i) Diurnal variations of mass fraction of aerosol chemical compositions (identified by color and marker) during the three periods."

*In the description of this figure 4 the word 'peak' is frequently used, but I see only slight maxima between different times of the day. This has to be checked and needs to be adapted.*

**Response: Thanks for your suggestion. It should be a maxima between different times of the day and is indicative of a peak in the afternoon. We have revised corresponding descriptions of figure 4 as:**

"For particle sizes larger than 100 nm (shown in both Fig. 4 and Fig. S5), there were maxima in the afternoon for MAF, $NF_H$, $NF_V$ and $NF_{noBC}$, indicative of a peak during this time due to the increase in secondary aerosol compositions like nitrate and SOA, and the decrease of POA and BC."

"In the clean period, there was another maxima at midnight for MAF and $NF_{noBC}$, which may be attributed to the diurnal variations of secondary aerosol compositions like sulfate and SOA, and the decrease of BC and FFOA."

*Line 388: What means 'consistency' here? I simply do not understand it.*

**Response: We are referring to the agreement between different aerosol mixing state parameters and we have revised this sentence as:**

"The agreement between MAF and $NF_V$ was slightly higher than that between $NF_H$ and MAF (or $NF_V$) with similar correlation coefficients (~0.65) …"

*Figures 5 – 9: linear correlations were fitted here, but the results do not always look convincing. E.g., Figure 7: 2 lowest plots show dots widely distributed and one does not expect a linear correlation. What is the meaning of such a correlation? I strongly suggest to check the quality of these correlations and reduce to number of these plots.*

**Response: Thanks for your suggestion. In former studies on the aerosol mixing state, it is common to investigate the linear correlations between aerosol mixing state parameters as well as aerosol chemical compositions (Reference listed in the introduction like: Zhang et al., 2014; Hong et al., 2017; Kim et al., 2020; Tao et al., 2021). In this study, the correlation analysis is used to examine whether the primary emissions or the secondary aerosol formations have significant impacts on the aerosol mixing state parameters, and for some cases there was no significant influences which also provide insights into investigating variations of mixing state parameters. In order to avoid misunderstanding and highlight the our findings, we have removed the fit lines. In detail, we have revised Figure 5-9 as:**

**Figure 5:**

[Figure]

**Figure 6:**

[Figure]

**Figure 7:**

[Figure]

**Figure 8:**

[Figure]

**Figure 9:**

[Figure]

*Figure 8: the lower plots seem to follow more an exponential growth, does the linear fit makes sense here?*

**Response: As we mentioned in the former response, the correlation analysis of this study is used to qualitatively explore whether the primary emissions or the secondary aerosol formations affect significantly on variations of aerosol mixing state**

**parameters, as commonly applied in former studies on the aerosol mixing state (Reference listed in the introduction like: Hong et al., 2017; Kim et al., 2020; Tao et al., 2021). We have removed the fit lines with correlation coefficient (r) less than 0.5 as shown in the former response.**

*Figure 9: what means OOA1 and OOA2?*

**Response: OOA1 and OOA2 are two SOA factors from the PMF analysis of organic aerosol. As mentioned in Section 2.1, these two SOA factors were found to display different spectral patterns, correlations with tracers and diurnal variations, suggesting that they resulted from different chemical processing, however, the formation mechanism and possible precursors are yet to be explored in future. For example, OOA1 had higher $CO_2^+/C_2H_3O^+$ (3.9) and O/C (0.91) ratios compared to OOA2 (2.1, 0.78). We have revised this caption as:**

**"Figure 9.  The correlation between the difference among the four aerosol mixing state parameters and mass fractions (MF) of secondary aerosol chemical components during different periods. OOA1 and OOA2 are two SOA factors resolved from AMS measurements using the PMF technique. Moderately polluted period: Blue circle; Heavily polluted period: Red square; Clean period: Green pentagon. $NF_A$-$NF_B$ ($NF_{noBC}$-$NF_H$, $NF_V$-$NF_H$, $NF_{noBC}$-$NF_V$): The difference between the number fraction of A and B. $NF_H$: Number Fraction of Hydrophilic aerosol whose hygroscopicity parameter is higher than ~0.07. $NF_V$: Number Fraction of Volatile aerosol whose Shrink Factor at 200 °C is lower than 0.85. $NF_{noBC}$: Number Fraction of BC-free particles."**

*All figure captions need more text to explain the figure. One should understand the general content of a figure without reading the full text around.* **°C**

**Response: Thanks for your suggestion. Besides the captions of Figures 1, 4 and 9 whose revision have been mentioned earlier, the captions of other figures are revised as:**

[revised manuscript text omitted]

*Line 472: exemplarily 'difference between NFV – NFH' means difference between NFV and NFH? This appears several times around this section.*

**Response: Yes, we have revised it as "difference between $NF_V$ and $NF_H$ ($NF_V$-$NF_H$)". Similar revisions includes:**

**L427-434: "The difference between $NF_{noBC}$ and $NF_H$ ($NF_{noBC}$-$NF_H$) both had significant positive correlations with $MF_{FFOA}$ and $MF_{BBOA}$ (r>0.5), suggesting that a substantial proportion of POA resided in BC-free aerosols and was volatile but contributed substantially to nearly hydrophobic aerosols. So was the difference between $NF_V$ and $NF_H$ ($NF_V$-$NF_H$). The mass fractions of BBOA and FFOA were poorly linked with the difference between MAF and $NF_V$ (MAF-$NF_V$), or MAF and $NF_{noBC}$ (MAF-$NF_{noBC}$), or $NF_V$ and $NF_{noBC}$ ($NF_V$-$NF_{noBC}$) (Fig. S7). The difference between MAF and $NF_H$ (MAF-$NF_H$) had a positive correlation with $MF_{BBOA}$, further suggesting BBOA contributed to nearly hydrophobic aerosols under subsaturated conditions, however, their hygroscopicity was enhanced and became CCN-active at supersaturated conditions."**

**L464: "Difference between $NF_{noBC}$ and $NF_H$ ($NF_{noBC}$-$NF_H$) showed a strong negative correlation with $MF_{NH4}$ and $MF_{NO3}$. So did the Difference between $NF_V$ and $NF_H$ ($NF_V$-**

$NF_H$).”

**L483**: “The difference between $NF_{noBC}$ and $NF_V$ ($NF_{noBC}-NF_V$) is negatively correlated with $MF_{NO3}$, which is consistent with the semi-volatile nature of nitrate.”

**L555**: “… the two resolved SOA factors exhibited different impacts on the difference between $NF_V$ and $NF_H$ ($NF_V-NF_H$), …”

**Figure 10**: “… (d and e) The variations of the difference between Number Fraction of Volatile aerosol and BC-free particles ($NF_V-NF_{noBC}$, blue large circle) and the difference between $NF_V$ and the number fraction of thickly coated BC containing aerosols ($NF_{CBC}$) plus $NF_{noBC}$ ($NF_V-(NF_{noBC}+NF_{CBC})$, yellow small circle) with the mass concentration of SA at particle size of 200nm (d) and 300nm (e).”

*Minor comments/ typos:*

*Comment: I did not look explicitly for all typos, because I think, several parts need to be rewritten und after that it should be read again carefully.*

*Line 293: PA means probably POA*

*Line 319: in large diameters ranges*

*Line 334:a dot after 'size' is missing*

*Line 439: 'are presented' should be 'is presented'*

**Response: Thanks for your suggestion and we have revised them accordingly. We have also checked the manuscript again and other revisions include:**

**L333**: “This particle group had the  highest fraction (higher than 0.7) during the heavily polluted period and the  lowest fraction (down to 0.5) during the clean period, …”

**L352**: “non-negligible fractions of BC-free aerosols dominated  these less volatile aerosol components”

**L441**: “… had a strong positive correlation with $MF_{NH4}$  …”

**L412**: “… demonstrating  significant contributions …”

**L413**: “($NF_H$ and $NF_{noBC}$ are larger and smaller than 0.7 when $MF_{FFOA}$ was larger than 0.1, respectively)”

**L432**: “… further suggesting that BBOA contributed to nearly …”

---

## Author Comment (AC2)

Reviewer #2:

*General comments:*

*The present paper focuses on improving our understanding of the aerosol mixing state in a background site of the North China Plain in China. This is achieved by combining various techniques, including HTDMA, CCN counter, VTDMA, and SP2. The study provides a first-time intercomparison of the four aerosol mixing state parameters from the instruments above and offers insights into the interlink among these parameters and potential sources. I find this research to be important and interesting for aerosol mixing state studies. The manuscript is well-structured and scientifically engaging for the aerosol society. However, in terms of writing, it would be beneficial for non-expert readers if certain sentences were shortened and explanations were provided before reaching conclusions. Please see the detailed comments below. I suggest publishing the manuscript after a minor revision.*

**Response: Thanks for your comments. Suggestions and comments are addressed point-by-point and corresponding responses are listed below.**

*Specific comments:*

*1) Line 70. "..lead to substantial overestimation". Could you provide more details about the magnitude of the substantial overestimation?*

**Response: Thanks for your suggestion. There can be overestimation of NCCN from 10% to 30%, and we have revised this sentence as:**

**"Using simple internally mixing state assumptions for aerosol chemical compositions in estimating CCN number concentrations can lead to substantial overestimations (up to 30%, Deng et al., 2013; Farmer et al., 2015; Ren et al., 2018; Ching et al., 2017, 2019; Tao et al., 2021)."**

*2) Line 94. "highly correlated to those of a VTDMA at high temperature". Which temperature do you refer to and why?*

**Response: In general, in order to remove most non-refractory materials in aerosol, 300 °C is used in VTDMA measurement (Philippin et al., 2004; Wehner et al., 2009; Zhang et al., 2014; Hong et al., 2017; Wang et al., 2022). But this temperature can be lower to 200 °C depending on the aerosol chemical compositions. We have revised this sentence as:**

**"Thus, measurements of an SP2 are highly correlated to those of a VTDMA at high temperatures (higher than 200 °C and up to 300 °C), with their differences reflecting**

**variations in aerosol density, shape or volatility (Philippin et al., 2004; Wehner et al., 2009; Adachi et al., 2018, 2019; Wang et al., 2022)."**

*3) Line 127-129. Please summarize the key messages of the meteorology influences on the aerosol mixing state.*

**Response: Thanks for your suggestion. As reported by Kuang et al., 2020, the secondary aerosol formations under low RH conditions, mainly taken place in gaseous phase, would change to that mainly taken place in aqueous phase under high RH conditions. As secondary aerosol formed through different mechanism have different chemical compositions and add mass to different particle groups, secondary aerosol formations under different meteorological conditions can affect the aerosol mixing states (Tao et al., 2021). We have revised this sentence as:**

**"Meteorological conditions can greatly impact the secondary aerosol formation in the NCP, which can be significantly exacerbated during severe pollution events. The secondary aerosol formations under low RH conditions, mainly taken place in gaseous phase, would change to that mainly taken place in aqueous phase under high RH conditions (Kuang et al., 2020). As secondary aerosols formed through different mechanisms have different chemical compositions and add mass to different particle groups, secondary aerosol formations under different meteorological conditions can affect the aerosol mixing states (Tao et al., 2021)."**

*4) Line 168 BBOA, line 173 FFOA.., please explain the full name when introducing a new term and check out the remaining of manuscript.*

**Response: Thanks for your suggestion. The full names of BBOA and FFOA are Biomass Burning Organic Aerosol and Fossil Fuel Organic Aerosol, respectively. We have added a table listing the definition and description of the abbreviations as follow:**

**Table 1. Definition and description of the abbreviations.**

| Abbreviations | Full name and/or Definition |
|---|---|
| BBOA | Biomass Burning Organic Aerosol

In this study, characterized by obvious m/z 60 (mainly $C_2H_4O_2^+$) and 73 (mainly $C_3H_5O_2^+$), which are two indicators of biomass burning |
| FFOA | Fossil Fuel Organic Aerosol |

| | |
|---|---|
| | A mixed factor in this study that comprises traffic emissions and coal combustions, which was characterized by typical hydrocarbon ion series |
| OOA | Oxidized Organic Aerosol |
| OOA1 and OOA2 | Two SOA factors derived from the PMF analysis |
| SOA | Secondary Organic Aerosols

Summation of BBOA and FFOA |
| POA | Primary Organic Aerosols

Summation of OOA1 and OOA2 |
| $PM_{2.5}$ | Particulate Matter with aerodynamic diameter less than 2.5 µm |
| $PM_1$ | Particulate Matter with aerodynamic diameter less than 1 µm |
| $NR\text{-}PM_1$ | Non-refractory $PM_1$ |
| MF | Mass Fraction |
| $D_p$ | Particle diameter after humidification or heating |
| $D_d$ | Particle diameter under dry conditions without humidification or heating |
| κ | Hygroscopicity parameter |
| SS | Super-saturation |
| SPAR | Size-resolved Particle Activation Ratio

Size-dependent CCN activity under a specific SS |
| MAF | Maximum Activation Fraction

An asymptote of the measured SPAR curve at large particle sizes and represents the number fraction of CCNs to total particles |

| | |
|---|---|
| $D_a$ | Midpoint activation diameter

Linked to the hygroscopicity of CCNs |
| GF | Growth factor

The ratio between particle with and without humidification, and is linked to aerosol hygroscopicity |
| SF | Shrink Factor

The ratio between particle with and without heating, and is linked to aerosol volatility |
| PDF | Probability Distribution Function |
| $NF_H$ | Number Fraction of Hydrophilic aerosol whose hygroscopicity parameter is higher than ~0.07. |
| $NF_V$ | Number Fraction of Volatile aerosol whose Shrink Factor at 200 °C is lower than 0.85. |
| $NF_{noBC}$ | Number Fraction of BC-free particles |
| $NF_{CBC}$ | Number Fraction of thickly coated BC particles |
| $R_{exBC}$ | Ratio of the number concentration between externally BC particles and BC-containing particles.

Externally BC particles are defined as BC-containing particles in this study. |
| $NF_A$-$NF_B$

($NF_{noBC}$-$NF_H$,$NF_V$-$NF_H$, $NF_{noBC}$-$NF_V$,$NF_V$-MAF, $NF_{noBC}$-MAF) | The difference between the number fraction of A and B. |

**In addition, we have also added the definition and description of the abbreviations when first introduced in each section and also in the caption of the figures for clarification.**

*5) Line 177, what do you mean by "different chemical process" and could you give more details?*

**Response: Secondary organic aerosol formations originated from volatile organic compounds precursors could be formed in differ formation pathways such as aqueous phase reactions, heterogeneous reactions or gas phase reactions and also might be oxidized under different conditions, for example oxidized under different nitrogen oxide conditions with different oxidation capacity and oxidants. The following sentences is added in the revised manuscript.**

"Secondary organic aerosol formations originated from volatile organic compounds precursors could be formed in differ formation pathways such as aqueous phase reactions, heterogeneous reactions or gas phase reactions and also might be oxidized under different conditions, for example, oxidized under different nitrogen oxide conditions with different oxidation capacity and oxidants. Two resolved SOA factors were found to display different spectral patterns, correlations with tracers and diurnal variations, suggesting that they resulted from different chemical processing, however, their formation mechanisms remain to be explored in our future studies. In general, the OOA factor 1 (Oxidized Organic Aerosol, OOA1) had higher $CO_2^+/C_2H_3O^+$ (3.9) and O/C (0.91) ratios compared to the OOA factor 2 (OOA2, 2.1 and 0.78)."

*6) Line 187, why do you choose these three supersaturations for CCN measurements?*

**Response: As particle size is the most important parameter in determining CCN activity (Duesk et al., 2006), measurement of CCN activity can indicate particle hygroscopicity in different particle size ranges. In general, the three supersaturations indicate the particle hygroscopicity in particle size range from 100 nm to 200 nm.**

**In order to perform intercomparisons among instruments, three supersaturations (SSs) of 0.08%, 0.14% and 0.22% were applied in a single cycle of about 15 minutes. CCN measurement under these three SSs reveals mainly CCN activity of aerosols reside in accumulation mode aerosol with diameter range of about 100-200 nm, which are close to diameters of HV-TDMA measurements, and higher SSs would reveal CCN activity of smaller aerosol particles (<100 nm) where DMA-SP2 measurement is not available:**

"In order to perform intercomparisons among instruments, three supersaturations (SSs) of 0.08%, 0.14% and 0.22% were applied in a single cycle of about 15 minutes. CCN measurement under these three SSs reveals mainly CCN activity of aerosols reside in accumulation mode aerosol with diameter range of about 100-200 nm, which are close to diameters of HV-TDMA measurements, and higher SSs would reveal CCN activity of smaller aerosol particles (<100 nm) where DMA-SP2 measurement is not available."

*7) Line 211, the maximum temperature you chose is 200 degree Celsius, why do you choose this threshold?*

**Response: The HV-TDMA were scanning at different temperatures and diameters for the HV-TDMA system, to ensure the time duration of one full cycle is about 3 h, we limited the number of temperatures and diameters. Most importantly, results of previous studies in the North China Plain have shown that 200 degree Celsius is enough for removing most non-refractory aerosols (>80%) (Xu et al., 2019).**

*8) Line 225-229, regarding the chosen size for SP2, which system was conducted for this study, with or without thermodenuder-bypass? Since you are expected to compare with HTDMA and VTDMA, why not choose the same sizes to measure for the three instruments?*

**Response: The DMA-SP2 system was conducted both with and without thermodenuder-bypass depends on time, and detailed periods are added in the revised manuscript. Compared to HTDMA and VTDMA, more particle sizes are selected in the measurement DMA-SP2 system for obtaining more information of BC mass concentration and mixing states at different particle diameters for other scientific purposes. Because the time needed for a single particle size measurement of DMA-SP2 system is much shorter than that of HTDMA and VTDMA, and one full cycle for H/VTDMA lasts 3 hours. We have added corresponding description into the manuscript as:**

**"The DMA-SP2 setup was able to measure the mixing states of aerosols at diameters of 100 nm, 120 nm, 160 nm, 200 nm, 235 nm, 270 nm, 300 nm, 335 nm, 370 nm, 400 nm, 435 nm, 470 nm, 500 nm, 535 nm, 570 nm, 600 nm, 635 nm, 670 nm, 700 nm within 20 minutes, when it wasn't placed after a thermodenuder-bypass switch system (13th-24th October, 09:00 am of 5th November to 09:00 am of 8th November). However, it only measured mixing states at diameters of 120 nm, 160 nm, 200 nm, 250 nm, 300 nm, 400 nm, and 500 nm when it was placed after a thermodenuder-bypass switch system (11:00 am 24th October to 08:00 am 5th November, and 09:00 am of 8th November to 06:00 pm of 17th November). Because the measurements of HTDMA and VTDMA are conducted solely by a single H/VTDMA system working in different mode, the time needed for a single particle size measurement of HTDMA and VTDMA is much longer than that of DMA-SP2 system. Thus, more particle sizes are selected in the measurement DMA-SP2 system for acquiring BC mass concentration and mixing state at more diameters, compared to those of HTDMA and VTDMA"**

*9) Line 235, does the flow rate influence the measurements and by how much?*

**Response: This change satisfied the flowrate requirements of this instrument (0.03 to 0.18 L/min), and 0.12 L/min was typically used. The flow rate change does not affect the measurements when aerosol number concentration is not small. Actually, at the very beginning, 0.1 L/min (less than the typical one 0.12L/min) was usually used because the NCP is generally polluted, and higher flow rate would produce larger data storage, however, does not affect the statistical results. We change to 0.12 L/min is because that we realized that we scan up to 700 nm using the DMA-SP2 which is different with previous studies where aerosol number concentration is much smaller and a larger sample flow rate should be better.**

*10) Section 2.3.1, the MAF is a fitting parameter from eq.7, what is the physical meaning of this parameter? Is it the maximum activation fraction?*

**Response: Yes, it's the maximum activation fraction and we have revised the corresponding description as:**

**".. MAF (Maximum Activation Fraction) is an asymptote of the measured SPAR curve at large particle sizes as shown in Fig. S4, and it represents the number fraction of CCNs to total particles. …"**

**To be noted, a schematic of the SPAR parameterization scheme and the corresponding fitting parameters is added into the supplement for clarification as:**

[Figure]

**Fig. S4. Schematic of the parameterization scheme of SPAR curves. The black solid**

curve and the black crossing are the measured SPAR and fitted SPAR with the parameterization scheme. The red, green and blue dashed lines indicate the fitting parameters of Maximum Activation Fraction (MAF), the midpoint activation diameter (Da) and s, respectively.

11) Line 267, add sizes for the GF "The GFC for the four measured particle sizes were 1.1, 1.15, 1.175 and 1.2".

Response: The GFC for particle size of 50, 100, 150 and 200 nm are 1.1, 1.15, 1.175 and 1.2, respectively. We have revised it as:

"The $GF_C$ for the four measured particle sizes of 50 nm, 100 nm, 150 nm and 200 nm were 1.1, 1.15, 1.175 and 1.2, respectively"

12) Section 2.3.3. Here you use the lag time between the peak of the scattering signal and the incandescence signal to classify the bare and coated BC. Is it related to the BC-coating mass ratio? The mass ratio is more commonly used and intuitive to understand.

Response: Thanks for your suggestion. It is related to the coating thickness of the BC-containing aerosols. The BC-coating mass ratio cannot be directly obtained in SP2 measurement, due to the lack in the accurate density and shape of the BC core. In addition, the lag time is positively correlated to the coating thickness, but their relation cannot be directly quantified and also calibrated. Thus, a critical value of lag time rather than coating thickness or coating mass ratio is used to classify the bare and coated BC. We have revised this sentence as:

"In this study, a two-mode distribution of the lag time (Δt) was observed. As the lag time is positively correlated to the coating thickness, a critical lag time (0.8 μs) was used to classify the BC-containing particles into thinly coated (or bare) BC (Δt < 0.8 μs) and thickly coated BC (Δt ≥ 0.8 μs), respectively."

13) Line 297-299, please give exact values of PM mass for the heavily polluted and clean period.

Response: Thanks for your suggestion. Non-refractory $PM_1$ mass for the heavily polluted and clean period are 49.5±22.5 and 5.1±3.3 μg/m³, respectively. We have revised this sentence as:

"The mass concentrations of different aerosol compositions increased significantly from October 23rd to November 6th (heavily polluted period with average nonrefractory PM$_1$ mass concentration of 49.5±22.5 μg/m$^3$) and decreased to much lower levels after November 6th(clean period with non-refractory PM$_1$ mass concentration of 5.1±3.3 μg/m$^3$)."

14) Line 315-316. "At lower SSs, the rapid increases in SPAR curves occur at larger particle sizes and the maximum AR of SPAR curves becomes smaller". Please briefly explain why.

Response: Thanks for your suggestion. For lower SSs, particle size need for CCN activation is larger, thus SPAR curve start to increase from 0 at larger particle size. Because only SPAR in particle size lower than 300 nm is presented and there was less particle to be CCN active under low SSs, the maximum AR of SPAR curves becomes smaller under low SSs. We have revised this sentence as:

"At lower SSs, the rapid increases in SPAR curves occur at larger particle sizes, since particle size need for CCN activation is larger. In addition, as SPAR in particle size lower than 300 nm is presented, the maximum AR of SPAR curves becomes smaller as there was less particle to be CCN active under low SSs."

15) Line 318, add SS for the "increases in SPAR curves, are approximately 90 nm, 120 nm and 180 nm"

Response: Thanks for your suggestion. We have revised it as:

"… increases in SPAR curves, are approximately 90 nm, 120 nm and 180 nm for the three SSs of 0.08%, 0.14% and 0.22%, respectively."

16) Fig 2. Are bars representing the standard deviation of the campaign average?

Response: Yes, and we have added corresponding description in the end of the caption of Figure 2 as:

"The shaded areas indicate the standard deviations."

17) Line 331-333, "In general, the size dependence of MAF, NFH, NFV and NFnoBC were similar to one another, suggesting they were dominated by the same particle group, namely BC-free particles". I think this statement is not well supported, I would suggest weakening it or proving it with more evidence. For example, thickly coated BC particles can be very CCN-activate, hydrophilic and volatile, if mostly contain SIA.

Response: Thanks for your suggestion and we fully agree. We have revised this sentence as:

"In general, the size dependence of MAF, $NF_H$, $NF_V$ and $NF_{noBC}$ were similar to one another, suggesting they were likely dominated by the same particle group, namely BC-free particles."

*18) Line 335, please provide exact values of the fraction of BC-containing particles and the applied diameter range, because the terms "higher" or "larger" are not accurate. Check out similar issues for the remaining manuscript too.*

**Response: Thanks for your suggestion. We have revised this sentence as:**

"This suggests that primary emissions tend to have higher fractions of BC-containing particles in larger diameter ranges, for example, the fraction of BC-containing particles increases from ~0.1 to ~0.4 as particle size enlarges from 200 nm to 500 nm during the clean period."

**We have also checked the manuscript and revised the following:**

**L333:** "This particle group had the highest fraction (higher than 0.7) during the heavily polluted period and the lowest fraction (down to 0.5) during the clean period, …"

**L347:** "… when the nitrate fraction was the highest (~30%) and the SOA fraction was the lowest (~7%) among all three periods, …"

**L354-355:** "but were larger than $NF_H$ (by ~0.2) during the moderately and heavily polluted periods when the POA/SOA fractions were higher (~40% vs ~35%)."

*19) Line 342, what do you mean by "the more efficient secondary aerosol formation", increase by secondary aerosol mass or particle size?*

**Response: Here we are referring to that the formation rate of secondary aerosol mass is more efficient on larger particle, and we have revised this sentence as:**

"… while the decrease of RexBC with increasing particle diameter size in the polluted period confirms secondary aerosol formation to be more efficient on particles with larger diameter."

*20) Line 356-357, what is the kappa value for hydrophobic mode aerosol?*

**Response: The kappa value for hydrophobic mode aerosol is less than 0.07 and we have revised this sentence as:**

"The critical $\kappa$ of hydrophilic mode aerosols was 0.07, suggesting that a higher fraction of aerosols had $\kappa$ below 0.07 (i.e. hydrophobic mode aerosol in this study) during the

*21) Line 361, how do you get this statement with "lower than 0.07 but still CCN active"? please explain in detail.*

**Response: In this part we are referring to that the difference among MAF, NF$_V$, NF$_H$ and NF$_{noBC}$ and we found that NF$_H$ is significantly smaller than the other three parameters. This may indicate that a portion of particles to be CCN active but not hydrophilic, i.e. with $\kappa$ lower than 0.07. We have revised this sentence as:**

**"The NF$_H$ was consistently lower than NF$_V$ and NF$_{noBC}$ (the average difference between NFH and NF$_{noBC}$ was about 0.2), especially during the moderately polluted period. As mentioned above that NF$_H$ was also lower than MAF during the moderately polluted periods, there may be a significant fraction of volatile BC-free aerosols had hygroscopicity lower than critical $\kappa$ of 0.07 but were still CCN-active and therefore not fully hydrophobic."**

*22) Fig4, I would suggest simplifying the plot and keeping the sizes with most concurrent measurements, e.g. 150, 200 and 300 nm. Put other sizes to the supplement.*

**Response: Thanks for your suggestion. We have revised Figure 4 and its caption as:**

[Figure]

**"Figure 4. (a-l) Diurnal variations of aerosol mixing state parameters (identified by color and marker) at different particle sizes (50, 150, 200 and 300 nm) during the three periods. The shaded areas indicate the standard deviations. MAF (Maximum Activation Fraction): An asymptote of the measured SPAR curve at large particle. $NF_H$: Number Fraction of Hydrophilic aerosol whose hygroscopicity parameter is higher than ~0.07. $NF_V$: Number Fraction of Volatile aerosol whose Shrink Factor at 200 °C is lower than 0.85. $NF_{noBC}$: Number Fraction of BC-free particles. $R_{exBC}$: Number fraction of externally BC particles in total BC-containing particles. (m-o) Diurnal variations of mass fractions of aerosol chemical compositions including secondary organic aerosols (SOA), biomass burning organic aerosol (BBOA), fossil fuel organic aerosols (FFOA), and**

inorganic ions including sulfate (SO4), nitrate (NO3) and ammonium (NH4) (identified by color and marker) during the three periods."

**We have revised the corresponding description of Figure 4 as:**

"For particle sizes larger than 100 nm (shown in both Fig. 4 and Fig. S5), there were maxima in the afternoon for MAF, NF$_H$, NF$_V$ and NF$_{noBC}$, indicative of a peak during this time due to the increase in secondary aerosol compositions like nitrate and SOA, and the decrease of POA and BC."

**We have also added Figure S4 into the supplement as:**

[Figure]

"Fig. S5. (a-l) Diurnal variations of aerosol mixing state parameters (identified by color and marker) at different particle sizes (50, 150, 200 and 300 nm) during the three periods. The shaded areas indicate the standard deviations. (m-o) Diurnal variations of mass fractions of aerosol chemical compositions (identified by color and marker) during the three periods."

*Line 362- 366, the diurnal variations should be described more explicitly as the pattern of RexBC is clearly different from the other three mixing state parameters and explain why.*

**Response: Thanks for your suggestion. We have added more discussion in the end of this paragraph as:**

"R$_{exBC}$ tended to be lower during the daytime and its diurnal variation was more

significant in larger particle sizes. In general, these diurnal variations for $R_{exBC}$ are opposite to those of $NF_{noBC}$ and secondary aerosol mass fractions, and agree better with those of the primary aerosol mass fractions. This is because BC particles originate from primary emissions and are mainly externally mixed. After experiencing aging process in the atmosphere, BC particles can be coated by secondary aerosol formed on, resulting in more coated BC particles and less externally mixed BC particles. As the secondary aerosol tends to form on larger particles, the diurnal variations of secondary aerosol formations may affect more significantly on those of mixing state of BC particles and thus RexBC in larger particle sizes."

*23) Line 384, table S1 is quite interesting for readers thus I suggest putting it or making a correlation plot into the main context.*

**Response: Thanks for your suggestion. We agree that useful information is contained in this table, we also struggled before we decided to put it in the supplement. We want readers focus more on key parts of those intercomparison results, however, it was also available in the supplement in case that readers want to know all scenarios.**

*24) Line 385, why do you choose these three sizes? The critical size for the setting SS?*

**Response: As shown in Figure 2, the particle size where the rapid increases in SPAR curves are approximately 90 nm, 120 nm and 180 nm for the three SSs of 0.22%, 0.14% and 0.08%, respectively. And the diameter range of rapid increases in SPAR curves are determined by aerosol hygroscopicity in this particle size ranges. Thus, the three particle sizes of 100 nm, 150 nm and 200 nm are chosen in comparison to the MAF at the three SSs of 0.22%, 0.14% and 0.08%, respectively. We have revised this sentence as:**

"Note that MAF at SSs of 0.08%, 0.14% and 0.22% was used for comparison at particle sizes of 200 nm, 150 nm and 100 nm. This is because that the diameter range of rapid increases in SPAR curves are determined by aerosol hygroscopicity in this particle size range, and the midpoint of rapid increase diameter ranges of SPAR curves at SSs of 0.08%, 0.14% and 0.22% are approximately 180 nm, 120 nm and 90 nm (as shown in Fig. 2). "

*25) Line 386. A classification of the correlation should be clarified, such as the r range for the weak, moderate, and strong correlation.*

**Response: Thanks for your suggestion. The value range of correlation coefficient for**

weak, moderate and strong was generally less than 0.3, from 0.3 to 0.5 and larger than 0.5. We have added detailed value of correlation coefficient into the manuscript including:

L386: "… moderate correlations (r~0.5) .."

L392: "… the correlation became weaker (r~0.4), …"

L421: "…, and weak correlations (r<0.3) …"

L440: "… a strong positive correlation with $MF_{SO4}$ (r>0.5). …"

L445: "… the weaker correlations with SOA (r~0.3) seen in Fig. 8."

L454: "…, the strong positive correlations between $NF_V$ and secondary aerosol formations (r~0.6) …"

L457: "… strong positive correlations (r~0.5) …"

L465: "… a strong negative correlation with $MF_{NH4}$ and $MF_{NO3}$ (mainly -0.6) …"

*26) Line 392, what do you mean by saying "..while the degree was the least for the correlation.."?*

**Response: We are referring to that the degree of the reduction of correlation was the least for the correlation between MAF and NFV, and we have revised this sentence as:**

"At smaller particle size, the correlation became weaker (r~0.4), while the degree of the reduction was the least for the correlation between MAF and $NF_V$."

*27) Fig. 5, what is the r in the plot? It would be more intuitive to use the same marker to represent different periods in the plot.*

**Response: The variable r represent the correlation coefficient and we have added corresponding description into the caption as "with r representing the correlation coefficient." At the request of the Copernicus Publications, the marker used to present different periods are set to be different in order to making this figure friendly to readers with color vision deficiencies.**

*28) A summary table (or correlation matrix plot) of r in Fig5-7 will be helpful for readers to better understand the interlink between mixing state parameters and chemical composition.*

**Response: Thanks for your suggestion and the correlation between mixing state parameters and aerosol chemical composition as well as detailed correlation during**

different pollution periods were summarized in Figures S6 and S8 (Figures S5 and S6 in old version). We have added introduction of these figures into the manuscript as:

In the end of Section 3.3: "In addition, the correlation between mixing state parameters and primary aerosol composition during the campaign and different pollution periods were summarized in Fig. S7."

In the end of last second paragraph of Section 3.4: "Besides, the correlation between mixing state parameters and secondary aerosol composition during the campaign and different pollution periods were summarized in Fig. S9."

*29) Line 400, please give values to the sentence "correlation with MFFFOA was much weaker compared to MFBBOA".*

Response: Thanks for your suggestion. We have revised this sentence as:

"However, the correlation with $MF_{FFOA}$ (-0.45~-0.74) was much weaker compared to $MF_{BBOA}$ (-0.10~-0.45)."

*30) Fig.7. Which size of data do you use?*

Response: The size is 200 nm and we have added corresponding description into the caption of Figure 7 as "The correlation between the difference among the four aerosol mixing state parameters at particle size of 200 nm and MF of primary organic aerosol components during different periods."

*31) Line 428, please introduce what the difference (NFnoBC-NFH and NFV-NFH) represents first before jumping to the results.*

Response: Thanks for your suggestion. We have added the definition of these abbreviations as:

L427-434: "The difference between $NF_{noBC}$ and $NF_H$ ($NF_{noBC}$-$NF_H$) had significant positive correlations with $MF_{FFOA}$ and $MF_{BBOA}$ (r>0.5), suggesting that a substantial proportion of POA resided in BC-free aerosols and was volatile but contributed substantially to nearly hydrophobic aerosols. So did the difference between $NF_V$ and $NF_H$ ($NF_V$-$NF_H$). The mass fractions of BBOA and FFOA were poorly linked with the difference between MAF and $NF_V$ (MAF-$NF_V$), or MAF and $NF_{noBC}$ (MAF-$NF_{noBC}$), or $NF_V$ and $NF_{noBC}$ ($NF_V$-$NF_{noBC}$) (Fig. S7). The difference between MAF and NFH (MAF-$NF_H$) had a positive correlation with $MF_{BBOA}$, further suggesting BBOA contributed to nearly hydrophobic aerosols under subsaturated conditions, however, their hygroscopicity

was enhanced and became CCN-active at supersaturated conditions."

L464: "Difference between $NF_{noBC}$ and $NF_H$ ($NF_{noBC}$-$NF_H$) showed a strong negative correlation with $MF_{NH4}$ and $MF_{NO3}$. So did the Difference between $NF_V$ and $NF_H$ ($NF_V$-$NF_H$). So did the difference between $NF_V$ and $NF_H$ ($NF_V$-$NF_H$)."

L483: "The difference between $NF_{noBC}$ and $NF_V$ ($NF_{noBC}$-$NF_V$) is negatively correlated with $MF_{NO3}$, which is consistent with the semi-volatile nature of nitrate."

L555: "… the two resolved SOA factors exhibited different impacts on the difference between $NF_V$ and $NF_H$ ($NF_V$-$NF_H$), …"

Figure 10: "……(d and e) The variations of the difference between $NF_V$ and $NF_{noBC}$ ($NF_V$-$NF_{noBC}$, blue large circle) and the difference between $NF_V$ and $NF_{noBC}$+$NF_{CBC}$ ($NF_V$-($NF_{noBC}$+$NF_{CBC}$), yellow small circle) with the mass concentration of SA at particle size of 200nm (d) and 300nm (e). MAF (Maximum Activation Fraction): An asymptote of the measured SPAR curve at large particle. $NF_H$: Number Fraction of Hydrophilic aerosol whose hygroscopicity parameter is higher than ~0.07. $NF_V$: Number Fraction of Volatile aerosol whose Shrink Factor at 200 °C is lower than 0.85. $NF_{CBC}$: Number Fraction of thickly coated BC particles."

*32) Line 438, why do you choose 200nm?*

**Response: This is mainly because we focus on the comparison of the four aerosol mixing state as well as their relationship with aerosol chemical compositions, but only in 200 nm were all the four aerosol mixing state parameters measured. We have added corresponding description into the manuscript as:**

"To be noted, in order to compare the four aerosol mixing state as well as their relationships with aerosol chemical compositions at the same time, the analysis is conducted in only 200 nm where all the four aerosol mixing state parameters were measured."

*33) Line 459-462, out of curiosity, does the transport of ageing aerosols play a role in the increasing fraction of non-BC particles?*

**Response: The reviewer raised a very interesting topic. Indeed, the transport of aging aerosols could play a role in variations in fraction of non-BC particles, for example, during the clean period. However, for periods of the moderately to heavily polluted, the wind speed generally lower than 2 m/s, with strong local emissions (represented quick increase of rBC and POA in the afternoon) of secondary aerosols formations (represented by quick nitrate and SOA formations), the transport of aging aerosols**

should play a negligible role.

Reference:

Dusek, U., Frank, G., Hildebrandt, L., Curtius, J., Schneider, J., Walter, S., Chand, D., Drewnick, F., Hings, S., and Jung, D.: Size matters more than chemistry for cloud-nucleating ability of aerosol particles, Science, 312, 1375–1378, 2006.

Hong, J., Äijälä, M., Häme, S. A. K., Hao, L., Duplissy, J., Heikkinen, L. M., Nie, W., Mikkilä, J., Kulmala, M., Prisle, N. L., Virtanen, A., Ehn, M., Paasonen, P., Worsnop, D. R., Riipinen, I., Petäjä, T., and Kerminen, V.-M.: Estimates of the organic aerosol volatility in a boreal forest using two independent methods, Atmos. Chem. Phys., 17, 4387–4399, https://doi.org/10.5194/acp-17-4387-2017, 2017.

Kuang, Y., Huang, S., Xue, B., Luo, B., Song, Q., Chen, W., Hu, W., Li, W., Zhao, P., Cai, M., Peng, Y., Qi, J., Li, T., Chen, D., Yue, D., Yuan, B., and Shao, M.: Contrasting effects of secondary organic aerosol formations on organic aerosol hygroscopicity, Atmos. Chem. Phys. Discuss., 2021, 1-27, 10.5194/acp-2021-3, 2021.

Kuang, Y., He, Y., Xu, W., Yuan, B., Zhang, G., Ma, Z., Wu, C., Wang, C., Wang, S., Zhang, S., Tao, J., Ma, N., Su, H., Cheng, Y., Shao, M., and Sun, Y.: Photochemical Aqueous-Phase Reactions Induce Rapid Daytime Formation of Oxygenated Organic Aerosol on the North China Plain, Environmental Science & Technology, 54, 3849–3860, https://doi.org/10.1021/acs.est.9b06836, 2020.

Philippin, S., Wiedensohler, A., and Stratmann, F.: Measurements of non-volatile fractions of pollution aerosols with an eight-tube volatility tandem differential mobility analyzer (VTDMA-8), Journal of Aerosol Science, 35, 185–203, https://doi.org/10.1016/j.jaerosci.2003.07.004, 2004.

Tao, J., Kuang, Y., Ma, N., Hong, J., Sun, Y., Xu, W., Zhang, Y., He, Y., Luo, Q., Xie, L., Su, H., and Cheng, Y.: Secondary aerosol formation alters CCN activity in the North China Plain, Atmos. Chem. Phys., 21, 7409–7427, https://doi.org/10.5194/acp-21-7409-2021, 2021.

Wehner, B., Berghof, M., Cheng, Y. F., Achtert, P., Birmili, W., Nowak, A., Wiedensohler, A., Garland, R. M., Pöschl, U., Hu, M., and Zhu, T.: Mixing state of nonvolatile aerosol particle fractions and comparison with light absorption in the polluted Beijing region, Journal of Geophysical Research: Atmospheres, 114, https://doi.org/10.1029/2008JD010923, 2009.

Wang, Y., Hu, R., Wang, Q., Li, Z., Cribb, M., Sun, Y., Song, X., Shang, Y., Wu, Y., Huang, X., and Wang, Y.: Different effects of anthropogenic emissions and aging processes on the mixing state of soot particles in the nucleation and accumulation modes, Atmos. Chem. Phys., 22, 14133–14146, https://doi.org/10.5194/acp-22-14133-2022, 2022.

Xu, W., Xie, C., Karnezi, E., Zhang, Q., Wang, J., Pandis, S. N., Ge, X., Zhang, J., An, J., Wang, Q., Zhao, J., Du, W., Qiu, Y., Zhou, W., He, Y., Li, Y., Li, J., Fu, P., Wang, Z., Worsnop, D. R., and Sun, Y.: Summertime aerosol volatility measurements in Beijing, China, Atmos. Chem. Phys., 19, 10205-10216, 10.5194/acp-19-10205-2019, 2019.

Zhang, S. L., Ma, N., Kecorius, S., Wang, P. C., Hu, M., Wang, Z. B., Größ, J., Wu, Z. J., and Wiedensohler, A.: Mixing state of atmospheric particles over the North China Plain, Atmospheric Environment, 125, 152–164, 2016.

---

## Referee Report (RR1)

**Second Review on "Markedly different impacts of primary emissions and secondary aerosols formations on aerosol mixing states revealed by simultaneous measurements of CCNC, V/HTDMA and SP2" by Jiangchuan Tao et al.**

The manuscript has improved significantly compared to the first version. Thanks a lot for the improvement!

However, there is still some work, which needs to be done, mainly in the discussion of the results in section 3. Here, a lot of text describes too many similar looking figures. It needs to be indicated, what is really new. Other detailed comments are given below.

Furthermore, this study claims to use for the first time all these measurements and methods in parallel. But, what is the outcome? Which methods are comparable? Are all of them needed? E.g., if one has to reduce to setup to 2 mixing state parameters, which would you recommend? Any other general conclusions regarding the methodology?

Since this is definitely a new approach, it should be discussed and interpreted in the conclusion. Please also indicates in the conclusions which findings are new or you assume them to be new. There are too many 'findings' listed and the reader does not know, what is important.

One general formal comment. The unit liter has the abbreviation 'l', not the capital 'L'. This should be corrected through the whole text.

The table with abbreviations also helps a lot. But could you please put it in an alphabetical order? This would be even better! And the term 'SA' is missing there. Maybe also indicate which parameters refer to certain diameters?!

Comments in detail:

Line number in the following mean the corresponding lines in the manuscript with tracked changes.

Line 158 – 160: please keep the old version to recognize the origin of the abbreviation.

Line 223 ff: Do you think that AMS measurements and PNSD experience similar losses? If you say, that they agree well, this is hypothetical to my impression.

Line 346: How did you choose e.g., the critical GF? Did you plot for each diameter the PDF? Other studies used a common GFc for all diameters, why do you think this is different here? Does this correspond to Figure 2c?

Line 361 and many other times: coating thickness of aerosols: it might be that I am a bit too picky here, but I think a coating thickness is always related to aerosol particles and not to aerosols (mixture of gas and particles), therefore I suggest to check the usage of the word 'aerosols'. I think in most cases it should be 'aerosol particles'.

Line 375, 425 and caption of Figure 1 and 3: The word 'compositions' should be 'components'. I do not see too much sense in 'mass concentrations of aerosol composition'

Figures 4 – 9: Similar parameters are plotted, but I personally think, there are too many figures. For some the particle diameter (200 nm) is given, for others not in the caption. Please add this information in the figure caption. Are all figures really necessary? It would be better to exclude one or two from the main paper or combine some of the results. The reader feels a bit overloaded with so many scatter dots.

Line 532 – 538: The sentence us too long and not understandable for me. Please avoid such long sentences with too much additional information in brackets.

Line 569: the word 'compositions' should be components, as written in the corresponding caption of Figure 8.

Line 633: better 'components' instead of 'compositions'

Line 640: please use 'components' instead of 'compositions'

Section 3, in particular 3.3 and 3.4 are too long and not well connected to other studies. What is really new in your study? Many correlations are obvious and well-known, here you should compare with literature. E.g., that SA increases the hygroscopicity of hydrophobic particles is not new. There are similar examples. I would strongly recommend to remove some of the figures and reduce the text. For the results indicate the well-known facts with references or remove them and highlight those results which are new or opposite to former findings.

Is it really necessary to jump always between NF and MF? This is very confusing for the reader. In between, the word 'fraction' is used and nobody knows, what you mean here, e.g. line 604. Please also state clearly which parameters are related to certain diameters or diameter ranges. This is not clear in the text.

---

## Author Response (AR2)

*Cover letter:*

*We sincerely appreciate your careful inspection of our manuscript; we have taken great care in thoroughly revising the document in line with your invaluable feedback. Additionally, we have further enhanced the quality of this manuscript through using expertise of Elsevier Language Editing Services. We have attached the certificate of language editing for your reference, and you will find our responses to your comments comprehensively addressed below.*

*In addition, we agree with the editor that we should avoid excessive definition of the same parameter in figure captions. However, one of the reviewers said "In my view it is required to explain those abbreviations, which are not widely known, such as MAF, CCOA, regularly again, also in figure captions", this is why we defined these parameters repeatedly in figure captions. We have refined our approach in the revised version. We now provide the initial definitions of these parameters with their respective abbreviations in Table 1. Furthermore, in the figure captions, we present both the abbreviation and the complete term but have refrained from repetitive definitions, according to the guidance provided by the language editor.*

[Figure]

**Certificate of Elsevier**
**Language Editing Services**

**The following article was edited by Elsevier Language Editing Services:**

Markedly different impacts of primary emissions and secondary aerosol formation on aerosol mixing states revealed by simultaneous measurements of CCNC, V/HTDMA and SP2

**Ordered by:**

Jiangchuan Tao

**Estimated Delivery date:**

2023-08-31

**Order reference:**

ASLESTD1017819

[Figure]

**Suggestions and comments are addressed point-by-point and corresponding responses are listed below.**

*L135: "different particle groups": particles are internally mixed, so you cannot talk of particle groups*

**Response: Thanks for your suggestion. We have revised this sentence as:**

**"Because SAs formed through different mechanisms, have different chemical compositions and add mass to different aerosol populations, …"**

*L168: "Inlet changes would affect the dry state aerosol sampling": not clear what this means*

**Response: Thanks for your suggestion. We have revised this part as:**

**"The inlet was switched among three impactors: TSP (Total Suspended Particles), PM2.5 (Particulate Matter with an aerodynamic diameter of less than 2.5 μm), and PM1 (Particulate Matter with an aerodynamic diameter of less than 1 μm). Inlet changes among impactors affect dry-state aerosol sampling owing to ambient aerosols are enlarged through aerosol hygroscopic growth or activation."**

*L172: "by two parallelly assembled Nafion": why two?*

**Response: Thanks for your suggestion. We have revised this part as:**

**"The sampled aerosol was dried by two parallelly assembled Nafion dryers with a length of 1.2 m, two Nafion driers was used because of the high RH and sample flow rate (~16 L/min) during the campaign to ensure drying efficiency. In addition, during autumn and winter in the NCP, ambient air temperature (<20 °C and sometimes <0 °C) can be significantly lower than the room temperature (~24 °C). Therefore, this dryer system can maintain the RH of sampled aerosols to below 20%."**

*L214: "volume concentration derived from AMS and rBC measurements": which density was used?*

**Response: Thanks for your suggestion. We have revised this sentence as:**

**"The average ratio between volume concentration derived from AMS and rBC measurements (densities of compounds are the same as Kuang et al., 2021)"**

*L245: "without conditioning": must be dried. At which RH?*

**Response: Thanks for your suggestion. We have revised this sentence as:**

**"…selecting dried particles without conditioning (RH ~15%) …"**

*L259: "measured in the V mode.": residence time?*

**Response: Thanks for your suggestion. We have revised this sentence as:**

**"…were measured in the V-mode (residence time inside the heated tube to be about 1.6 s; Hong et al., 2017)."**

*L288: "(flow rate range of SP2: 0.03 to 0.18 L/min).": why this variability?*

**Response: Thanks for your suggestion. We have revised this sentence as:**

**"(allowed flow rate range of SP2: 0.03–0.18 L/min from the specification) …"**

*L366: "SOA, POA and BC mass all reached 10": unclear: together or individually?*

**Response: Thanks for your suggestion. We have revised this sentence as:**

**"…the highest mass concentrations of SOA, POA, and BC reached 10 beyond ug/m$^3$"**

*L431: "that some BC-free aerosols were characterized as low volatile and non-negligible fractions of BC-free aerosols dominated within these less volatile aerosol components": could also contain BC with smaller size than minimum threshold*

**Response: Thanks for your suggestion. We have revised this sentence as:**

**"However, during the cleaning period, NF$_V$ was even lower than NF$_{noBC}$, suggesting that some BC-free aerosols were characterized as low volatile, which were likely less volatile organic aerosols (not likely contributed by BC-containing particles with a BC smaller than the SP2 detection limit, because the SF of this type of volatile BC-**

**containing aerosols has an SF lower than 80/200, which is substantially lower than the threshold SF of 0.85 for NF$_V$ calculation). …"**

*L440: "As mentioned above that NFH was also lower 441 than MAF during the moderately*

*polluted period, suggesting periods, there may be a significant fraction of volatile BC-free aerosols": not clear*

**Response: Thanks for your suggestion. We have revised this sentence as:**

**"As mentioned above, NF$_H$ was also lower than MAF during moderately polluted periods, and there may be a significant fraction of volatile BC-free aerosols with hygroscopicity lower than the critical $\kappa$ value of 0.07; however, they were still CCN-active and therefore not fully hydrophobic"**

*L492: "correlation with MFFFOA (-0.45~-0.74) was much weaker compared to MFBBOA. (-0. 10~-0.45).": do not mix up weaker correlation and stronger anticorrelation*

**Response: Thanks for your suggestion. We have revised this sentence as:**

**"However, the anticorrelation with MF$_{FFOA}$ (-0.45~-0.74) was much stronger than MF$_{BBOA}$ (-0.10~-0.45)."**

*L510: "was not contributed by BC-containing aerosols": again, pssibility of BC smaller than lower cut of SP2*

**Response: Thanks for your suggestion. This is not likely, a BC-containing aerosols of 200 nm with BC core smaller than 80 nm was quite aged in the air. This sentence is revised as:**

**"was not contributed by BC-containing aerosols (BC-containing aerosols of 200 nm with BC core smaller than 80 nm which is smaller than the detection limit of SP2 likely to be quite aged in the air, thus not possible to be nearly hydrophobic)"**

**Sincerely Yours**

**Ye Kuang and Li Liu**

**Reference:**

**Kuang, Y., Huang, S., Xue, B., Luo, B., Song, Q., Chen, W., Hu, W., Li, W., Zhao, P., Cai, M., Peng, Y., Qi, J., Li, T., Chen, D., Yue, D., Yuan, B., and Shao, M.: Contrasting effects of secondary organic aerosol formations on organic aerosol hygroscopicity, Atmos. Chem. Phys. Discuss., 2021, 1-27, 10.5194/acp-2021-3, 2021.**

**Hong, J., Äijälä, M., Häme, S. A. K., Hao, L., Duplissy, J., Heikkinen, L. M., Nie, W.,**

**Mikkilä, J., Kulmala, M., Prisle, N. L., Virtanen, A., Ehn, M., Paasonen, P., Worsnop, D. R., Riipinen, I., Petäjä, T., and Kerminen, V.-M.: Estimates of the organic aerosol volatility in a boreal forest using two independent methods, Atmos. Chem. Phys., 17, 4387–4399, https://doi.org/10.5194/acp-17-4387-2017, 2017.**

Reviewer #1:

*General comments:*

*The presented study investigates the mixing state of aerosol particles using different techniques: H- and V-TDMA, CCNC, and SP2 measurements are available in connection with chemical measurements for a 1-month campaign in the North China plain (NCP). This combination provides a useful data set to investigate the aerosol mixing state. However, this combination of measurements gives a lot of information and in this study many parameters were calculated. To understand the relationships and differences between these parameters, they need to be explained and presented in more detail. I believe, the data itself are worth to be published, but the quality of analysis and publication should be improved. The authors use too many abbreviations that disrupt the flow of reading. Some abbreviations are not explained at all, but I think even those that are well known in a particular community should be written out at least once. Furthermore, the statistical analysis is not convincing. Linear correlations are applied to all data points, but in my view, they do not well describe the data in all cases. A critical analysis is needed here to determine which of these statistical results are meaningful. This is my main criticism of this work.*

*The quality of the language is not very good and the manuscript is not easy to read. I*

*recommend a complete check by a native speaker.*

*Thus, the paper needs major revision regarding the statistical analysis. After that, the text should be partly rewritten or at least significantly revised before it can be accepted for publication in ACP.*

**Response: Thanks for your comments. This study provides a first-time intercomparison of aerosol mixing state parameters from the instruments including DMA-SP2, DMA-CCN, HTDMA and VTDMA and offers insights into the interlink among these parameters and potential influencing factors. Aerosol mixing states were usually investigated using one or two of instruments listed above, however, none of them could deliver a full picture of aerosol mixing state variations. The purpose of this paper is to investigate what's the difference among these mixing state parameters and mechanism behind those difference under atmospheric conditions of the campaign, which helps aerosol scientists to understand better aerosol mixing states obtained using different techniques and also design better their future aerosol experiments, because differences among mixing state parameters might deliver important message about physical and chemical properties of primary and secondary aerosols as discussed in this study. Some observed differences can be qualitatively explained based on existing knowledge; however, some differences help us explore possible properties of primary and secondary aerosols and might deliver phenomena that urge explanation in the future. If we go very detail into the variations of each aerosol mixing state parameters, the manuscript would be very long and more difficult for readers because it was very difficult to find readers who understand very well all instruments listed above (DMA-SP2, DMA-CCN, HTDMA, VTDMA) and aerosol primary emissions as well as atmospheric chemistry related with secondary aerosol formations. The authors struggled in writing this manuscript because this is also difficult for us, although the first author has very good records of research using DMA-CCN and HTDMA, and the corresponding author have good records of aerosol physical properties and atmospheric chemistry. We decided to write this paper because we find this might be important and interesting for aerosol community, and also helpful for us and we want to share these insights. Actually, we plan to dig more into these variations based on insights obtained in this research in the our future studies.**

**We agree with the reviewer that some places should be explained more in detail, and therefore more explanations were added before reaching conclusions in some parts as recommended by the reviewer#2.**

**In terms of statistical analysis, we use linear correlations to examine whether the primary emissions or the secondary aerosol formations have significant impacts on the aerosol mixing state parameters, rather than getting linear relationships. Linear**

fittings in the manuscript delivered false impression thus all linear fittings are removed in related figures and leave only correlation coefficients. We discussed this with authors and believe that there are no explicit relations among these parameters, thus correlation test is the only way we could have now based on our limited measurements to explore potential influencing factors as what was done in most previous papers that discuss possible mechanisms behind variations in mixing state parameters.

In terms of writing, the reviwer#2 have raised a lot of comments to help improve the readability, and we revised the manuscript based on comments of both reviewers which is beneficial for non-expert readers. We have also improved this manuscript through Elsevier Language Editing Services:

[Figure]

**Certificate of Elsevier Language Editing Services**

**The following article was edited by Elsevier Language Editing Services:**

Markedly different impacts of primary emissions and secondary aerosol formation on aerosol mixing states revealed by simultaneous measurements of CCNC, V/HTDMA and SP2

**Ordered by:**

Jiangchuan Tao

**Estimated Delivery date:**
2023-08-31
**Order reference:**
ASLESTD1017819

[Figure]

*Comments in detail:*

*There are basic criticisms of the manuscript, so I will go into less detail. Most of my comments are more general, only few of them focus on typos and so on, which does not mean, that these are all minor comments. But I would focus on the detail after the rest is done.*

*Examples for abbreviations, that are never written out:*

*SOA, POA, SSOA, BBOA, CCOA, FFOA, MAF..*

*Some of them are well known, others not. I do not know all of them which makes the reading really difficult. Each abbreviation has to be explained once, but I would suggest to use less abbreviations in general. Even if abbreviations are explained in the technical section and used later without explanations does not really help. I prefer written text, it helps a lot to understand the text much better. In my view it is required to explain those abbreviations, which are not widely known, such as MAF, CCOA, regularly again, also in figure captions.*

**Response: Thanks for your suggestion. We have added a table listing the definition and description of the abbreviations as follow:**

**Table 1. Definition and description of abbreviations.**

| Abbreviation | Full name and/or Definition |
| --- | --- |
| BBOA | Biomass Burning Organic Aerosol

Characterized by obvious m/z 60 (mainly $C_2H_4O_2^+$) and 73 (mainly $C_3H_5O_2^+$), which are two indicators of biomass burning |
| FFOA | Fossil Fuel Organic Aerosol

A mixed factor that comprises traffic emissions and coal combustion, which was characterized by typical hydrocarbon ion series |
| OOA | Oxygenated Organic Aerosol |
| OOA1 and OOA2 | Two OOA factors resolved from the PMF analysis |
| SOA | Secondary Organic Aerosol

Summation of OOA1 and OOA2 |
| POA | Primary Organic Aerosol

Summation of BBOA and FFOA |
| SIA | Secondary Inorganic Aerosols, including nitrate, sulfate, and ammonium |
| $PM_{2.5}$ | Particulate Matter with an aerodynamic diameter <2.5 μm |

| | |
|---|---|
| $PM_1$ | Particulate Matter with an aerodynamic diameter <1 μm |
| $NR\text{-}PM_1$ | Non-refractory $PM_1$ |
| MF | Mass Fraction |
| $D_p$ | Particle diameter after humidification or heating |
| $D_d$ | Particle diameter under dry conditions without humidification or heating |
| κ | Hygroscopicity parameter |
| SS | Supersaturation |
| SPAR | Size-resolved Particle Activation Ratio
Size-dependent CCN activity under a specific SS |
| MAF | Maximum Activation Fraction
An asymptote of the measured SPAR curve at large particle sizes and represents the number fraction of CCNs to total particles |
| $D_a$ | Midpoint activation diameter
Linked to the hygroscopicity of CCNs |
| GF | Growth factor
The ratio between particles with and without humidification and is linked to aerosol hygroscopicity |
| SF | Shrinkage Factor
The ratio between particles with and without heating and is linked to aerosol volatility |
| PDF | Probability Distribution Function |
| $NF_H$ | Number Fraction of Hydrophilic aerosol whose hygroscopicity parameter is >~0.07 |
| $NF_V$ | Number Fraction of Volatile aerosol whose Shrinkage |

| | |
|---|---|
| | **Factor at 200 °C is <0.85** |
| **NF$_{noBC}$** | **Number Fraction of black carbon (BC)-free particles** |
| **NF$_{CBC}$** | **Number Fraction of thickly coated BC particles** |
| **R$_{exBC}$** | **The number concentration ratio of externally mixed BC particles in total BC-containing particles** |
| | **Externally mixed BC particles are defined as identified bare/thinly coated BC-containing particles** |
| **NF$_A$-NF$_B$** **(NF$_{noBC}$-NF$_H$, NF$_V$-NF$_H$, NF$_{noBC}$-NF$_V$, NF$_V$-MAF, NF$_{noBC}$-MAF)** | **The difference between the number fraction of A and B** |

**In addition, we have also added the definition and description of the abbreviations when first introduced in each section and also in the caption of the figures for clarification.**

*Section 2.2:*

*Some more technical details about the aerosol measurements would be helpful. What type of inlet was used? Was the measurement flow dried? How was the relative humidity in the inlet flow?*

**Response: Thanks for your suggestion. We have added technical details about the aerosol measurements in Section 2.1 as follow:**

**"The inlet was switched among three impactors: TSP (Total Suspended Particles), PM$_{2.5}$ (Particulate Matter with an aerodynamic diameter of less than 2.5 μm), and PM$_1$ (Particulate Matter with an aerodynamic diameter of less than 1 μm). Inlet changes among impactors affect dry-state aerosol sampling owing to ambient aerosols are enlarged through aerosol hygroscopic growth or activation. However, the aerosol mixing state and aerosol chemical composition measurements were made on submicron aerosols, and the inlet change almost did not affect those measurements under conditions of RH less than 90%. The sampled aerosol was dried by two parallelly assembled Nafion dryers with a length of 1.2 m. Two Nafion driers was used because of the high RH and sample flow rate (~16 L/min) during the campaign to ensure drying efficiency. In addition, during autumn and winter in the NCP, ambient air temperature (<20 °C and sometimes <0 °C) can be significantly lower than the room temperature**

**(~24 °C). Therefore, this dryer system can maintain the RH of sampled aerosols to below 20%."**

*Were losses in inlet line and sampling systems considered?*

**Response: losses in inlet line and sampling systems are not considered in this study. reasons are listed below: (1) investigated mixing state parameters are represented by number fractions of different diameters which are much less affected by losses in sampling systems compared with absolute umber concentrations; (2) good consistency was achieved between measurements of particle number size distributions (PNSD) and mass concentrations measured by AMS, with the average ratio between volume concentration derived from AMS and rBC measurements and volume concentration derived from PNSD measurements is 0.79 (R=0.97, as shown in the following), which is consistent with previous reports due to that AMS cannot detect aerosol components such as dust (Kuang et al., 2021). This means that almost same aerosol populations were sampled by AMS and instruments of measuring aerosol mixing states.**

[Figure]

**Fig. S3. Comparison between aerosol volume concentration derived from measurements of PNSD and aerosol chemical compositions.**

**The following sentences are added in the revised manuscript.**

**"This study did not consider losses in the inlet line and sampling systems for the following reasons: (1) investigated mixing state parameters are represented by number fractions (NFs) of different diameters, which are much less affected by losses**

in sampling systems compared with absolute number concentrations; and (2) good consistency was achieved between measurements of particle number size distributions (PNSD) and mass concentrations measured by AMS. The average ratio between volume concentration derived from AMS and rBC measurements (densities of compounds are the same as Kuang et al., 2021) and the volume concentration derived from PNSD measurements was 0.79 (R=0.97, as shown in Fig. S3), consistent with previous reports as AMS cannot detect aerosol components, such as dust (Kuang et al., 2021).
"

*Dd is probably the dry diameter?! This is not explained. What means 'dry'? Just not humidified?*

**Response: Yes, Dd is the dry diameter particle, which corresponds to particle diameter under dry conditions (RH<20%) and not humidified. For clarification, we have revised the description of Equation (1) as:**

"… H/V-TDMA can operate in either H- or V-mode, controlled by a three-way solenoid valve. A Nafion humidifier was used in the H-mode to condition the selected dry particles to 90% RH equilibrium. The number-size distribution of humidified particles ($D_p$) was measured using DMA2 and CPC (Model 3772, TSI Inc.). The RH-dependent hygroscopic growth factor (GF) at a specific diameter ($D_d$) was calculated as follows:

$$GF = \frac{D_p(RH)}{D_d} \qquad (1)$$

where $D_p(RH)$ is the size of particles undergoing humidification. …"

*The same diameters Dd and DP are used in the definition for the shrinking factor, what is the meaning here?*

**Response: Thanks for your suggestion. In the definition for the shrinking factor, Dd is the dry diameter particle, which corresponds to particle diameter under dry conditions and not heated while Dp is the particle diameter after heating. we have revised the description of Equation (2) as:**

"The temperature-dependent shrinkage factor (SF), which is the ratio of heated particle size to dry particle size without heating ($D_d$), is defined as:

$$SF = \frac{D_p(T)}{D_d} \qquad\qquad (2)$$

where $D_p(T)$ denotes the particle diameter during heating. …"

*Section 2.3:*

*Parameterization of the SPAR function is not easy to understand without knowing how it looks like. Can the authors give an example?*

**Response: Thanks for your suggestion. As shown in Figure S4, the measured SPAR is generally characterized as a sigmoidal curve (the black line). MAF is the asymptote of the measured SPAR curve at large particle sizes and Da indicates the diameter where SPAR equals the half of the MAF value. The parameter s corresponds to the slope of steep increase of SPAR curves when diameter is close to Da.**

[Figure]

**Fig. S4. Schematic of the parameterization scheme of SPAR curves. The black solid curve and the black crossing are the measured SPAR and fitted SPAR with the parameterization scheme. The red, green and blue dashed lines indicate the fitting parameters of Maximum Activation Fraction (MAF), the midpoint activation diameter (Da) and s, respectively.**

**We have added this figure into the supplement and revised the description of SPAR parameterization scheme as:**

"The SPAR curves were parameterized using a sigmoidal function with three parameters. As shown in Fig. S4, a sigmoidal curve generally characterized the measured SPAR. This parameterization assumes that the aerosol is an external mixture of CCN-active hydrophilic and CCN-inactive hydrophobic particles (Rose et al., 2010). The formula used to parameterize the SPAR ($R_a(D_d)$) for a specific SS is as follows (Rose et al., 2008):

$$R_a(D_d) = \frac{MAF}{2} \left( 1 + erf\left(\frac{D_d - D_a}{\sqrt{2\pi}\sigma}\right) \right) \tag{7}$$

where erf denotes the error function. The Maximum Activation Fraction (MAF) is an asymptote of the measured SPAR curve for large particles, as shown in Fig. S4, representing the fraction of CCNs relative to the total number of particles. $D_a$ is the midpoint activation diameter, is linked to the hygroscopicity of the CCNs, and indicates the diameter where the SPAR equals half of the MAF value. The $\sigma$ is the standard deviation of the cumulative Gaussian distribution function and characterizes the heterogeneity of CCN hygroscopicity. In Fig. S4, the $\sigma$ indicates the slope of the steep increase in the SPAR curves when the diameter is close to Da.   ..."

*People, who are not familiar with the SP2 do not understand the explanation given here. What does the lag time mean? Why is it called lag time? By the way, there are three different ways of writing in the manuscript: lagtime, lag-time and lag time, for consistency one should be chosen. I would take the latter one.*

**Response: Thanks for your suggestion. The lag time is defined as the time difference between the occurrences of the peaks of the incandescence and scattering signals measured by SP2 (Moteki & Kondo, 2007; Sedlacek et al., 2012; Subramanian et al., 2010), and for coated BC particles, the incandescence signals is generally detected later than the scattering signals. As shown in former studies (Zhang et al., 2018; Zhao et al., 2021), the distribution of the lag time for ambient particles exhibits a clear two-mode distribution and this lag time can be used to indicate the coating thickness of the BC-containing aerosols.**

**We have revised lagtime and lag-time to lag time, and have revised this paragraph as:**

"... For the measurement of coated BC particles at SP2, the incandescence signal is generally detected later than the scattering signals and the time difference between the occurrence of the peaks of the incandescence and scattering signals is defined as the lag time (Moteki & Kondo, 2007; Sedlacek et al., 2012; Subramanian et al., 2010). The coating thickness of BC-containing aerosols in the SP2 measurement can be

indicated by the lag time (Moteki and Kondo, 2007; Schwarz et al., 2006; Sedlacek et al., 2012; Subramanian et al., 2010; Metcalf et al., 2012), which has exhibited a clear two-mode distribution in previous studies (Zhang et al., 2018; Zhao et al., 2021). A critical lag time threshold can be used to differentiate between the different types of BC-containing aerosols and calculate the NF of bare and coated BC particles in the total identified aerosols. …"

"

*Section 3:*

*Figure 1 is very complex. Figures c – e also have a color scale on the right hand side, but this is not explained at all.*

**Response: Thanks for your suggestion. We have revised the caption of Figure 1 by adding descriptions of each panel and the color scale as:**

"Figure 1. Overview of the measurements during the campaign: (a) meteorological parameters: wind speed (dots) and relative humidity (RH) (black line), with colors of dots representing wind direction; (b) mass concentrations of aerosol chemical compositions: secondary inorganic aerosols (SIA, red circle), secondary organic aerosols (SOA, green plus), primary organic aerosols (POA, blue x) and black carbon (BC, black dots); (c) Size-resolved Particle Activation Ratio (SPAR) under supersaturation (SS) of 0.08% observed by the DMA-CCNC, with warmer colors corresponding to higher values; (d) Probability Density Function (PDF) of growth factor (GF-PDF) at 200 nm observed by the HTDMA; (e) PDF of shrinkage factor (SF-PDF) at 200 nm and 200 °C observed by the VTDMA; (f) PDF of lag time at 200 nm observed by the DMA-SP2. The blue, red, and green shaded periods represent the three periods with moderate pollution, heavy pollution, and clean conditions, respectively."

*Line 316 ff: '…corresponding fitting parameters, Da…' Da is just one parameter and means probably the mean diameter? How are these diameters obtained? Da should probably be Da?! Other fitting parameters are needed?*

**Response: Da is the midpoint activation diameter, not the mean diameter. Da is not shown in Figure 2 and may be mistaken as particle size Dp. Here we are referring to Da values during the campaign. It can be found that Da value agree with the particle size where SPAR equals about 0.5. We have revised this sentence as:**

"For the three measured SSs, the particle sizes where SPAR equals approximately 0.5

are approximately 90, 120, and 180 nm for the three SSs of 0.08%, 0.14%, and 0.22%, respectively, consistent with the average $D_a$ (see Eq. 7) values of the campaign."

*MAF seems to be another fitting parameter, but what does MAF mean?*

**Response: MAF is Maximum Activation Fraction and an asymptote of the measured SPAR curve at large particle sizes. We have added the description of MAF where MAF is first mentioned in this section as:**

"The NF of CCN-active particles in large-diameter ranges (which varies with SS and, for example, is greater than 200 nm for 0.08%) can be indicated by the gradual increase in the SPAR curves and quantified by the fitting parameter, MAF (see Eq. 7)."

*There appear again lots of abbreviations, such as RexBC. This is explained once, but since it is not common, I had to look it up again and again. I would prefer reading without so many abbreviations.*

**Response: Thanks for your suggestion. We added a table listing the description of these abbreviations as mentioned earlier, and have added the explanations of these abbreviations like RexBC where they are first introduced in each section and in caption of each figure.**

*Line 371, caption figure 4 and others: the word 'composition' is used in the wrong context. The authors mean probably component(s). This appears several times in the text.*

**Response: Thanks for your suggestion. We have revised it accordingly including:**

L47: "BC-containing aerosols emitted from fossil fuel combustion tend to be more externally mixed with other aerosol components …"

L298: "The mass concentrations of different aerosol components …"

L308: "The diurnal variations in mass concentrations of different aerosol chemical components …"

L363: "the mass fractions of aerosol chemical components"

L398 : "the mass fraction of each primary organic aerosol components"

Figure 7, 8 and 9: "the mass fraction of secondary aerosol chemical components"

Figure 4: "mass fraction of aerosol chemical components"

*Figure 4: what are the shaded areas? Standard deviations? Uncertainty? This has to be explained in the figure caption! My question is, if the differences e.g., between the different diameters are significant? For me, the shapes of the curves of NF for different diameters look very similar, in particular if the shaded area represents an uncertainty range.*

**Response: Thanks for your suggestion. The shaded areas indicate the standard deviations, The difference between those of different diameters are not significant, especially for particle diameters larger than 100nm. As the reviewer #2 suggested, we keep the sizes with most concurrent measurements, e.g. 150, 200 and 300 nm and move the rest particle sizes into the supplement. In detail, we have revised Figure 4 and its caption as:**

[Figure]

"**Figure 4. (a–l) Diurnal variations of aerosol mixing state parameters (identified by color and marker) at different particle sizes (50, 150, 200, and 300 nm) during the three periods. The shaded areas indicate the standard deviations. (m–o) Diurnal variations of mass fractions (MFs) of aerosol chemical compositions, including secondary organic aerosols (SOA), biomass burning organic aerosol (BBOA), fossil fuel organic aerosols (FFOA), and inorganic ions including sulfate ($SO_4$), nitrate ($NO_3$), and ammonium ($NH_4$) (identified by color and marker) during the three periods.**"

**We have revised the corresponding description of Figure 4 as:**

"**For particles >100 nm (Fig. 4 and S5), there was a maximum in the afternoon for MAF, $NF_H$, $NF_V$, and $NF_{noBC}$, indicating a peak during this time due to an increase in SA compositions, such as nitrate and SOA, and a decrease in POA and BC.**"

**We have also added Figure S4 into the supplement as:**

[Figure]

"**Fig. S5. (a-f) Diurnal variations of aerosol mixing state parameters (identified by color and marker) at different particle sizes (100 and 250 nm) during the three periods. The shaded areas indicate the standard deviations. (g-i) Diurnal variations of mass fraction of aerosol chemical compositions (identified by color and marker) during the three periods.**"

*In the description of this figure 4 the word 'peak' is frequently used, but I see only slight maxima between different times of the day. This has to be checked and needs to be adapted.*

**Response: Thanks for your suggestion. It should be a maxima between different times of the day and is indicative of a peak in the afternoon. We have revised corresponding descriptions of figure 4 as:**

"For particles >100 nm (Fig. 4 and S5), there was a maximum in the afternoon for MAF, $NF_H$, $NF_V$, and $NF_{noBC}$, indicating a peak during this time due to an increase in SA compositions, such as nitrate and SOA, and a decrease in POA and BC."

"In the clean-air period, there was another maximum at midnight for MAF and $NF_{noBC}$, which may be attributed to the diurnal variations in SA compositions, such as sulfate and SOA, and the decrease in BC and FFOA."

*Line 388: What means 'consistency' here? I simply do not understand it.*

**Response: We are referring to the agreement between different aerosol mixing state parameters and we have revised this sentence as:**

"The agreement between MAF and $NF_V$ was slightly higher than that between MAF and $NF_H$ or between $NF_H$ and $NF_V$ with similar correlation coefficients (~0.65)."

*Figures 5 – 9: linear correlations were fitted here, but the results do not always look convincing. E.g., Figure 7: 2 lowest plots show dots widely distributed and one does not expect a linear correlation. What is the meaning of such a correlation? I strongly suggest to check the quality of these correlations and reduce to number of these plots.*

**Response: Thanks for your suggestion. In former studies on the aerosol mixing state, it is common to investigate the linear correlations between aerosol mixing state parameters as well as aerosol chemical compositions (Reference listed in the introduction like: Zhang et al., 2014; Hong et al., 2017; Kim et al., 2020; Tao et al., 2021). In this study, the correlation analysis is used to examine whether the primary emissions or the secondary aerosol formations have significant impacts on the aerosol mixing state parameters, and for some cases there was no significant influences which also provide insights into investigating variations of mixing state parameters. In order to avoid misunderstanding and highlight the our findings, we have removed the fit lines. In detail, we have revised Figure 5-9 as:**

**Figure 5:**

[Figure]

**Figure 6:**

[Figure]

**Figure 7:**

[Figure]

**Figure 8:**

[Figure]

**Figure 9:**

[Figure]

*Figure 8: the lower plots seem to follow more an exponential growth, does the linear fit makes sense here?*

**Response: As we mentioned in the former response, the correlation analysis of this study is used to qualitatively explore whether the primary emissions or the secondary aerosol formations affect significantly on variations of aerosol mixing state**

parameters, as commonly applied in former studies on the aerosol mixing state (Reference listed in the introduction like: Hong et al., 2017; Kim et al., 2020; Tao et al., 2021). We have removed the fit lines with correlation coefficient (r) less than 0.5 as shown in the former response.

*Figure 9: what means OOA1 and OOA2?*

**Response: OOA1 and OOA2 are two SOA factors from the PMF analysis of organic aerosol. As mentioned in Section 2.1, these two SOA factors were found to display different spectral patterns, correlations with tracers and diurnal variations, suggesting that they resulted from different chemical processing, however, the formation mechanism and possible precursors are yet to be explored in future. For example, OOA1 had higher $CO_2^+/C_2H_3O^+$ (3.9) and O/C (0.91) ratios compared to OOA2 (2.1, 0.78). We have revised this caption as:**

**"Figure 9. The correlation between the difference among the four aerosol mixing state parameters and mass fraction (MF) of secondary aerosol (SA) chemical components during different periods. OOA1 and OOA2 are two secondary organic aerosol (SOA) factors resolved from aerosol mass spectrometer (AMS) measurements using the Positive Matrix Factorization (PMF) technique. Moderately polluted period: blue circle; heavily polluted period: red square; clean period: green pentagon."**

*All figure captions need more text to explain the figure. One should understand the general content of a figure without reading the full text around.*

**Response: Thanks for your suggestion. Besides the captions of Figures 1, 4 and 9 whose revision have been mentioned earlier, the captions of other figures are revised as:**

[revised manuscript text omitted]

*Line 472: exemplarily 'difference between NFV – NFH' means difference between NFV and NFH? This appears several times around this section.*

**Response: Yes, we have revised it as "difference between NFV and NFH (NFV-NFH)". Similar revisions includes:**

**L427-434: "The difference between NF$_{noBC}$ and NF$_H$ (NF$_{noBC}$-NF$_H$) was significantly positively correlated with MF$_{FFOA}$ and MF$_{BBOA}$ (r>0.5), suggesting that a substantial proportion of POA resided in BC-free aerosols and was volatile, but contributed substantially to nearly hydrophobic aerosols; as did the differences between NF$_V$ and NF$_H$ (NF$_V$-NF$_H$). The MFs of BBOA and FFOA were poorly correlated with the differences between the MAF and NF$_V$ (MAF-NF$_V$), MAF and NF$_{noBC}$ (MAF-NF$_{noBC}$), and NF$_V$ and NF$_{noBC}$ (NF$_V$-NF$_{noBC}$) (Fig. S7). The difference between MAF-NF$_H$ was positively correlated with MF$_{BBOA}$, further suggesting that BBOA contributed to nearly hydrophobic aerosols under subsaturated conditions; however, their hygroscopicity was enhanced, and they became CCN-active under supersaturated conditions."**

**L464: "The difference between NF$_{noBC}$ and NF$_H$ (NF$_{noBC}$-NF$_H$) showed a strong negative correlation with MF$_{NH4}$ and MF$_{NO3}$ (mainly -0.6), as did the differences between NF$_V$ and NF$_H$ (NF$_V$-NF$_H$)."**

**L483: "The difference between NF$_{noBC}$ and NF$_V$ (NF$_{noBC}$-NF$_V$) was negatively correlated with MF$_{NO3}$, which is consistent with the semi-volatile nature of nitrate."**

**L555: "… the two resolved SOA factors exhibited different impacts on the differences between NF$_V$ and NF$_H$ (NF$_V$-NF$_H$), …"**

**Figure 10: "… (d and e) The variations of the difference between NF$_V$ and NF$_{noBC}$ (NF$_V$-NF$_{noBC}$, blue large circle) and the difference between NF$_V$ and NF$_{noBC}$+NF$_{CBC}$ (NF$_V$-(NF$_{noBC}$+NF$_{CBC}$), yellow small circle)…"**

*Minor comments/ typos:*

*Comment: I did not look explicitly for all typos, because I think, several parts need to be rewritten und after that it should be read again carefully.*

*Line 293: PA means probably POA*

*Line 319: in large diameter range*

*Line 334:a dot after 'size' is missing*

*Line 439: 'are presented' should be 'is presented'*

**Response: Thanks for your suggestion and we have revised them accordingly. We have also checked the manuscript again and improved this manuscript by Elsevier Language Editing Services as mentioned earlier.**


**Response: Thanks for your suggestion. The full names of BBOA and FFOA are Biomass Burning Organic Aerosol and Fossil Fuel Organic Aerosol, respectively. We have added a table listing the definition and description of the abbreviations as follow:**

**Table 1. Definition and description of abbreviations.**

| Abbreviation | Full name and/or Definition |
|---|---|
| BBOA | **Biomass Burning Organic Aerosol**
Characterized by obvious m/z 60 (mainly $C_2H_4O_2^+$) and 73 (mainly $C_3H_5O_2^+$), which are two indicators of biomass burning |
| FFOA | **Fossil Fuel Organic Aerosol**
A mixed factor that comprises traffic emissions and coal combustion, which was characterized by typical hydrocarbon ion series |
| OOA | **Oxygenated Organic Aerosol** |
| OOA1 and OOA2 | Two OOA factors resolved from the PMF analysis |
| SOA | **Secondary Organic Aerosol**
Summation of OOA1 and OOA2 |
| POA | **Primary Organic Aerosol**
Summation of BBOA and FFOA |
| SIA | **Secondary Inorganic Aerosols, including nitrate, sulfate, and ammonium** |
| $PM_{2.5}$ | **Particulate Matter with an aerodynamic diameter <2.5 µm** |

| | |
|---|---|
| $PM_1$ | Particulate Matter with an aerodynamic diameter <1 μm |
| NR-$PM_1$ | Non-refractory $PM_1$ |
| MF | Mass Fraction |
| $D_p$ | Particle diameter after humidification or heating |
| $D_d$ | Particle diameter under dry conditions without humidification or heating |
| κ | Hygroscopicity parameter |
| SS | Supersaturation |
| SPAR | Size-resolved Particle Activation Ratio
Size-dependent CCN activity under a specific SS |
| MAF | Maximum Activation Fraction
An asymptote of the measured SPAR curve at large particle sizes and represents the number fraction of CCNs to total particles |
| $D_a$ | Midpoint activation diameter
Linked to the hygroscopicity of CCNs |
| GF | Growth factor
The ratio between particles with and without humidification and is linked to aerosol hygroscopicity |
| SF | Shrinkage Factor
The ratio between particles with and without heating and is linked to aerosol volatility |
| PDF | Probability Distribution Function |
| $NF_H$ | Number Fraction of Hydrophilic aerosol whose hygroscopicity parameter is >~0.07 |
| $NF_V$ | Number Fraction of Volatile aerosol whose Shrinkage Factor at 200 °C is <0.85 |

| | |
|---|---|
| $NF_{noBC}$ | Number Fraction of black carbon (BC)-free particles |
| $NF_{CBC}$ | Number Fraction of thickly coated BC particles |
| $R_{exBC}$ | The number concentration ratio of externally mixed BC particles in total BC-containing particles

Externally mixed BC particles are defined as identified bare/thinly coated BC-containing particles |

In addition, we have also added the definition and description of the abbreviations when first introduced in each section and also in the caption of the figures for clarification.

*5) Line 177, what do you mean by "different chemical process" and could you give more details?*

**Response: Secondary organic aerosol formations originated from volatile organic compounds precursors could be formed in differ formation pathways such as aqueous phase reactions, heterogeneous reactions or gas phase reactions and also might be oxidized under different conditions, for example oxidized under different nitrogen oxide conditions with different oxidation capacity and oxidants. The following sentences is added in the revised manuscript.**

**"Secondary organic aerosol formation from volatile organic compound precursors could occur in different formation pathways, such as aqueous-phase, heterogeneous, or gas-phase reactions. It might also be oxidized under different conditions, such as oxidation under different nitrogen oxide conditions with different oxidation capacities and oxidants. The two resolved OOA factors displayed different spectral patterns, correlations with tracers, and diurnal variations, suggesting that they resulted from different chemical processes. However, their formation mechanisms remain to be explored in future studies. In general, the OOA factor 1 (OOA1) has higher $CO_2^+/C_2H_3O^+$ (3.9) and O/C (0.91) ratios than OOA factor 2 (OOA2) with 2.1 and 0.78, respectively."**

*6) Line 187, why do you choose these three supersaturations for CCN measurements?*

**Response: As particle size is the most important parameter in determining CCN activity (Duesk et al., 2006), measurement of CCN activity can indicate particle hygroscopicity in different particle size ranges. In general, the three supersaturations indicate the particle hygroscopicity in particle size range from 100 nm to 200 nm.**

In order to perform intercomparisons among instruments, three supersaturations (SSs) of 0.08%, 0.14% and 0.22% were applied in a single cycle of about 15 minutes. CCN measurement under these three SSs reveals mainly CCN activity of aerosols reside in accumulation mode aerosol with diameter range of about 100-200 nm, which are close to diameters of HV-TDMA measurements, and higher SSs would reveal CCN activity of smaller aerosol particles (<100 nm) where DMA-SP2 measurement is not available:

"To compare the instruments, three supersaturations (SSs) of 0.08%, 0.14%, and 0.22% were applied in a single cycle of approximately 15 min. CCN measurements under these three SSs revealed that the CCN activity of aerosols resides in the accumulation mode with an aerosol diameter range of approximately 100–200 nm, which is close to the diameters of the HV-TDMA measurements. Higher SSs would reveal CCN activities of smaller aerosol particles (<100 nm), where the DMA-SP2 measurement is unavailable."

*7) Line 211, the maximum temperature you chose is 200 degree Celsius, why do you choose this threshold?*

**Response: The HV-TDMA were scanning at different temperatures and diameters for the HV-TDMA system, to ensure the time duration of one full cycle is about 3 h, we limited the number of temperatures and diameters. Most importantly, results of previous studies in the North China Plain have shown that 200 degree Celsius is enough for removing most non-refractory aerosols (>80%) (Xu et al., 2019).**

*8) Line 225-229, regarding the chosen size for SP2, which system was conducted for this study, with or without thermodenuder-bypass? Since you are expected to compare with HTDMA and VTDMA, why not choose the same sizes to measure for the three instruments?*

**Response: The DMA-SP2 system was conducted both with and without thermodenuder-bypass depends on time, and detailed periods are added in the revised manuscript. Compared to HTDMA and VTDMA, more particle sizes are selected in the measurement DMA-SP2 system for obtaining more information of BC mass concentration and mixing states at different particle diameters for other scientific purposes. Because the time needed for a single particle size measurement of DMA-SP2 system is much shorter than that of HTDMA and VTDMA, and one full cycle for H/VTDMA lasts 3 hours. We have added corresponding description into the manuscript as:**

"The DMA-SP2 setup was able to measure the mixing states of aerosols with diameters (detection limit of approximately 80 nm based on the calibration) of 100, 120, 160, 200, 235, 270, 300, 335, 370, 400, 435, 470, 500, 535, 570, 600, 635, 670, and 700 nm within 20 min when it was not placed after an denuder-bypass switch system (the 13th to the 24th of October, 09:00 am of the 5th of November to 09:00 am of the 8th of November). However, it only measured mixing states at diameters of 120, 160, 200, 250, 300, 400, and 500 nm when it was placed after a thermodenuder-bypass switch system (11:00 am of the 24th of October to 08:00 am of the 5th of November, and 09:00 am of the 8th of November to 06:00 pm of the 17th of November). Because the HTDMA and VTDMA measurements were conducted solely by a single H/VTDMA system operating in different modes, the time needed for a single particle size measurement of HTDMA and VTDMA was much longer than that of the DMA-SP2 system. Thus, more particle sizes were selected in the DMA-SP2 system for acquiring the BC mass concentration and mixing state at larger diameters than HTDMA and VTDMA."

9) *Line 235, does the flow rate influence the measurements and by how much?*

Response: This change satisfied the flowrate requirements of this instrument (0.03 to 0.18 L/min), and 0.12 L/min was typically used. The flow rate change does not affect the measurements when aerosol number concentration is not small. Actually, at the very beginning, 0.1 L/min (less than the typical one 0.12L/min) was usually used because the NCP is generally polluted, and higher flow rate would produce larger data storage, however, does not affect the statistical results. We change to 0.12 L/min is because that we realized that we scan up to 700 nm using the DMA-SP2 which is different with previous studies where aerosol number concentration is much smaller and a larger sample flow rate should be better.

10) *Section 2.3.1, the MAF is a fitting parameter from eq.7, what is the physical meaning of this parameter? Is it the maximum activation fraction?*

Response: Yes, it's the maximum activation fraction and we have revised the corresponding description as:

".. The Maximum Activation Fraction (MAF) is an asymptote of the measured SPAR curve for large particles, as shown in Fig. S4, representing the fraction of CCNs relative

**To be noted, a schematic of the SPAR parameterization scheme and the corresponding fitting parameters is added into the supplement for clarification as:**

[Figure]

Fig. S4. Schematic of the parameterization scheme of SPAR curves. The black solid curve and the black crossing are the measured SPAR and fitted SPAR with the parameterization scheme. The red, green and blue dashed lines indicate the fitting parameters of Maximum Activation Fraction (MAF), the midpoint activation diameter (Da) and s, respectively.

*11) Line 267, add sizes for the GF "The GFC for the four measured particle sizes were 1.1, 1.15, 1.175 and 1.2".*

**Response: The GFC for particle size of 50, 100, 150 and 200 nm are 1.1, 1.15, 1.175 and 1.2, respectively. We have revised it as:**

"The $GF_C$ for the four measured particle sizes of 50, 100, 150, and 200 nm were 1.1, 1.15, 1.175, and 1.2, respectively,"

12) Section 2.3.3. Here you use the lag time between the peak of the scattering signal and the incandescence signal to classify the bare and coated BC. Is it related to the BC-coating mass ratio? The mass ratio is more commonly used and intuitive to understand.

**Response: Thanks for your suggestion. It is related to the coating thickness of the BC-**

containing aerosols. The BC-coating mass ratio cannot be directly obtained in SP2 measurement, due to the lack in the accurate density and shape of the BC core. In addition, the lag time is positively correlated to the coating thickness, but their relation cannot be directly quantified and also calibrated. Thus, a critical value of lag time rather than coating thickness or coating mass ratio is used to classify the bare and coated BC. We have revised this sentence as:

"In this study, a two-mode distribution of the lag time ($\Delta t$) was observed, and a critical value of 0.8 µs was used to classify the BC-containing particles into thinly coated (or bare) BC ($\Delta t < 0.8$ µs) and thickly coated BC ($\Delta t \geq 0.8$ µs)."

13) Line 297-299, please give exact values of PM mass for the heavily polluted and clean period.

Response: Thanks for your suggestion. Non-refractory $PM_1$ mass for the heavily polluted and clean period are 49.5±22.5 and 5.1±3.3 mg/m³, respectively. We have revised this sentence as:

"The mass concentrations of different aerosol components increased significantly from the 23rd of October to the 6th of November (heavily polluted period with an average non-refractory $PM_1$ mass concentration of 49.5±22.5 µg/m³) and decreased too much lower levels after the 6th of November (clean period with a non-refractory $PM_1$ mass concentration of 5.1±3.3 µg/m³)."

14) Line 315-316. "At lower SSs, the rapid increases in SPAR curves occur at larger particle sizes and the maximum AR of SPAR curves becomes smaller". Please briefly explain why.

Response: Thanks for your suggestion. For lower SSs, particle size need for CCN activation is larger, thus SPAR curve start to increase from 0 at larger particle size. Because only SPAR in particle size lower than 300 nm is presented and there was less particle to be CCN active under low SSs, the maximum AR of SPAR curves becomes smaller under low SSs. We have revised this sentence as:

"At lower SSs, the particle size required for CCN activation was larger; thus, rapid increases in the SPAR curves occurred at larger particle sizes. In addition, the maximum AR of the SPAR curves decreases as fewer particles are CCN-active under low SSs."

15) Line 318, add SS for the "increases in SPAR curves, are approximately 90 nm, 120 nm and 180 nm"

**Response: Thanks for your suggestion. We have revised it as:**

**"For the three measured SSs, the particle sizes where SPAR equals approximately 0.5 are approximately 90, 120, and 180 nm for the three SSs of 0.08%, 0.14%, and 0.22%, respectively,"**

*16) Fig 2. Are bars representing the standard deviation of the campaign average?*

**Response: Yes, and we have added corresponding description in the end of the caption of Figure 2 as:**

**"The shaded areas indicate the standard deviations."**

*17) Line 331-333, "In general, the size dependence of MAF, NFH, NFV and NFnoBC were similar to one another, suggesting they were dominated by the same particle group, namely BC-free particles". I think this statement is not well supported, I would suggest weakening it or proving it with more evidence. For example, thickly coated BC particles can be very CCN-activate, hydrophilic and volatile, if mostly contain SIA.*

**Response: Thanks for your suggestion and we fully agree. We have revised this sentence as:**

**"In general, the size-dependent characteristics of MAF, $NF_H$, $NF_V$, and $NF_{noBC}$ were similar, suggesting that they were likely dominated by the same particle group, namely BC-free particles. "**

*18) Line 335, please provide exact values of the fraction of BC-containing particles and the applied diameter range, because the terms "higher" or "larger" are not accurate. Check out similar issues for the remaining manuscript too.*

**Response: Thanks for your suggestion. We have revised this sentence as:**

**"This suggests that primary emissions tend to have higher fractions of BC-containing particles in larger diameter ranges; for example, the fraction of BC-containing particles increases from ~0.1 to ~0.4 as the particle size increases from 200 to 500 nm during the cleaning period."**

**We have also checked the manuscript and revised the following:**

**L333: "This particle group had the highest fraction (>0.7) during the heavily polluted period and the lowest fraction (down to 0.5) during the clean period, with the fraction decreasing with increasing particle size."**

**L347:** "… when the nitrate fraction was the highest (~30%). The SOA fraction was the lowest (~7%) among all three periods,…"

**L354-355:** "However, they were larger than the NF$_H$ during the moderately and heavily polluted periods (by ~0.2) when the POA/SOA fractions were higher (~40% vs. ~35%)."

*19) Line 342, what do you mean by "the more efficient secondary aerosol formation", increase by secondary aerosol mass or particle size?*

**Response: Here we are referring to that the formation rate of secondary aerosol mass is more efficient on larger particle, and we have revised this sentence as:**

"… while the decrease in R$_{exBC}$ with increasing particle diameter in the polluted period confirmed that SA formation is more efficient for particles with larger diameters."

*20) Line 356-357, what is the kappa value for hydrophobic mode aerosol?*

**Response: The kappa value for hydrophobic mode aerosol is less than 0.07 and we have revised this sentence as:**

"The critical k of hydrophilic mode aerosols was 0.07, suggesting that a higher fraction of aerosols had k below 0.07 (i.e., hydrophobic mode aerosols in this study) during the moderately polluted period."

*21) Line 361, how do you get this statement with "lower than 0.07 but still CCN active"? please explain in detail.*

**Response: In this part we are referring to that the difference among MAF, NFV, NFH and NFnoBC and we found that NFH is significantly smaller than the other three parameters. This may indicate that a portion of particles to be CCN active but not hydrophilic, i.e. with $\kappa$ lower than 0.07. We have revised this sentence as:**

"As mentioned above, NF$_H$ was also lower than MAF during moderately polluted periods, and there may be a significant fraction of volatile BC-free aerosols with hygroscopicity lower than the critical $\kappa$ value of 0.07; however, they were still CCN-active and therefore not fully hydrophobic."

*22) Fig4, I would suggest simplifying the plot and keeping the sizes with most concurrent measurements, e.g. 150, 200 and 300 nm. Put other sizes to the supplement.*

**Response: Thanks for your suggestion. We have revised Figure 4 and its caption as:**

[Figure]

"**Figure 4. (a–l) Diurnal variations of aerosol mixing state parameters (identified by color and marker) at different particle sizes (50, 150, 200, and 300 nm) during the three periods. The shaded areas indicate the standard deviations. (m–o) Diurnal variations of mass fractions (MFs) of aerosol chemical compositions, including secondary organic aerosols (SOA), biomass burning organic aerosol (BBOA), fossil fuel organic aerosols (FFOA), and inorganic ions including sulfate ($SO_4$), nitrate ($NO_3$), and ammonium ($NH_4$) (identified by color and marker) during the three periods.**"

**We have revised the corresponding description of Figure 4 as:**

"**For particles >100 nm (Fig. 4 and S5), there was a maximum in the afternoon for MAF, $NF_H$, $NF_V$, and $NF_{noBC}$, indicating a peak during this time due to an increase in SA**

compositions, such as nitrate and SOA, and a decrease in POA and BC."

We have also added Figure S4 into the supplement as:

[Figure]

"Fig. S5. (a-l) Diurnal variations of aerosol mixing state parameters (identified by color and marker) at different particle sizes (50, 150, 200 and 300 nm) during the three periods. The shaded areas indicate the standard deviations. (m-o) Diurnal variations of mass fractions of aerosol chemical compositions (identified by color and marker) during the three periods."

*Line 362- 366, the diurnal variations should be described more explicitly as the pattern of RexBC is clearly different from the other three mixing state parameters and explain why.*

**Response: Thanks for your suggestion. We have added more discussion in the end of this paragraph as:**

"$R_{exBC}$ tended to be lower during the daytime, and its diurnal variation was more significant for larger particle sizes. In general, the diurnal variations for $R_{exBC}$ were opposite to those of $NF_{noBC}$ and agreed better with those of the primary aerosol MFs. This is because BC particles originate from primary emissions and are mainly mixed externally. After aging in the atmosphere, BC particles can be coated by SAs, resulting in more coated BC particles and fewer externally mixed BC particles. As SAs tend to form on larger particles, the diurnal variations in SA formation may significantly affect the RexBC of larger particle sizes."

*23) Line 384, table S1 is quite interesting for readers thus I suggest putting it or making a correlation plot into the main context.*

**Response: Thanks for your suggestion. We agree that useful information is contained in this table, we also struggled before we decided to put it in the supplement. We want readers focus more on key parts of those intercomparison results, however, it was also available in the supplement in case that readers want to know all scenarios.**

*24) Line 385, why do you choose these three sizes? The critical size for the setting SS?*

**Response: As shown in Figure 2, the particle size where the rapid increases in SPAR curves are approximately 90 nm, 120 nm and 180 nm for the three SSs of 0.22%, 0.14% and 0.08%, respectively. And the diameter range of rapid increases in SPAR curves are determined by aerosol hygroscopicity in this particle size ranges. Thus, the three particle sizes of 100 nm, 150 nm and 200 nm are chosen in comparison to the MAF at the three SSs of 0.22%, 0.14% and 0.08%, respectively. We have revised this sentence as:**

**"Note that the MAF at SSs of 0.08%, 0.14%, and 0.22% were used for comparison at 200, 150, and 100 nm particle sizes. This is because the diameter range of rapid increases in the SPAR curves is determined by aerosol hygroscopicity in this particle size range. The midpoints of the rapidly increasing diameter ranges of the SPAR curves at SSs of 0.08%, 0.14%, and 0.22% were approximately 180 nm, 120 nm, and 90 nm, respectively (as shown in Fig. 2). "**

*25) Line 386. A classification of the correlation should be clarified, such as the r range for the weak, moderate, and strong correlation.*

**Response: Thanks for your suggestion. The value range of correlation coefficient for weak, moderate and strong was generally less than 0.3, from 0.3 to 0.5 and larger than 0.5. We have added detailed value of correlation coefficient into the manuscript including:**

**L386: "… moderate correlations (r=~0.5) .."**

**L392: "… the correlation became weaker (r=~0.4), …"**

**L421: "…, and weak correlations (r<0.3) …"**

**L440: "… a strong positive correlation with $MF_{SO4}$ (r>0.5). …"**

**L445: "… the weaker correlations with SOA (r~0.3) seen in Fig. 8."**

**L454**: "…, the strong positive correlations between NFV and secondary aerosol formations (r=~0.6) …"

**L457**: "… strong positive correlations (r=~0.5) …"

**L465**: "… a strong negative correlation with $MF_{NH4}$ and $MF_{NO3}$ (mainly -0.6) …"

*26) Line 392, what do you mean by saying "..while the degree was the least for the correlation.."?*

**Response: We are referring to that the degree of the reduction of correlation was the least for the correlation between MAF and NFV, and we have revised this sentence as:**

"For smaller particle sizes, the correlation became weaker (r=~0.4), whereas the degree of reduction was the lowest for the correlation between MAF and NF$_V$."

*27) Fig. 5, what is the r in the plot? It would be more intuitive to use the same marker to represent different periods in the plot.*

**Response: The variable r represent the correlation coefficient and we have added corresponding description into the caption as "with r representing the correlation coefficient." At the request of the Copernicus Publications, the marker used to present different periods are set to be different in order to making this figure friendly to readers with color vision deficiencies.**

*28) A summary table (or correlation matrix plot) of r in Fig5-7 will be helpful for readers to better understand the interlink between mixing state parameters and chemical composition.*

**Response: Thanks for your suggestion and the correlation between mixing state parameters and aerosol chemical composition as well as detailed correlation during different pollution periods were summarized in Figures S6 and S8 (Figures S5 and S6 in old version). We have added introduction of these figures into the manuscript as:**

**In the end of Section 3.3:** "The correlations between the mixing-state parameters and primary aerosol composition during the campaign and different pollution periods are summarized in Fig. S7."

**In the end of last second paragraph of Section 3.4:** "The correlations between the mixing state parameters and SA composition during the campaign and different pollution periods are summarized in Fig. S9."

*29) Line 400, please give values to the sentence "correlation with MFFFOA was much weaker compared to MFBBOA".*

**Response: Thanks for your suggestion. We have revised this sentence as:**

"However, the anticorrelation with MF$_{FFOA}$ (-0.45~-0.74) was much stronger than MF$_{BBOA}$ (-0.10~-0.45)."

*30) Fig.7. Which size of data do you use?*

**Response: The size is 200 nm and we have added corresponding description into the caption of Figure 7 as "The impact of primary emissions on the differences among the four aerosol mixing state parameters at a particle size of 200 nm was analyzed and is shown in Fig. 7."**

*31) Line 428, please introduce what the difference (NFnoBC-NFH and NFV-NFH) represents first before jumping to the results.*

**Response: Thanks for your suggestion. We have added the definition of these abbreviations as:**

**L427-434:** "The difference between NF$_{noBC}$ and NF$_H$ (NF$_{noBC}$-NF$_H$) was significantly positively correlated with MF$_{FFOA}$ and MF$_{BBOA}$ (r>0.5), suggesting that a substantial proportion of POA resided in BC-free aerosols and was volatile, but contributed substantially to nearly hydrophobic aerosols; as did the differences between NF$_V$ and NF$_H$ (NF$_V$-NF$_H$). The MFs of BBOA and FFOA were poorly correlated with the differences between the MAF and NF$_V$ (MAF-NF$_V$), MAF and NF$_{noBC}$ (MAF-NF$_{noBC}$), and NF$_V$ and NF$_{noBC}$ (NF$_V$-NF$_{noBC}$) (Fig. S7). The difference between MAF-NF$_H$ was positively correlated with MF$_{BBOA}$, further suggesting that BBOA contributed to nearly hydrophobic aerosols under subsaturated conditions; however, their hygroscopicity was enhanced, and they became CCN-active under supersaturated conditions."

**L464:** "The difference between NF$_{noBC}$ and NF$_H$ (NF$_{noBC}$-NF$_H$) was significantly positively correlated with MF$_{FFOA}$ and MF$_{BBOA}$ (r>0.5), suggesting that a substantial proportion of POA resided in BC-free aerosols and was volatile, but contributed substantially to nearly hydrophobic aerosols; as did the differences between NF$_V$ and NF$_H$ (NF$_V$-NF$_H$)."

**L483:** "The difference between NF$_{noBC}$ and NF$_V$ (NF$_{noBC}$-NF$_V$) was negatively correlated with MF$_{NO3}$, which is consistent with the semi-volatile nature of nitrate."

**L555:** "… the two resolved SOA factors exhibited different impacts on the differences between NF$_V$ and NF$_H$ (NF$_V$-NF$_H$), …"

**Figure 10:** "… (d and e) The variations of the difference between $NF_V$ and $NF_{noBC}$ ($NF_V$-$NF_{noBC}$, blue large circle) and the difference between $NF_V$ and $NF_{noBC}$+$NF_{CBC}$ ($NF_V$-($NF_{noBC}$+$NF_{CBC}$), yellow small circle) with the mass concentration of SA at particle size 200 nm (d) and 300 nm (e) $NF_{CBC}$: Number Fraction of thickly coated black carbon (BC) particles. ."

*32) Line 438, why do you choose 200nm?*

**Response: This is mainly because we focus on the comparison of the four aerosol mixing state as well as their relationship with aerosol chemical compositions, but only in 200 nm were all the four aerosol mixing state parameters measured. We have added corresponding description into the manuscript as:**

"The analysis is conducted at only 200 nm, where all four aerosol mixing state parameters were measured to compare the four aerosol mixing state parameters and their relationships with aerosol chemical compositions simultaneously."

*33) Line 459-462, out of curiosity, does the transport of ageing aerosols play a role in the increasing fraction of non-BC particles?*

**Response: The reviewer raised a very interesting topic. Indeed, the transport of aging aerosols could play a role in variations in fraction of non-BC particles, for example, during the clean period. However, for periods of the moderately to heavily polluted, the wind speed generally lower than 2 m/s, with strong local emissions (represented quick increase of rBC and POA in the afternoon) of secondary aerosols formations (represented by quick nitrate and SOA formations), the transport of aging aerosols should play a negligible role.**

---

## Author Response (AR3)

**Editor:**

*As also noted by the referees the paper has improved substantially, however, there is still more to be done (hence major revison with additional review by the referees)*

**Response: Thanks for your comments. Thank you for your feedback and for taking the time to review our manuscript, we really appreciate your insightful comments**

**Suggestions and comments are addressed point-by-point and corresponding responses are listed below.**

*Referee 2 gives some hints, please consider them with the exception mentioned below; further comments are given in the following:*

*- I agree that the paper lists observations on similarities and differences in the various fractions, however, more needs to be said what is new, what is expected, and what is in contrast to expectations. You should also give recommendations for future research with such a setup, and list the advantages and disadvantages.*

**Response: Thanks for your comments. We have added some discussions in sections 3.3 and 3.4 to better claim what is new and what is expected as follow:**

**L397: We have added a sentence as:**

**"This may be due to the high relative humidity during the pollution period, and the formation of SA occurs mainly in the aqueous phase, which contributes to the formation of particles with larger diameters (accumulation-mode and droplet-mode particles, Kuang et al., 2020)."**

**L503: We have added a paragraph as:**

[revised manuscript text omitted]

- Then, most importantly, there are still a lot of unclear sentences (e.g. L 477-481), or unsupported statements (e.g. L 483), which make it impossible to accept the paper at the current stage.

**Response: Thanks for your comments. We have revised these unclear sentences and responded to the corresponding comments later. In addition, we have checked the manuscript and revised the unclear sentences and unsupported statements as follow:**

**L166: "The flow rate is carefully adjusted in the inlet in order to ensure accurate aerosol particle size cutoff."**

**L269: "… when it was placed after a thermodenuder-bypass switch system (during the following time periods: 11:00 am of the 24th of October …"**

**L272-273: "Thus, for the same measurement cycle (2h), more particle sizes were selected in the DMA-SP2 system to acquire the BC mass concentration and mixing state at larger diameters than HTDMA and VTDMA."**

*- Please check the attached annotated manuscript; these comments are just examples and not meant to be complete.*

**Response: Thanks for your comments. We have revised the manuscript based on the comments in the attached annotated manuscript.**

*- In contrast what Referee 2 suggested you are fine with using L as symbol for liter; according to a recent decision by the ACP Editorial board the symbol L will be used for liter in all future publications, to avoid confusion with the capital I.*

**Response: Thanks for your kind reminding.**

*- Concerning the number of figures I leave this up to the authors: Please decide what you consider appropriate number of figures, based on the suggestion of Referee 2 in terms of readability of the paper.*

**Response: Thanks for your comments. After careful consideration, we have decided to retain all the figures in the manuscript. While we acknowledge that some figures may appear similar at first glance, each figure conveys distinct messages and contributes to the comprehensive understanding of our research findings. We believe that retaining all figures is crucial in presenting a thorough analysis and ensuring that all aspects of our study are adequately represented.**

*1. Abstract: The journal now has new guidelines, see https://www.atmospheric-chemistry-and-physics.net/policies/guidelines_for_authors.html.*

*Specifically, the abstract should have less than 250 words (but also check the other items of the guidelines)*

**Response: Thanks for your kind reminder, we have shortened the abstract to less than 250 words as follow:**

"This study compares aerosol mixing state parameters obtained via simultaneous measurements using DMA-CCNC, H/V-TDMA, and DMA-SP2, shedding light on the impacts of primary aerosol emissions and secondary aerosol (SA) formation. The analysis reveals significant variations in mixing-state parameters among different techniques, with V-TDMA and DMA-SP2 indicating that non-volatile particles mainly stem from BC-containing aerosols, while a substantial proportion of nearly hydrophobic aerosols originates from fossil fuel combustion and biomass burning emissions. Synthesizing the results, some nearly hydrophobic BC-free particles were found to be CCN-inactive under supersaturated conditions, likely from fossil fuel combustion emissions, while others were CCN-active, linked to biomass burning emissions. Moreover, BC-containing aerosols emitted from fossil fuel combustion exhibit more external mixing with other aerosol components compared to those from biomass burning. Secondary nitrate and organic aerosol formation significantly affect aerosol mixing states, enhancing aerosol hygroscopicity and volatility while reducing heterogeneity among techniques. The study also highlights distinct physical properties of two resolved secondary organic aerosol factors, hinting at formation through different mechanisms. These findings underscore the importance of comparing aerosol mixing states from different techniques as a tool in understanding aerosol physical properties from different sources and their responses to SA formation, as well as aiding in the exploration of SA formation mechanisms."

*2. English:*

*L187: "… by obvious m/z 60 (mainly $C_2H_4O_2^+$) and 73 (mainly $C_3H_5O_2^+$)":*

*L276: "The BC-containing particles passing through the laser beam **became** incandescent by absorbing radiation."*

*L498: "The difference between MAF-NFH …"*

**Response: Thanks for your comments. We have revised these sentences as:**

**"by abundant fragments of m/z 60 (mainly C2H4O2+) and 73 (mainly C3H5O2+)"**

**"The BC-containing particles passing through the laser beam become incandescent by absorbing radiation."**

**"The difference between MAF and NF$_H$ …"**

*3. L265: "… (the 13th to the 24th of October, 09:00 am of the 5th of November to 09:00 am of the 8th of November)": Not clear*

**Response: Thanks for your comments. We are referring to the time periods that DMA-SP2 was not placed after an denuder-bypass switch system and we have revised this sentence as:**

**"… when it was not placed after an denuder-bypass switch system (during the following time periods: the 13th to the 24th of October, 09:00 am of the 5th of November to 09:00 am of the 8th of November)."**

*4. L348: "In contrast, the highest mass concentrations of SOA, POA, and BC reached 10 μg/m3." : still not clear: each component beyond 10 or the sum of all three together beyond 10?*

**Response: Thanks for your comments. It should each component of SOA, POA, and BC reached 10 μg/m3 and we have revised this sentence as:**

**"the highest mass concentrations of SOA, POA, and BC all reached 10 μg/m3"**

*5. L390: "… during the clean period": clean period, not cleaning period. Multiple instances*

**Response: Thanks for your comments. We have revised them accordingly.**

*6. L393: "RexBC": define at first instance*

**Response: Thanks for your comments. We have revised this sentence as:**

**"As for RexBC , which is defined as the number concentration ratio of externally mixed BC particles in total BC-containing particles, the small …"**

*7. L407: "… because the SF of this type of volatile BC-containing aerosols has an SF lower than **80/200** …": 80/200 not defined (it's probably both nm, but this needs to be mentioned)*

**Response: Thanks for your comments. We have revised this sentence as:**

**"because this type of volatile BC-containing particles aerosols has an SF lower than 0.4 (=80nm/200nm),"**

*8. L452-453: "The agreement between MAF and NFV was slightly higher than that between MAF and NFH or between NFH and NFV with similar correlation coefficients (~0.65).": this sentence is not clear, there also seems to be a discrepancy to the sentence before (unless I got it wrong, but then it is indeed not clear)*

**Response: Thanks for your comments. We have revised this sentence as:**

**"The agreement between MAF and NF$_V$ was slightly higher than that between MAF and NFH or between NF$_H$ and NF$_V$. In detail, compared to the other two, the agreement between MAF and NF$_V$ has a similar correlation coefficients (r~0.65) and a smaller systematic differences (slope and intercept were much closer to 1 and 0, respectively)."**

*9. L454: "However, smaller systematic differences (slope and intercept) were much closer to 1 and 0, respectively.": not clear*

**Response: Thanks for your comments. This sentence is wrong and we have revised as mentioned in the response of the comment before.**

*10. L457: "… whereas the degree of reduction was the lowest …": what does this mean: still highest correlation between MAF and NFV ?*

**Response: Yes, the correlation between MAF and NFV is the highest and r is about .We have revised this sentence as:**

**"whereas the degree of reduction was the lowest for the correlation between MAF and NF$_V$ (r~0.53)."**

*11. L476-482: "At the same MFFFOA, NFH was obviously lower than NFnoBC (NFH and NFnoBC were larger and smaller than 0.7 when MFFFOA was larger than 0.1), demonstrating that a substantial portion of nearly hydrophobic aerosols was not contributed by BC-containing aerosols (BC-containing aerosols of 200 nm with BC core smaller than 80 nm which is smaller than the detection limit of SP2 likely to be quite aged in the air, thus not*

*possible to be nearly hydrophobic), but likely by FFOA- or BBOA-dominant aerosols (NFH also had a negative correlation with MFBBOA)." : sentence not clear. Shorten*

**Response: Thanks for your comments. We have revised this sentence as:**

**"At the same $MF_{FFOA}$, for example, when conditions of $MF_{FFOA} > 0.1$ were met, $NF_H$ (<0.7) demonstrated a noticeable decrease compared to $NF_{noBC}$ (>0.7), and $NF_H$ showed a negative correlation with both $MF_{BBOA}$ and $MF_{FFOA}$, suggesting that a substantial portion of nearly hydrophobic particles originated from FFOA- or BBOA-dominant rather than BC-containing particles. Additionally, markedly different correlations were observed between MAF and $MF_{FFOA}$ (r=-0.62), and between MAF and $MF_{BBOA}$ (r=-0.2), implying that nearly hydrophobic but CCN-active aerosols likely originated from biomass burning."**

*12. L483-484: "… between MAF and MFBBOA (r=-0.2) imply that nearly hydrophobic but CCN-active aerosols were likely contributed by biomass-burning emissions.": I don't understand this conclusion. You compare MAF and MFBBOA here, nothing about NFH. If you could still show this with the appropriate data it would need an interpretation as it is quite opposite to the expectations.*

**Response: Thanks for your comments. We have revised this sentence as shown in the response to the previous comment.**

*13. L489: "… suggesting that BC-containing aerosols emitted from fossil fuel combustion tended to be more externally mixed with other aerosol components than those emitted from __biomass burning__.": This is a good example where the finding should be put into the context of previous literature (as requested from a referee). This finding probably relates to the fact (described in the literature) that the BC fraction in fossil fuel combustion emissions is higher than in biomass burning emissions.*

**Response: Thanks for your comments. In a field campaign in the North China Plain, Zhang et al. (2020) reported that BC-containing particles originated from the source of fossil fuel tends to be more externally mixed than that originated from biomass burning. Thus our finding agree with the previous literature. We have revised this sentence as:**

**"… BC-containing particles emitted from fossil fuel combustion tended to be more externally mixed with other aerosol components than those emitted from biomass burning, which is consistent with the results of previous studies (Schwarz et al., 2008; Laborde et al., 2013; Liu et al., 2017; Zhang et al., 2020)."**

*14. L499-501: "… suggesting that BBOA contributed to nearly hydrophobic aerosols under subsaturated conditions; however, their hygroscopicity was enhanced, and they became CCN-active under supersaturated conditions." another finding without an attempt of an explanation. Could this be weakly soluble compounds that are soluble at high LWC, but not at low LWC?*

**Response: Thanks for your comments. The enhanced hygroscopicity of BBOA under supersaturated conditions may be attributed to:**

**(1) surface tension lowered by surface-active organic solutes (Hodas et al., 2016; Ruehl et al., 2016);**

**(2) variations of both particle diameter and surface tension due to liquid–liquid phase separation (Ovadnevaite et al., 2017; Liu et al., 2018);**

**(3) dissolution of sparingly soluble compounds at higher saturated conditions (Wex et al., 2009; Dusek et al., 2011);**

**(4) highly viscous organic aerosol which takes up water by surface water adsorption under sub-saturated conditions and by absorption of water under super-saturated conditions (Pajunoja et al., 2015);**

**We have added a sentence into the manuscript as:**

**"The enhanced hygroscopicity of BBOA under supersaturated conditions may be attributed to: (1) surface tension lowered by surface-active organic solutes (Hodas et al., 2016; Ruehl et al., 2016); (2) liquid–liquid phase separation (Ovadnevaite et al., 2017; Liu et al., 2018); (3) dissolution of sparingly soluble compounds at higher saturated conditions (Wex et al., 2009; Dusek et al., 2011); (4) highly viscous organic aerosol which takes up water by surface water adsorption under sub-saturated conditions and by absorption of water under super-saturated conditions (Pajunoja et al., 2015)."**

*15. L512-513: "… the secondary inorganic aerosol components dominated over SA (approximately 50% vs. approximately 70%), …": Not clear. Do you mean dominated over SOA? And what are these percentages?*

**Response: We are referring to that SIA rather than SOA dominate SA and the percentage is about 70%. We have revised this sentence as:**

**"… the secondary inorganic aerosol components dominated SA (the mass ratio between SIA and SA is approximately 70%), …"**

*16. L527-530: "The increase in NFnoBC at 200 nm as a function of the SA MF suggests that SAs migrated to a higher fraction of BC-free aerosols smaller than 200 nm to particle size of 200 nm, highlighting that SAs tended to form more quickly on BC-free aerosols than on BC-containing aerosols.": What would be the reason for SA condening more quickly on BC-free aerosols? Partitioning? But this depends also on the chemical composition of the rest of the 200-nm particle; the BC core can be as small as 80 nm. I believe this conclusion is only partly true. A higher mass fraction simply means that the primary particle size (before the condensation) is smaller, hence possibly a smaller BC size, and therefore possibly smaller than the dection limit of the SP2*

**Response: Thanks for your comments. We agree with the reviewer that this conclusion is partly true can there can be BC-containing particles with BC cores smaller than the detection limit of the SP2. As shown in Fig.5c of Li et al. (2023), secondary aerosol formations mainly add mass to BC-free particles (particles with BC lower than detection limit is not excluded), and similar results is found in this campaign as shown in Fig. R1. We now are preparing a manuscript to address this phenomenon from the review of chemical mechanisms, the main reason behind this phenomenon is that water uptake abilities of BC-containing particles are generally small while some BC-free particles would uptake a lot of water which provide site for chemical reactions. However, BC-containing particles with BC mass lower than detection limit at 200 nm likely have abundant hygroscopic coating materials which also favor greatly aerosol chemical compositions, therefore we agree with the reviewer that this conclusion is only partly true.**

**We have revised this part as:**

**"The increase in NF$_{noBC}$ at 200 nm as a function of the SA MF suggests that SAs migrated to a higher mass fraction of BC-free particles smaller than 200 nm to particle size of 200 nm, suggesting that SAs tended to form more quickly on BC-free particles than on BC-containing particles with BC higher than SP2 detection limit."**

[Figure]

**Figure R1: Scatter plots of SA / rBC and the ratio between total volume (Vtot) of BC-free and BC-containing aerosols to mass concentrations of rBC.**

*17. L557-558: "… and is likely formed mainly on BC-free particles.": see comment above on BC-free particles*

**Response: Thanks for your comments. Based on the response before, we have revised this sentence as:**

"…indicates that the difference is smaller when there is more OOA2, implying that OOA2 is also a semi-volatile compound and is likely formed mainly on BC-free particles (particles with BC mass lower than detection limit are not excluded)."

*18. L571-572: "significantly enhanced aerosol mixing state parameters": not clear. A mixing state parameter cannot be enhanced*

**Response: Thanks for your comments. We have revised this sentence as:**

"this increase in SAs significantly enlarged the value of aerosol mixing state parameters"

*19: L598: "However, the differences between these mixing state parameters **vary** significantly under different conditions." : use consistent past or present tense*

**Response: Thanks for your comments. We have revised it accordingly**

**Reviewer #1:**

*General comments:*

*Thanks for the revision. The paper looks much improved, now easy to follow, and readable to non-expert audiences. I agree with the author's argument that the correlation is not causality but this could be the best available way to understand the interlinks among various chemical-physical properties. I have some further questions as below and would suggest accepting with a minor revision.*

**Response: Thanks for your comments, we really appreciate for your time and careful inspection of our manuscript. Suggestions and comments are addressed point-by-point and corresponding responses are listed below.**

*1. Regarding the two OOA factors from PMF, are they related to Less-oxygenated and More-oxygenated OOAs?*

**Response: Terms of Less-oxygenated and More-oxygenated OOAs were used in previous studies when they have distinct O/C ratios. O/C ratios of OOA1 and OOA2 have a difference, however, their difference is small. Therefore, we used terms of OOA1 and OOA2 to avoid misleading.**

*2. Reviewer2, comment 8. In the last sentence of your paragraph. ...Thus, for the same measurement cycle (2h), more particle sizes were selected in the DMA-SP2 system to acquire the BC concentration...*

**Response: Thanks for your suggestion. We have revised this sentence accordingly.**

*3. Reviewer2, comment 9. Are you adjusting the flow rate to ensure enough aerosol particles in the inlet? I do agree with you that the flow rate doesn't influence mixing state measurements as it is not influenced by total mass. If yes, I suggested to add 1-2 sentences to clarify in the paper.*

**Response: We adjust the flow rate in the inlet in order to ensure accurate aerosol particle size cutoff. We have added these sentences into the manuscript as:**

**"The flow rate is carefully adjusted in the inlet in order to ensure accurate aerosol particle size cutoff."**

*4. Reviewer2, comment 20. According to Fig1 in Petters and Kreidenweis (2007), to activate a particle with kappa of 0.07, the aerosol particle size should be around 200-300nm. What are the composition and sources of these aerosol particles?*

**Response: In the NCP, aerosol with hygroscopicity kappa lower than 0.07 are likely to be dominated by BC and primary organic aerosol, from biomass and fuel combustion (Tao et al., 2021; Shi et al., 2022).**

**Reviewer #2:**

*General comments:*

*The manuscript has improved significantly compared to the first version. Thanks a lot for the improvement!*

*However, there is still some work, which needs to be done, mainly in the discussion of the results in section 3. Here, a lot of text describes too many similar looking figures. It needs to be indicated, what is really new. Other detailed comments are given below.*

**Response: Thanks for your comments. Suggestions and comments are addressed point-by-point and corresponding responses are listed below.**

*Furthermore, this study claims to use for the first time all these measurements and methods in parallel. But, what is the outcome? Which methods are comparable? Are all of them needed? E.g., if one has to reduce to setup to 2 mixing state parameters, which would you recommend? Any other general conclusions regarding the methodology?*

**Response: Thanks for your comments. We have added some discussions in sections 3.3 and 3.4 to better claim what is new and what is expected in the context of existing literatures.**

*Since this is definitely a new approach, it should be discussed and interpreted in the conclusion. Please also indicates in the conclusions which findings are new or you assume them to be new. There are too many 'findings' listed and the reader does not know, what is important.*

**Response: Thanks for your comments. Conclusions are modified to better deliver new and important findings of this study in the context of existing literatures, and recommendations are also concluded in this part as follow:**

[revised manuscript text omitted]

*One general formal comment. The unit liter has the abbreviation 'l', not the capital 'L'. This should be corrected through the whole text.*

**Response: According to a recent decision by the ACP Editorial board, the symbol L will be used for liter in all future publications to avoid confusion with the capital I.**

*The table with abbreviations also helps a lot. But could you please put it in an alphabetical order? This would be even better! And the term 'SA' is missing there. Maybe also indicate which parameters refer to certain diameters?!*

**Response: Thanks for your suggestions. We have revised this table as:**

**Table 1. Definition and description of abbreviations.**

| Abbreviation | Full name and/or Definition |
|---|---|
| BBOA | Biomass Burning Organic Aerosol
Characterized by obvious m/z 60 (mainly $C_2H_4O_2^+$) and 73 (mainly $C_3H_5O_2^+$), which are two indicators of biomass burning |
| $D_a$ | Midpoint activation diameter
Linked to the hygroscopicity of CCNs |
| $D_d$ | Particle diameter under dry conditions without humidification or heating |
| $D_p$ | Particle diameter after humidification or heating |
| GF | Growth factor
The ratio between particles with and without humidification and is linked to aerosol hygroscopicity |
| κ | Hygroscopicity parameter |
| MF | Mass Fraction |
| MAF | Maximum Activation Fraction
An asymptote of the measured SPAR curve at large particle sizes and represents the number fraction of CCNs to total particles |
| $NF_H$ | Number Fraction of Hydrophilic aerosol whose hygroscopicity parameter is >~0.07 at particle size of 50, 100, 150 and 200 nm |
| $NF_V$ | Number Fraction of Volatile aerosol whose Shrinkage Factor at 200 °C is <0.85 at particle size of 50, 100, 150 and 200 nm |
| $NF_{noBC}$ | Number Fraction of black carbon (BC)-free particles at particle size of 200, 250, 300 and 370 nm |
| $NF_{CBC}$ | Number Fraction of thickly coated BC particles at particle size of 200, 250, 300 and 370 nm |
| $NF_A$-$NF_B$ ($NF_{noBC}$-$NF_H$, $NF_V$-$NF_H$, $NF_{noBC}$-$NF_V$, $NF_V$-MAF, $NF_{noBC}$-MAF) | The difference between the number fraction of A and B at particle size of 200 nm |
| OOA1 and OOA2 | Two OOA factors resolved from the PMF analysis |
| PDF | Probability Distribution Function |
| $PM_{2.5}$ | Particulate Matter with an aerodynamic diameter <2.5 μm |
| $PM_1$ | Particulate Matter with an aerodynamic diameter <1 μm |
| POA | Primary Organic Aerosol
Summation of BBOA and FFOA |
| $R_{exBC}$ | The number concentration ratio of externally mixed BC particles in total BC-containing particles |

| | |
|---|---|
| | **Externally mixed BC particles are defined as identified bare/thinly coated BC-containing particles at particle size of 200, 250, 300 and 370 nm** |
| **SA** | **Secondary Aerosols, including nitrate, sulfate, ammonium and the two OOA factors** |
| **SF** | **Shrinkage Factor**
**The ratio between particles with and without heating and is linked to aerosol volatility** |
| **SIA** | **Secondary Inorganic Aerosols, including nitrate, sulfate, and ammonium** |
| **SOA** | **Secondary Organic Aerosols, including the two OOA factors** |
| **SPAR** | **Size-resolved Particle Activation Ratio**
**Size-dependent CCN activity under a specific SS** |

*Comments in detail:*

*Line number in the following mean the corresponding lines in the manuscript with tracked changes.*

*Line 158 – 160: please keep the old version to recognize the origin of the abbreviation.*

**Response: Thanks for your suggestion. We have revised it accordingly.**

*Line 223 ff: Do you think that AMS measurements and PNSD experience similar losses? If you say, that they agree well, this is hypothetical to my impression.*

**Response: Thanks for your comments. The agreement between AMS measurements and PNSD is confirmed in their comparison as shown in Figure S3:**

[Figure]

**Fig. S3. Comparison between aerosol volume concentration derived from measurements of PNSD and aerosol chemical compositions.**

*Line 346: How did you choose e.g., the critical GF? Did you plot for each diameter the PDF? Other studies used a common GFc for all diameters, why do you think this is different here? Does this correspond to Figure 2c?*

**Response: Thanks for your comments. The critical GFs are determined based on the GF PDF for each diameter. The critical GF in this study is different from those in other studies, because there is difference in aerosol micro-physical properties and we want to distinguish between different aerosol groups for comparison with aerosol groups inferred from measurements of other instruments. We have added corresponding description into the manuscript as:**

**"These values of GFC and SFC divide the probability density functions (PDFs) of SF and GF into two modes as shown in Figure 2c and 2d, consistent with prior NCP studies (Liu et al., 2011; Zhang et al., 2016), and may be different from those GFC and SFC in other studies because of the difference in aerosol micro-physical properties."**

*Line 361 and many other times: coating thickness of aerosols: it might be that I am a bit too picky here, but I think a coating thickness is always related to aerosol particles and not to aerosols (mixture of gas and particles), therefore I suggest to check the usage of the word 'aerosols'. I think in most cases it should be 'aerosol particles'.*

**Response: Thanks for your suggestion. We have checked the usage of the word 'aerosols' and revised the following:**

**"BC-containing aerosols" to "BC-containing particles"**

**"BC-free aerosols" to "BC-free particles"**

*Line 375, 425 and caption of Figure 1 and 3: The word 'compositions' should be 'components'. I do not see too much sense in 'mass concentrations of aerosol composition'*

**Response: Thanks for your comments. We have revised them accordingly.**

*Figures 4 – 9: Similar parameters are plotted, but I personally think, there are too many figures. For some the particle diameter (200 nm) is given, for others not in the caption. Please add this information in the figure caption. Are all figures really necessary? It would*

*be better to exclude one or two from the main paper or combine some of the results. The reader feels a bit overloaded with so many scatter dots.*

**Response: Thanks for your comments. After careful consideration, we have decided to retain all the figures in the manuscript. While we acknowledge that some figures may appear similar at first glance, each figure conveys distinct messages and contributes to the comprehensive understanding of our research findings. We believe that retaining all figures is crucial in presenting a thorough analysis and ensuring that all aspects of our study are adequately represented.**

**In addition, we have taken your suggestion into account and will ensure that each figure caption includes essential information, such as the particle diameter, for clarity and consistency.**

*Line 532 – 538: The sentence us too long and not understandable for me. Please avoid such long sentences with too much additional information in brackets.*

**Response: Thanks for your comments. We have revised this sentence as:**

**"At the same $MF_{FFOA}$, for example, when conditions of $MF_{FFOA}$ >0.1 were met, $NF_H$ (<0.7) demonstrated a noticeable decrease compared to $NF_{noBC}$ (>0.7), and $NF_H$ showed a negative correlation with both $MF_{BBOA}$ and $MF_{FFOA}$, suggesting that a substantial portion of nearly hydrophobic particles originated from FFOA- or BBOA-dominant rather than BC-containing particles. Additionally, markedly different correlations were observed between MAF and $MF_{FFOA}$ (r=-0.62), and between MAF and $MF_{BBOA}$ (r=-0.2), implying that nearly hydrophobic but CCN-active aerosols likely originated from biomass burning."**

*Line 569: the word 'compositions' should be components, as written in the corresponding caption of Figure 8.*

**Response: Thanks for your comments. We have revised it accordingly.**

*Line 633: better 'components' instead of 'compositions' Line 640: please use 'components' instead of 'compositions'*

**Response: Thanks for your comments. We have revised them accordingly.**

*Section 3, in particular 3.3 and 3.4 are too long and not well connected to other studies. What is really new in your study? Many correlations are obvious and well-known, here you*

*should compare with literature. E.g., that SA increases the hygroscopicity of hydrophobic particles is not new. There are similar examples. I would strongly recommend to remove some of the figures and reduce the text. For the results indicate the well-known facts with references or remove them and highlight those results which are new or opposite to former findings.*

**Response: Thanks for your comments. We have added discussions about the our results in the context of existing literatures as shown below. In addition, as mentioned in the response to the comment earlier, we have decided to retain all the figures in the manuscript after careful consideration and we have also revised the conclusion to better deliver new and important findings of this study.**

**L397: We have added a sentence as:**

**"This may be due to the high relative humidity during the pollution period, and the formation of SA occurs mainly in the aqueous phase, which contributes to the formation of particles with larger diameters (accumulation-mode and coarse-mode particles, Kuang et al., 2020)."**

**L489: We have revised this sentence as:**

[revised manuscript text omitted]

---

## Editor Decision (ED3)

The paper has substantially improved. I have one remaining comment that needs to be addressed:

L 409: It is not stated in the text how the mass fractions of the aerosol chemical components is measured. I would especially wonder how accurate the numbers for the size of 50 nm are.

Minor issues (red: corrections, red italic: comments):

L 31:     under the measured supersaturated conditions (*you cannot say anything about higher supersaturations*)

L 91:     formations

L 249:   after an denuder-bypass

L 442:   has a similar correlation coefficients (r~0.65) and a smaller systematic differences: *either singular or plural but not a combination of both*

L 503:   low soluble components: *incorrect English*

L 506:   Po¨hlker: *typo, also other instances*

L 548: SAs migrated to a higher mass fraction of BC-free particles smaller than 200 nm to particle size of 200 nm: *not clear*

L 553:   that contributes

L 557:   semi-volatile compound component

L 1194: curve at large particle: *incorrect English*

---

## Author Response (AR4)

**Editor:**

*Comments on Tao et al.*

*The paper has substantially improved. I have one remaining comment that needs to be addressed:*

**Response: Thanks for your comments, we really appreciate your feedback.**

**Suggestions and comments are addressed point-by-point and corresponding responses are listed below.**

*L 409: It is not stated in the text how the mass fractions of the aerosol chemical components is measured. I would especially wonder how accurate the numbers for the size of 50 nm are.*

**Response: Thanks for your comments. The mass fractions of aerosol compositions were calculated using the bulk measurements of the AMS, not size-resolved mass concentrations, because the measurements of AMS at 50 nm are quite noisy. We have added the description about the calculation of the mass fractions of the aerosol chemical components in L185 as follow:**

**"The mass fraction (MF) of each chemical composition is calculated as the bulk mass fraction of each chemical composition in in non-refractory $PM_1$ (NR-$PM_1$)."**

*Minor issues (red: corrections, red italic: comments):*

*L 31:* under the measured supersaturated conditions *(you cannot say anything about higher supersaturations)*

**Response: Thanks for your comments. We have revised it accordingly.**

*L 91:* formations

**Response: Thanks for your comments. We have revised it accordingly.**

*L 249:* after an denuder-bypass

**Response: Thanks for your comments. We have revised it accordingly.**

*L 442:* has a similar correlation coefficients (r~0.65) and a smaller systematic differences*: either singular or plural but not a combination of both*

**Response: Thanks for your comments. We have revised it as:**

**"has a similar correlation coefficient (r~0.65) and a smaller systematic difference"**

*L 503:* low soluble components*: incorrect English*

**Response: Thanks for your comments. We have revised it as "poorly water soluble substances".**

*L 506:* Po˙hlker*: typo, also other instances*

**Response: Thanks for your comments. We have revised it as "Pöhlker" and that in L 511.**

*L 548: SAs migrated to a higher mass fraction of BC-free particles smaller than 200 nm to particle size of 200 nm: not clear*

**Response: Thanks for your comments. We have revised it as:**

**"SAs migrated higher mass fraction of BC-free particles with particle size smaller than 200 nm to particle size of 200 nm"**

*L 553:* that contributes

**Response: Thanks for your comments. We have revised it accordingly.**

*L 557:* semi-volatile  component

**Response: Thanks for your comments. We have revised it accordingly.**

*L 1194:* curve at large particle*: incorrect English*

**Response: Thanks for your comments. We have revised it as "curve at large particle size".**